# The mouse cortico–basal ganglia–thalamic network

Nicholas N. Foster[1,2 ✉], Joshua Barry[3], Laura Korobkova[2], Luis Garcia[1,2], Lei Gao[1,2], Marlene Becerra[2], Yasmine Sherafat[2], Bo Peng[4], Xiangning Li[5,6], Jun-Hyeok Choi[7], Lin Gou[1,2], Brian Zingg[1,2], Sana Azam[2], Darrick Lo[1,2], Neda Khanjani[2], Bin Zhang[1,2], Jim Stanis[2], Ian Bowman[1,2], Kaelan Cotter[2], Chunru Cao[1,2], Seita Yamashita[1,2], Amanda Tugangui[1,2], Anan Li[5,6,8], Tao Jiang[6], Xueyan Jia[6], Zhao Feng[6], Sarvia Aquino[2], Hyun-Seung Mun[1,2], Muye Zhu[1,2], Anthony Santarelli[2], Nora L. Benavidez[1,2], Monica Song[1,2], Gordon Dan[2], Marina Fayzullina[1,2], Sarah Ustrell[2], Tyler Boesen[1,2], David L. Johnson[2], Hanpeng Xu[1,2], Michael S. Bienkowski[2], X. William Yang[3,9], Hui Gong[5,6,8], Michael S. Levine[3], Ian Wickersham[10], Qingming Luo[5,6,11], Joel D. Hahn[12], Byung Kook Lim[7], Li I. Zhang[4], Carlos Cepeda[3], Houri Hintiryan[1,2] & Hong-Wei Dong[1,2 ✉]

The cortico–basal ganglia–thalamo–cortical loop is one of the fundamental network motifs in the brain. Revealing its structural and functional organization is critical to understanding cognition, sensorimotor behaviour, and the natural history of many neurological and neuropsychiatric disorders. Classically, this network is conceptualized to contain three information channels: motor, limbic and associative[1–4]. Yet this three-channel view cannot explain the myriad functions of the basal ganglia. We previously subdivided the dorsal striatum into 29 functional domains on the basis of the topography of inputs from the entire cortex[5]. Here we map the multi-synaptic output pathways of these striatal domains through the globus pallidus external part (GPe), substantia nigra reticular part (SNr), thalamic nuclei and cortex. Accordingly, we identify 14 SNr and 36 GPe domains and a direct cortico-SNr projection. The striatonigral direct pathway displays a greater convergence of striatal inputs than the more parallel striatopallidal indirect pathway, although direct and indirect pathways originating from the same striatal domain ultimately converge onto the same postsynaptic SNr neurons. Following the SNr outputs, we delineate six domains in the parafascicular and ventromedial thalamic nuclei. Subsequently, we identify six parallel cortico–basal ganglia–thalamic subnetworks that sequentially transduce specific subsets of cortical information through every elemental node of the cortico–basal ganglia–thalamic loop. Thalamic domains relay this output back to the originating corticostriatal neurons of each subnetwork in a bona fide closed loop.

The striatum, pallidum and substantia nigra are key components of the basal ganglia, and they process inputs from the entire neocortex[5,6]. They constitute a critical node in the cortico–basal ganglia–thalamo–cortical loop[1,7–12]. This recurrent network is associated with diverse functions and behaviours[13–17], and aberrant basal ganglia function is implicated in movement disorders[17–19], neuropsychiatric disorders[20–22] and drug addiction[23]. Identifying the specific subnetworks within the loop is key to understanding how this multitude of functions and pathologies is governed. The consensus view is there are three parallel channels of information flow through the basal ganglia: associative, limbic and sensorimotor[1–4] (Extended Data Fig. 1a). Previous efforts to refine the three-channel model have postulated a number of specific parallel subnetworks[1,11]. Yet it has not been possible to provide a definitive wiring diagram to support a more refined model owing to incomplete basal ganglia connectional data at sufficiently high spatial resolution. However, a recent multi-scale network organization of the mouse corticostriatal pathway subdivided the caudoputamen (CP) into 29 fine-scale network divisions termed domains[5] (Extended Data Fig. 1a–c), a finding

[1]UCLA Brain Research and Artificial Intelligence Nexus, Department of Neurobiology, David Geffen School of Medicine at UCLA, Los Angeles, CA, USA. [2]Mark and Mary Stevens Institute for Neuroimaging and Informatics, Keck School of Medicine, University of Southern California, Los Angeles, CA, USA. [3]Jane and Terry Semel Institute for Neuroscience and Human Behavior, Department of Psychiatry and Biobehavioral Sciences, David Geffen School of Medicine at UCLA, Los Angeles, CA, USA. [4]Zilkha Neurogenetic Institute, Keck School of Medicine, University of Southern California, Los Angeles, CA, USA. [5]Britton Chance Center for Biomedical Photonics, Wuhan National Laboratory for Optoelectronics, MoE Key Laboratory for Biomedical Photonics, School of Engineering Sciences, Huazhong University of Science and Technology, Wuhan, China. [6]HUST-Suzhou Institute for Brainsmatics, JITRI Institute for Brainsmatics, Suzhou, China. [7]Neurobiology Section, Division of Biological Sciences, University of California, San Diego, La Jolla, CA, USA. [8]CAS Center for Excellence in Brain Science and Intelligence Technology, Chinese Academy of Science, Shanghai, China. [9]Center for Neurobehavioral Genetics, Jane and Terry Semel Institute for Neuroscience, Los Angeles, CA, USA. [10]McGovern Institute for Brain Research, Massachusetts Institute of Technology, Cambridge, MA, USA. [11]School of Biomedical Engineering, Hainan University, Haikou, China. [12]Department of Biological Sciences, University of Southern California, Los Angeles, CA, USA. ✉e-mail: nnfoster@mednet.ucla.edu; HongWeiD@mednet.ucla.edu

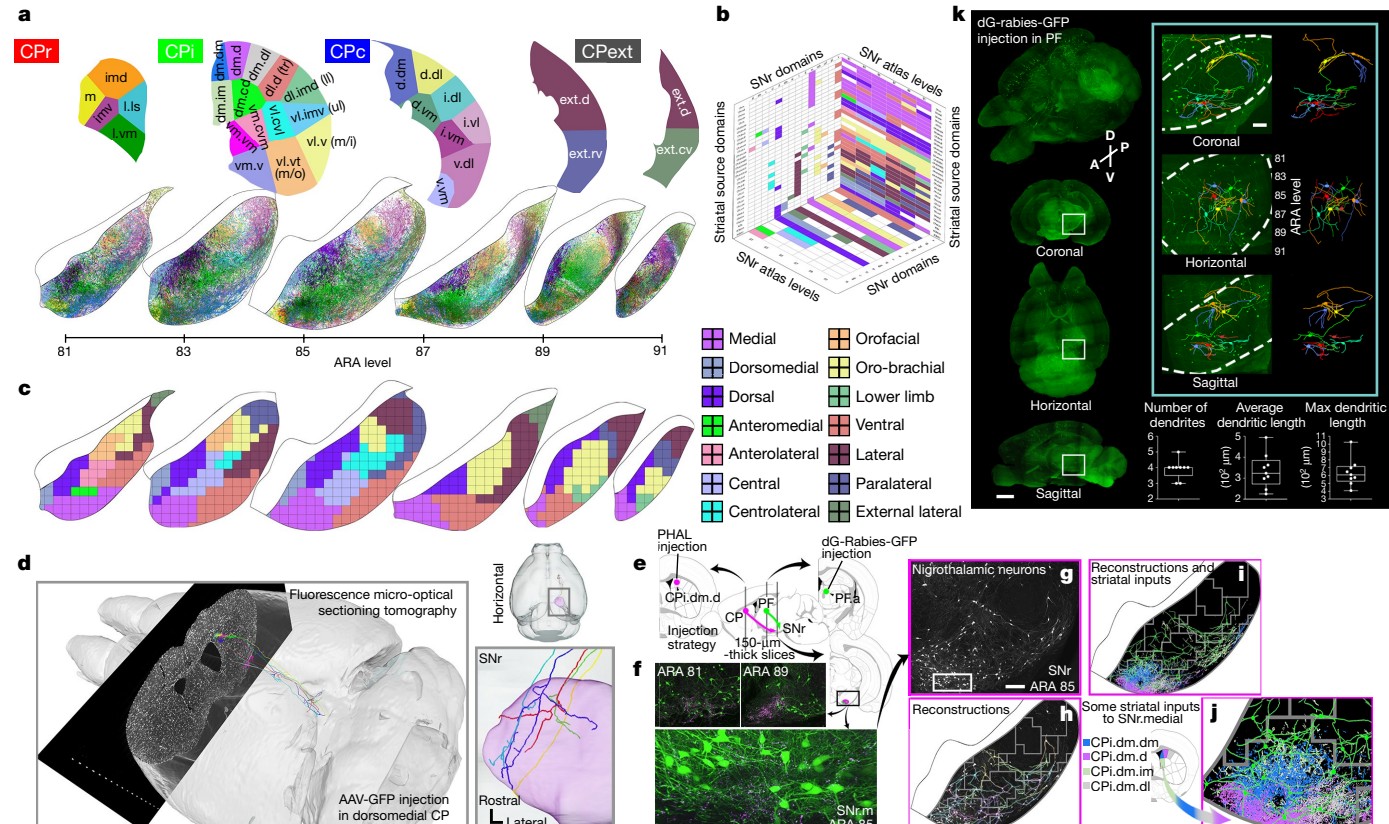

**Fig. 1 | SNr domain structure. a**, Reconstructed axonal projections from each striatal domain (top) are coloured to match their source domain and projected onto SNr maps (bottom) from six representative levels of the Allen Reference Atlas[51] (ARA). **b**, 3D striatonigral matrix (see Extended Data Fig. 7). **c**, SNr domain map derived from analysis of axonal projections in **a**. **d**, fMOST imaging reveals that individual striatal neurons project through the entire frontal axis of SNr (see also Extended Data Fig. 9). **e**–**g**, Anterograde tracer (PHAL, pink) in striatal domain CPi.dm.d and retrograde tracer (dG-rabies-GFP, green) in thalamic domain PF.a (**e**) reveal that input striatonigral axonal topography matches output nigrothalamic neuronal topography from rostral through caudal SNr.medial domain (**f**, **g**). Region outlined in **g** is expanded (**f**, bottom). Images are representative of $n = 2$ biological replicates. Scale bars, 30 μm (**f**), 200 μm (**g**). **h**–**j**, Reconstructions of these neurons (**h**) are shown with 4 striatal inputs (**i**), with the bottom left region enlarged in **j**. **k**, Rabies-labelled, SHIELD-cleared brain reveals whole SNr neurons. Right, magnified view of outlined regions. Scale bars, 1 mm (left), 200 μm (right). Collectively, images in **e**–**k** suggest SNr neurons are capable of integrating most inputs to their domain at a given coronal level. Max, maximum.

that suggests that there is a more granular level of organization of the basal ganglia. Here we systematically map the multi-synaptic output pathways of all CP domains through each sequential node of the loop.

## Results

### Striatonigral pathway defines SNr domains

Data for the striatal output pathway analysis were produced by injection of anterograde tracer into each CP domain[5] as well as the core (ACBc), medial shell and lateral shell of nucleus accumbens (Extended Data Fig. 1d), yielding a representative set of 36 injections (Extended Data Fig. 2). Coronal images containing GPe and SNr were registered, segmented, reconstructed and quantified. Network analysis of the data partitioned the striatal domains with convergent axonal fields into communities, and their terminal zones were demarcated as a new domain of GPe or SNr. The axonal reconstructions were used to make projection maps that demonstrate the underlying axonal data defining each new domain (Extended Data Fig. 1e–k).

The topographical organization of the striatonigral (direct) pathway is depicted in the projection maps (Fig. 1a, Extended Data Figs. 3a, 4a) and aligns well with data from rat and monkey[8,10,12,24]. Following network analysis of these data, we identified 14 domains in SNr (Fig. 1b, c, Extended Data Figs. 5–7, Supplementary Table 1). Most SNr domains receive convergent inputs from multiple striatal domains (Supplementary Table 1), each of which in turn receives a unique set of distinct

cortical inputs[5]. For instance, SNr dorsal (SNr.d) and dorsomedial (SNr.dm) are limbic domains that collectively receive inputs from a number of limbic striatal domains (Extended Data Fig. 8a, b). In turn, these striatal domains receive inputs from limbic cortical areas that are themselves interconnected and constitute the lateral cortico-cortical subnetworks[5,25], which are involved in perception of internal states and memory associated with emotion[25,26]. The majority of SNr domains span two or more rostrocaudal levels and several run the entire length of SNr, reflecting their inputs: striatonigral projections terminate in discrete zones within SNr that are restricted mediolaterally and dorsoventrally, but most (75%) span the entire rostrocaudal extent of SNr, forming longitudinal columns of terminal axons (Extended Data Fig. 2, 3b). Individual striatal axons conform to this topography, which was confirmed at the single-cell level using fluorescence micro-optical sectioning tomography (fMOST) (Fig. 1d, Extended Data Fig. 9) and is consistent with primate data[27].

Graphic reconstruction of SNr neurons reveals a dendritic morphology that allows for receipt of convergent striatal inputs from multiple CP domains by individual SNr neurons. Thalamic injection of GFP-labelled glycoprotein-deleted rabies (dG-rabies-GFP) labels neurons with extensive dendritic arbors in the SNr medial domain (SNr.m) (Fig. 1f). Reconstructions of these neurons are shown overlaid onto a composite projection map of four convergent striatal terminal fields (Fig. 1g–j). Most of these neurons' dendritic arbors appear capable of contacting all four terminal fields. However, these neurons were

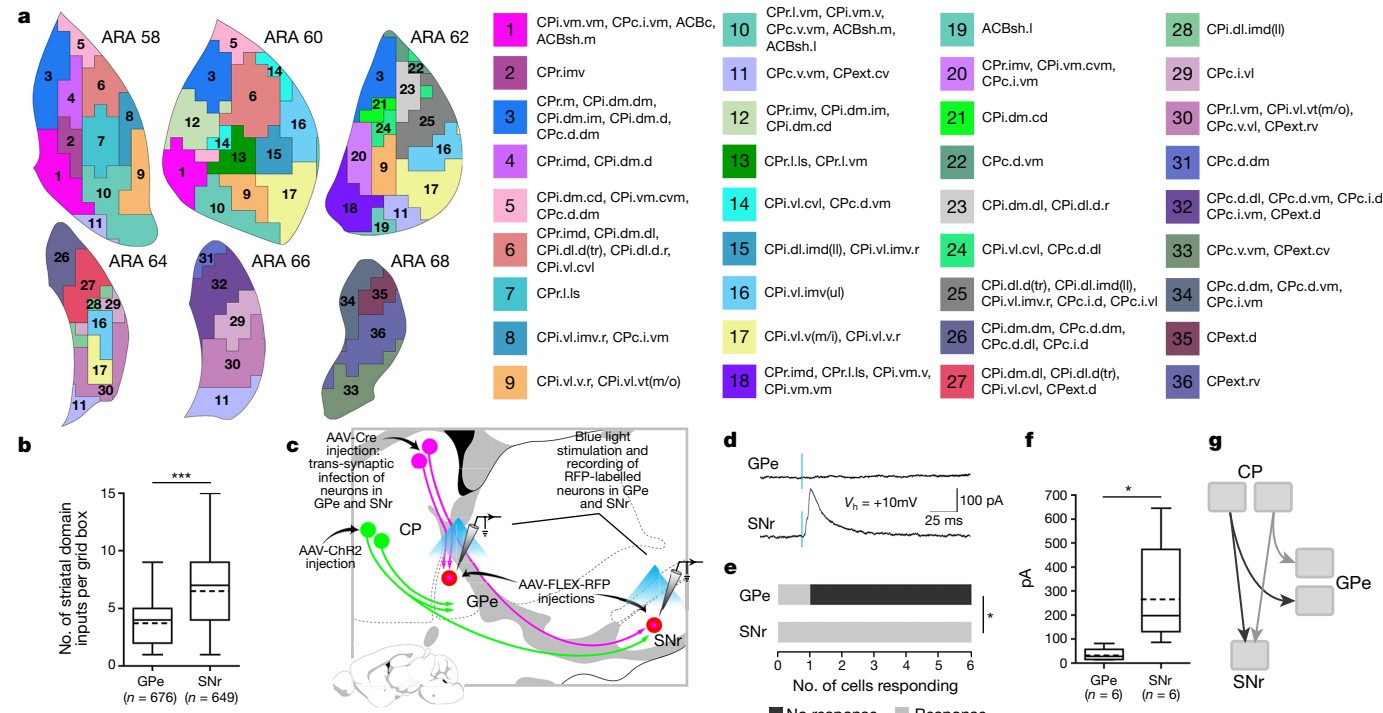

**Fig. 2 | Indirect pathway defines GPe domain structure and differs from direct pathway. a**, Striatal axonal inputs (see Extended Data Fig. 11) define GPe domains. Striatal inputs are listed on the right. **b**, Analysis of grid box data reveals that the mean number of striatal domains providing inputs to each grid box differs significantly between GPe and SNr, indicating greater convergence in the direct pathway (GPe: 3.75 ± 1.79, SNr: 6.49 ± 2.97; $P < 0.0001$, two-tailed $t$-test, $t = 20.26$, d.f. = 1,057; data are mean ± s.d.). Total numbers of grid boxes in GPe ($n = 676$) and SNr ($n = 649$) are similar, meaning their gross volumes are similar. **c**, Schematic of experimental strategy for electrophysiological test of parallelism in indirect pathway and convergence in direct pathway. **d**, Example recording traces from RFP-tagged neurons in GPe and SNr. **e**, Number of neurons responding to stimulus was significantly different between GPe ($n = 6$) and SNr ($n = 6$; $P = 0.0152$, Fisher's exact test, 3 mice). **f**, Peak amplitude of current after stimulation was significantly different between GPe and SNr neurons (GPe: 33.67 ± 24.74 pA, SNr: 267.5 ± 197.6 pA; $P = 0.0348$, two-tailed $t$-test, $t = 2.876$, d.f. = 5; data are mean ± s.d.). **g**, Simplified schematic of the direct and indirect pathways. In box plots, boxes demarcate first and third quartiles, whiskers show minimum and maximum values, the solid line is the median and the dashed line is the mean.

reconstructed from a tissue slice 150 μm thick, resulting in truncation of the dendrites in the rostrocaudal axis. Full SNr neuronal reconstructions in SHIELD-cleared intact tissue reveal that their dendrites spread partway into adjacent rostral and caudal levels, but do not span the entire length of SNr (Fig. 1k). Collectively, these data suggest that SNr projection neurons are capable of integrating most inputs to their domain at a given coronal level, but not across the rostrocaudal length of the large domains. Thus the small changes in striatal input structure across a domain at different rostrocaudal levels (Supplementary Table 1) could result in slightly different integration profiles of neurons along the length of a domain.

Finally, we traced a striato–nigro–thalamic pathway (Fig. 1e) by injecting the anterograde tracer *Phaseolus vulgaris* leucoagglutinin (PHAL) in striatal domain CPi.dm.d and retrograde tracer dG-rabies-GFP in the associative subregion of the parafascicular thalamic nucleus. The results reveal close and extensive juxtaposition of anterogradely labelled striatal axons and retrogradely labelled SNr perikarya through the entire rostrocaudal extent of SNr.m (Fig. 1f). The matching input–output structure of striatal terminals and thalamic-projecting nigral neurons supports the notion of longitudinal SNr domains as a functional unit.

## Parallel striatal projections to GPe

Network analysis of the striatopallidal (indirect) pathway data reveals 36 domains in GPe (Fig. 2a, Extended Data Fig. 10, Supplementary Table 2). The majority of domains (64%) have only one or two inputs, and most domains (69%) span just one coronal level of GPe. The majority of striatal domains project to GPe with restricted terminal fields that have little overlap and convergence (Extended Data Figs. 4, 11), indicating that the indirect pathway is characterized by a higher degree of

specificity and parallelization compared with the direct pathway. This is reflected in the projection maps, in which more individual colours can be seen in GPe and there is more overlap of coloured terminal fields in SNr (compare Fig. 1a, Extended Data Fig. 11a). To quantify this difference, convergence of striatal inputs in the grid box datasets of GPe and SNr was analysed by tallying the number of striatal domains terminating in each grid box (Extended Data Fig. 12a). Frequency distributions of these data show that SNr receives more inputs per grid box than GPe (compare Extended Data Fig. 12b, g), approximately two times more, on average ($P < 0.0001$; Fig. 2b).

To functionally validate this, we performed a recording experiment combining channelrhodopsin (ChR2)-assisted circuit mapping (CRACM) with anterograde transsynaptic tracing. One CP domain was injected with adeno-associated virus (AAV) expressing ChR2 (AAV-ChR2) and a separate CP domain was injected with AAV1-Cre. The AAV1-Cre is transported and infects postsynaptic neurons in GPe and SNr, which are visualized with injections of AAV-FLEX-RFP (Fig. 2c). The ChR2-labelled axons should colocalize with RFP-labelled neurons in SNr but not in GPe; hence RFP neurons in GPe should not respond to stimulation of ChR2 axons, whereas RFP neurons in SNr should respond. Patch clamp recordings of RFP-labelled neurons were consistent with this prediction, with nearly all GPe neurons (5 out of 6) showing no response and all SNr neurons (6 out of 6) showing a response to stimulation ($P = 0.0152$; Fig. 2d, e), and peak current after stimulation was also significantly different ($P = 0.0348$; Fig. 2f). Notably, both nigral and pallidal neurons were recorded from all subjects, enabling direct within-animal comparisons. These anatomical and functional findings support a model of indirect pathway parallelization and direct pathway convergence (Fig. 2g).

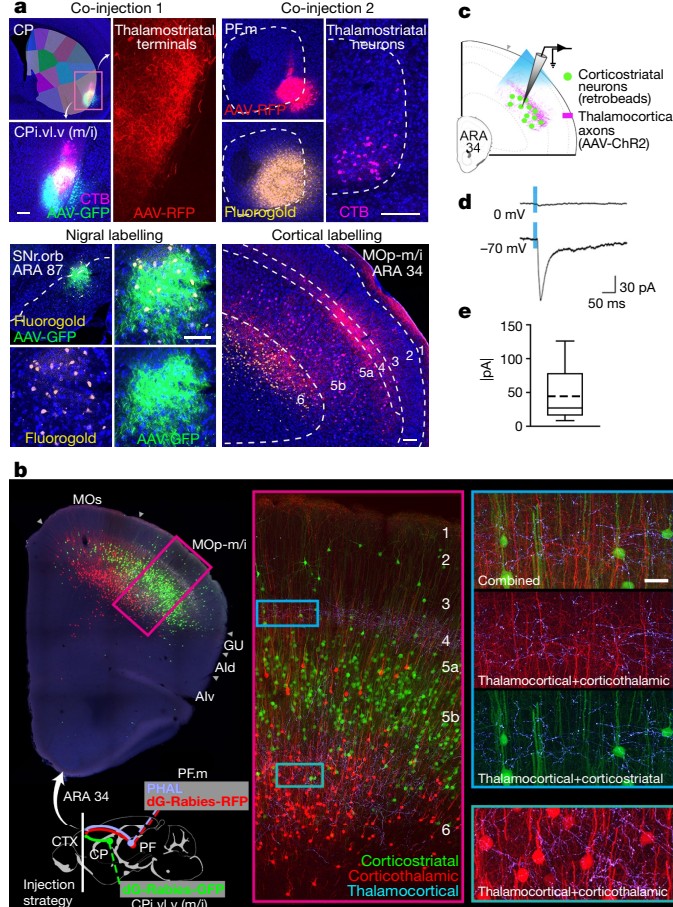

**Fig. 3 | Demonstration of the oro-brachial subnetwork loop. a**, Double co-injection labels the loop structure of the mouth component of the oro-brachial subnetwork loop. Co-injections of anterograde–retrograde tracer pairs into striatal and thalamic oral domains (CPi.vl.v and PF.m) reveal overlapping labelling in SNr oro-brachial domain and mouth primary motor cortex MOp-m/i. Images are representative of $n = 6$ replicates. Scale bars, 100 μm. CTB, cholera toxin subunit B–Alexa Fluor 647 conjugate. **b**, Labelling of corticostriatal (green) and corticothalamic (red) neurons and thalamocortical axons (light blue) from the striatal and thalamic mouth domains reveals two zones of overlap of labelling (insets), most densely in layer 4 of MOp-m/i cortex where thalamocortical axons overlap the apical dendrites of corticostriatal and corticothalamic neurons. Representative of $n = 2$ replicates. Scale bar, 30 μm. **c**, Mice ($n = 2$) were injected with AAV-ChR2 into PF.m to label thalamocortical axons with channelrhodopsin and fluorophore-tagged retrobeads into CPi.vl.v to retrogradely label corticostriatal neurons. **d**, **e**, In acute slice preparation, the majority of recorded labelled neurons (9 out of 13) showed excitatory postsynaptic current response to stimulation when clamped at −70 mV (44.89 ± 42.8 pA (mean ± sd)), and none showed inhibitory postsynaptic current at 0 mV. Box plot as described in Fig. 2.

## Direct and indirect pathways converge in SNr

Direct and indirect pathway neurons are intermingled in striatum[28], such that one domain of CP projects to specific domains of both SNr and GPe. The GPe relays indirect pathway information to SNr via a strong, direct projection[29]. Yet the topography and specificity of how these parallel pathways diverge and re-converge are unknown. The direct pathway has a 'bridging collateral' to GPe en route to the nigra[30]. We used Cre-dependent tracing in D1- and A2A-Cre mice to demonstrate that terminals of both direct and indirect pathways in GPe have the same topography when arising from the same striatal source, and fMOST imaging shows the same pattern at the single-cell level (Extended Data Fig. 13).

We next show that pallidonigral projections from a GPe domain converge with striatonigral axons arising from the same CP domain that serves as input source to both nuclei. Injection of anterograde tracer in limbic GPe domains 18 and 20 labels projections to nigral limbic domain SNr.d and limbic domains of CP and thalamus (Extended Data Fig. 8c–g). GPe.18 and 20 and SNr.d receive inputs from the same striatal domains (CPi.vm.v, CPi.vm.vm and CPi.vm.cvm)[5]. We validated this finding in a second pathway with a retrograde tracer injection in SNr.m (Extended Data Fig. 14). Labelling is seen in GPe.3, and in CP domains that innervate both SNr.m and GPe.3. Thus direct and indirect pathways arising from a common striatal source converge in SNr.

To ascertain whether homotypic direct and indirect pathways actually synapse onto the same postsynaptic neurons in SNr, we performed a CRACM–anterograde transsynaptic tracing experiment (Extended Data Fig. 8h). AAV1-Cre was injected into CP, which transsynaptically infected neurons in GPe and SNr, causing those cells to express Cre, and AAV-DIO-ChR2 was injected into GPe and AAV-FLEX-RFP was injected into SNr. RFP-expressing neurons in SNr were recorded during optical stimulation of pallidonigral terminals, evoking an inhibitory response that was capable of suppressing nigral neuronal activity (Extended Data Fig. 8i, j). The majority of recorded SNr neurons (10 out of 14) showed this inhibitory response, demonstrating re-convergence of separately processed information from the direct and indirect pathways onto individual SNr neurons (Extended Data Fig. 8k, l).

## Parallel output channels of parafascicular thalamus

The thalamus is the final node in the cortico–basal ganglia–thalamic loop. Two of the densest nigral outputs are to the parafascicular (PF) and ventromedial (VM) thalamic nuclei. The PF is intricately interconnected with the other nodes of the loop, such that topographically connected subregions of cortex, striatum and nigra connect topographically with a discrete PF subregion[4,31,32] (Extended Data Fig. 15a–c). Injections of anterograde and retrograde tracers in cortex, CP and SNr demonstrate this motif, showing highly specific connectivity patterns with six subregions of PF (Extended Data Fig. 15d, e). These subregions appear to be parallel output channels for integrating basal ganglia efferent signals with cortical inputs and conveying that computation to striatum and cortex. They correspond to associative (PF.a), trunk and lower limb (PF.tr/ll), upper limb (PF.ul), mouth (PF.m), limbic (PF.lim) and ventral striatal (PF.vs) domains.

For example, the ventral striatal subnetwork (Extended Data Fig. 15d, e, ventral striatal) contains infralimbic (ILA) and medial orbitofrontal (ORBm) cortex, which are interconnected and both project to ACBc and PF.vs; the ACBc also receives input from PF.vs and projects to SNr.dm, which in turn projects to PF.vs; the PF.vs projects back up to ILA, closing the loop. Anterograde transsynaptic tracing shows the actual synaptic specificity of the ACBc–SNr.dm–PF.vs pathway: AAV1-Cre was injected into ACBc and AAV-FLEX-RFP into medial SNr (Extended Data Fig. 16). Labelled neurons are seen specifically in SNr.dm, and their axonal labelling terminates precisely where the other nodes of the ventral striatal subnetwork connect, the PF.vs (and VM.vs; Extended Data Fig. 15d–g, SNr.dm).

## VM has distinct cortical innervation topography

The SNr also projects to VM, and its terminals there are even denser than to PF. The VM also contains six output channels (Extended Data Fig. 15f–h), but with a different organizational scheme: unlike PF, with separate domains for the body sub-regions, VM has one domain (VM.s) projecting to secondary motor cortex (MOs) and another (VM.p) projecting to primary motor (MOp) and primary somatosensory (SSp) cortex. The VM.s domain projects to all somatic subregions (that is, ul, ll and tr) within MOs, whereas VM.p projects to all somatic subregions within MOp and SSp (Extended Data Fig. 15g, h). A separate domain for the mouth pathways, VM.m, projects specifically to mouth and head regions of MOp, SSp and MOs, similar to PF.m. The associative

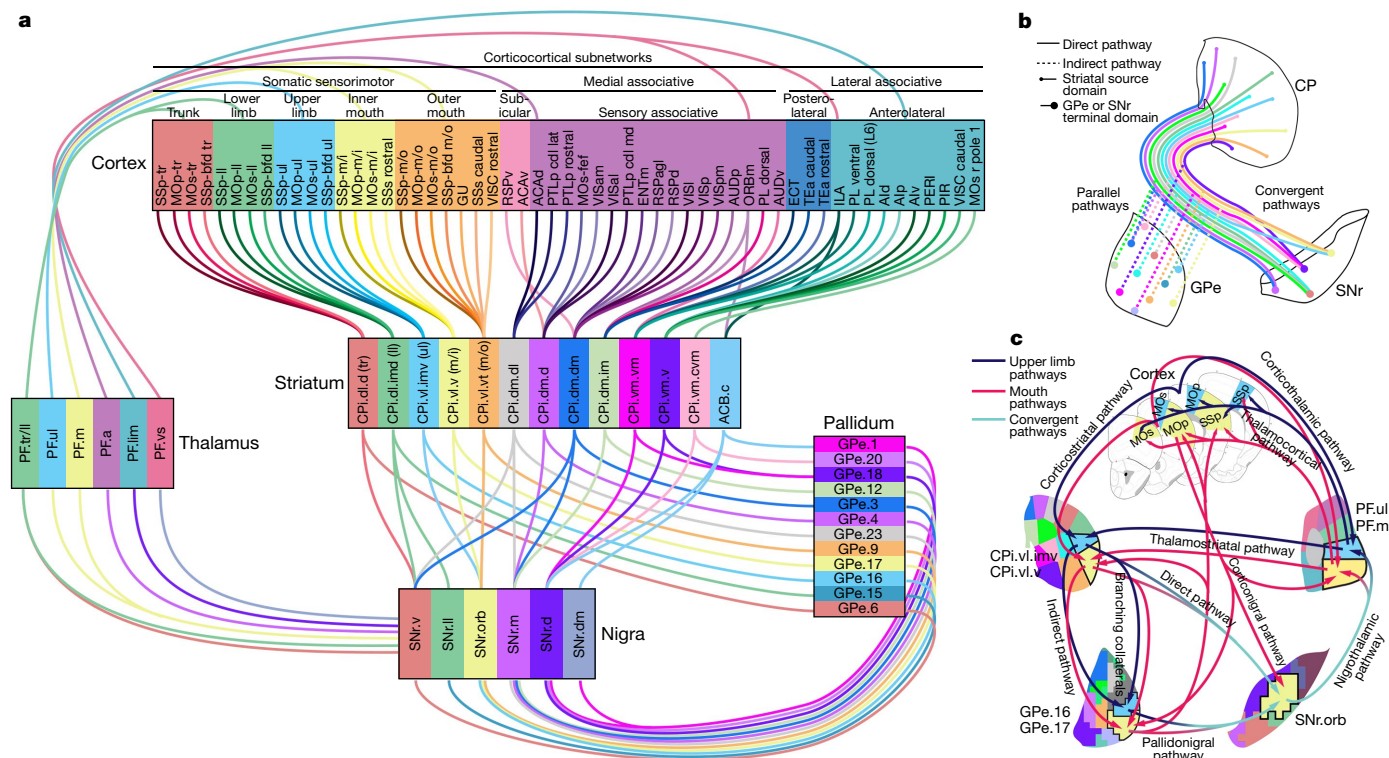

**Fig. 4 | Summary models of main findings. a**, Cortico–basal ganglia–thalamic loop model. In cortex, two medial associative subnetworks of exteroceptive sensory areas, two lateral associative subnetworks of interoceptive limbic areas and five somatic sensorimotor subnetworks of body regions project into largely distinct striatal subnetworks, whose outputs form the more parallel indirect (striatopallidal) pathway and the more convergent direct (striatonigral) pathway. The pallidal domains send convergent projections to the same nigral domains targeted by their input striatal domains. The nigral domains then project to six regions of the parafascicular thalamus, which in turn are interconnected with the originating cortical regions. **b**, Model of striatal output topography, illustrating convergence in the direct pathway and parallelism in the indirect pathway. Note that the small and large points represent the true topographic patterns of source and target zones, respectively. **c**, The oro-brachial subnetwork model highlights the precise separation between parallel pathways, the rich interconnectivity within pathways, and the complexity of the cortico–basal ganglia circuit. Not all pathways are labelled or depicted. Cortical areas: AI, agranular insular; ECT, ectorhinal; ENT, entorhinal; GUS, gustatory; PERI, perirhinal; PIR, piriform; PL, prelimbic; PTLp, posterior parietal association; RSP, retrosplenial; TEa, temporal association; VIS, visual; VISC, visceral.

(VM.a), limbic (VM.lim) and ventral striatal (VM.vs) channels are similar in topography to homologous regions of PF. The boundaries of each of these domains were established by the position of the thalamo-cortical neuron groups (Extended Data Fig. 15g). Each VM domain receives a specific nigral input from at least one of the SNr domains, and this nigrothalamic input conforms to the VM domain boundaries established by thalamocortical tracing (Extended Data Fig. 15g, h).

## Whole cortico–basal ganglia–thalamic loops

The cortico–basal ganglia–thalamic loop model[1,2,18] is supported by electrophysiological and anatomical experiments that demonstrate segments of this network[9–11,31,33–35]. Yet whole circuitous loops have not been demonstrated within a single animal using any methodology. Moreover, calling the network a 'loop' begs the question whether it is in fact a closed-circuit loop.

We first demonstrate whole loops using the double co-injection technique for mapping interconnected network structures (Fig. 3a, Extended Data Fig. 17b). Two anterograde–retrograde tracer pairs were injected into two non-adjacent nodes of the oro-brachial subnetwork. In a serially connected four-node loop, labelling from the co-injection pairs will converge and appose in the non-injected nodes. Injections into CP and PF orofacial regions reveal overlapping labelling in the SNr oro-brachial domain (SNr.orb) and a cortical column in mouth primary motor region MOp-m/i (Fig. 3a). The same strategy demonstrates the associative subnetwork loop (Extended Data Fig. 17a). Moreover, when co-injections are placed in nodes of two separate but neighbouring

loops, labelling is seen in separate, neighbouring regions of SNr and cortex (Extended Data Fig. 18), illustrating the largely parallel nature of these loops.

The colocalization of thalamocortical axons and corticostriatal neurons is strongly suggestive of recurrent feedback within a subnetwork loop. In the oro-brachial subnetwork, this probable closed-circuit zone lies in MOp-m/i (Fig. 3b). There, corticostriatal neurons providing input to CPi.vl.v have apical dendrites that ascend through a dense field of axons from PF.m terminating in layer 4 (Fig. 3b, insets; Supplementary Video 1). To unambiguously determine whether the loop is truly recurrent, we injected AAV-ChR2 into PF.m to opsin-label thalamocortical axons and fluorescent retrobeads into CPi.vl.v to retrogradely label corticostriatal neurons (Fig. 3c). Labelled corticostriatal neurons were patch-clamped in acute slice preparation during blue light stimulation. The majority (9 out of 13) showed an excitatory postsynaptic current to stimulation (Fig. 3d, e). The associative subnetwork loop is also demonstrated this way (Extended Data Fig. 17c). With the majority of recorded neurons exhibiting a specific monosynaptic excitatory response to thalamocortical stimulation, the cortico–basal ganglia–thalamic loop contains a substantial closed-circuit component.

## Cortex sends a direct projection to SNr

The hyperdirect pathway from cortex to subthalamic nucleus (STN) to SNr was thought to be the fastest route for cortical information to reach SNr[36]. We demonstrate that certain cortical areas of the oro-brachial subnetwork project directly to SNr.orb (Extended Data Fig. 19).

The corticonigral pathway is seen in sagittal section by injection of AAV1-Cre into MOp-m/i of Ai14 mice, labelling fine axonal fibres in caudal SNr (Extended Data Fig. 20a–d). This pathway was validated by injecting SNr.orb with AAVretro-Cre and MOp-m/i with AAV-FLEX-RFP (Extended Data Fig. 20e–g). Again, cortical axons are seen impinging deeply into SNr with topographic specificity (Extended Data Fig. 20g, h). Collaterals from this pathway are seen in STN and oromotor regions of CP, GPe and PF (Extended Data Fig. 20i–l). To verify that corticonigral axons bear boutons, we injected an AAV inducing expression of GFP-tagged synaptophysin and cytoplasmic RFP into MOp-m/i, labelling red corticonigral axons bearing green boutons in SNr.orb (Extended Data Fig. 20m). Finally, this pathway was functionally validated with antero-grade transsynaptic tracing, since only functional synapses transmit AAV. Injection of AAV1-Cre in MOp-m/i of Ai14 mice labels postsynaptic SNr neurons (Extended Data Fig. 20c). This characterization of the corticonigral pathway suggests that cortex can directly activate all components of the oro-brachial subnetwork (Extended Data Fig. 20n).

## Discussion

This work reveals that the canonical cortico–basal ganglia–thalamic network is composed of six parallel subnetworks, each of which is organized by a number of nodes that are precisely and richly intercon-nected (Fig. 4a). A model of the oro-brachial subnetwork exemplifies this interconnectivity (Fig. 4c). This comports with recent findings demonstrating specific parallel basal ganglia–thalamo–cortical pathways[31]. Together, the data presented here are consistent with a cortico–basal ganglia–thalamic closed-loop model that has long been hypothesized[1,2,18].

Although the pallidum and nigra have not previously been concep-tualized in a 'domain' structure of convergent striatal afferents per se, previous experimental findings have shown the same kinds of highly dense, topographically restricted striatal terminal fields in GPe and SNr[8,10,12,37,38] and topographical congruence between inputs and output zones in the striatonigral[37] and striatopallidal[39] pathways. Moreover, recordings of SNr neuronal activity have found that all neurons sam-pled within opsin-labelled direct- or indirect-pathway terminal fields responded to stimulation[40] (C. J. Wilson, personal communication), indicating that our domains and the convergent striatal projections that they represent are a good proxy for synaptic convergence of striatal information.

We demonstrate that the striatonigral pathway has greater conver-gence than the striatopallidal pathway (Fig. 4b). Anatomical studies in rats and monkeys lend support to this finding[8,10,38], and a similar pattern was found in ventral striatum[41]. Electrophysiological studies also demonstrated a lower degree of informational convergence in GPe[42–44] and a higher degree in SNr[34,45,46]. The greater specificity of the indirect pathway is likely to have a functional significance—for example, in Parkinson's disease. In monkeys rendered Parkinsonian by dopamine depletion, GPe neurons become responsive to a wider range of striatal inputs;[43,47] in humans with Parkinson's disease, activity in GPe neurons is reduced, as greater activity is driven through the indirect pathway[48]. A number of neuropsychiatric and movement disorders[17–23] probably involve alterations in specific subnetworks of the cortico–basal ganglia–thalamic network that perform specific cognitive and behavioural functions, and the combination of malfunctions in these circuits underpin complex disorders.

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

## Methods

### Subjects and surgeries

Subjects (in toto 268 male 2-month-old wild-type C57Bl6 and Ai14 mice, Jackson Laboratories) were anaesthetized with 2% isoflurane in oxygen. Buprenorphine SR (1 mg kg⁻¹) was administered at the beginning of the surgery as an analgesic. Glass micropipettes (10–30 μm diameter tip) filled with tracer were lowered into the target region and delivered an injection either by iontophoresis (1–10 min. infusion time, 5 μA, 7 s current pulses) or by pressure injection (20–80 nl volume). The tracers used were: PHAL (2.5%; Vector Laboratories, L-1110); AAV-GFP (AAV1-hSyn-EGFP-WPRE-bGH, Addgene); AAV-RFP (AAV1-CAG-tdTomato-WPRE-SV40, Addgene); red and green glycoprotein-deleted rabies (Gdel-RV-4tdTomato and Gdel-RV-4eGFP, I.W. laboratory) which are incapable of trans-synaptic spread; AAV1-hSyn-mRuby2-Syp-eGFP (B.K.L. laboratory); AAV1-Cre (AAV1-hSyn-Cre-WPRE, Addgene 105553); AAVretro-Cre (AAVretro-EF1a-Cre, Salk Institute); Cre-dependent AAV-FLEX-GFP (AAV1-CAG-Flex-eGFP-WPRE-bGH, Addgene); Cre-dependent AAV-FLEX-RFP (AAV1-CAG-Flex-tdTomato-WPRE-bGH, Addgene 100048); AAV-ChR2 (AAV1-hSyn-ChR2-EYFP-WPRE, Addgene 26973); Cre-dependent channelrhodopsin, AAV-DIO-ChR2 (AAV1-EF1a-DIO-ChR2-EYFP-WPRE, Addgene 20298); Fluorogold (FG, 1%, Fluorochrome); rhodamine-conjugated retrobeads (Lumifluor); and cholera toxin subunit B–Alexa Fluor 647 conjugate (CTB, 0.1–0.2%; Invitrogen). Most animals received multiple tracer injection combinations with non-overlapping fluorescence profiles, creating a pool of ~700 injections. A total of 138 mice were injected specifically for the striatal-output-pathway analysis, and the remainder were used for the other experiments in this manuscript. Animals were monitored daily after surgery until their body weight was on an increasing trajectory. All methods were approved by the Institutional Animal Care and Use Committees of the University of California, Los Angeles, the University of Southern California, the University of California, San Diego, and the Huazhong University of Science and Technology.

### Roster of injections for striatal output analyses

Altogether, 36 anterograde tracer injections were selected and analysed as representative injections from a data pool of 138 mice with iontophoretically delivered triple anterograde injections (PHAL, AAV-GFP and AAV-RFP) or double co-injections[25,49] (AAV-GFP and CTB; AAV-RFP and FG), collectively constituting 448 injections. All domains were injected more than once across this data pool, and injections within the same domain produced nearly identical labelling patterns. A representative injection for each domain of the caudoputamen was selected from this data pool based on precision of the injection within the targeted domain, quality of the axonal labelling and histological quality. Three injections were chosen for the nucleus accumbens (ACB), one in the core (ACBc) and two in the shell, medial (ACBsh.m) and lateral (ACBsh.l). The core, medial and lateral shell have divergent connectivity patterns, and although they mainly connect with a ventral basal ganglia network (ventral pallidum, substantia innominata and ventral tegmental area), they also send limited projections to restricted regions of the dorsal basal ganglia network[50]. We selected injections for 30 domains of the CP, the 29 domains from the rostral (CPr), intermediate (CPi), caudal (CPc) and caudal extreme (CPext) described in ref. [5], plus a new subdivision in the CPext. The CPext previously had a dorsal (CPext.d) and a ventral domain, but based on differing input and output patterns, the ventral domain here was split into the rostral ventral (CPext.rv) and caudal ventral (CPext.cv) domains. Additionally, three CP injections were chosen from a level intermediate to CPr and CPi, with projections to regions of the GPe not targeted by any of the other injections included in this experiment. Based on their relative position in the lateral CP, these injection sites appeared to be rostral associations of the somatomotor domains CPi.dl.d (tr), CPi.vl.imv (ul), and CPi.vl.v (m/i);

furthermore, their striatonigral projections were highly similar to those three domains. Because their striatopallidal projections terminated in regions in the GPe that were targeted by no other CP domains, they were included in the analysis of the GPe data. However, since their projections to the SNr and GPi were homologous to the CPi-level injections' projections patterns, they were excluded from the analyses of SNr data. Collectively, 29 mice yielded the 36 representative injections (some mice yielded more than one selected injection). The other major components of the cortico–basal ganglia–thalamic network, that is, the substantia nigra compact part, internal globus pallidus, subthalamic nucleus and various other thalamic nuclei, were not analysed in the present work.

### Histology and imaging

Animal subjects were deeply anaesthetized with an overdose bolus of sodium pentobarbital (Euthasol, 2 mg kg⁻¹, intraperitoneal injection), and cardiac perfused with normal saline followed by 4% boric acid-buffered paraformaldehyde. Brains were post-fixed overnight, embedded in 4% agarose, and sectioned on a vibratome at 50 μm thickness (50–150 μm for rabies-labelled tissues), collected in a 1-in-4 manner into 4 equivalent series, and stored in cryoprotectant at −20 °C until staining. Tissue series were stained with rabbit polyclonal anti-PHAL antibody (Vector Labs AS-2300) at 1:5,000 and donkey anti-rabbit AlexaFluor 647 (Jackson ImmunoResearch, 711-605-152). Nissl substance was stained with NeuroTrace 435/455 (ThermoFisher, N21479) at 1:500 to reveal cytoarchitecture. Sections were scanned on an Olympus VS120 epifluorescence microscope running VS-Desktop software with a 10× lens (Plan Apochromat) to capture the Nissl, FG, GFP, RFP and far red tracers in multichannel photomicrographs; these images were processed for the striatofugal network analysis. High-resolution images of some tissue samples (including the rabies-labelled tissue from Figs. 1e, 3b) were captured with an Andor Dragonfly spinning disk confocal microscope running Fusion software with a 60× lens with a z step of 1 μm.

### Image processing

Captured epifluorescence images were exported as large (14k × 11k pixel) multichannel tiff files (Extended Data Fig. 1e), and subsequently imported into our Connection Lens image processing software. After an initial atlas matching step, where each section was manually matched to its corresponding level of the ARA[51], images containing the pallidum and nigra were registered to the mouse brain atlas (Extended Data Fig. 1f). This work exclusively references levels of the ARA (for example, ARA 81 refers to level 81 of the ARA, Bregma = −2.78 mm). Our 1-in-4 series of 50-μm tissue sections gives us a view of the brain that is every other level in the ARA, itself based on a 1-in-2 series of 100-μm sections. Therefore, we registered our tissue sections onto every other atlas level of the pallidum and nigra, and for this purpose we chose the even levels of the pallidum (that is, ARA 58–68, even levels) and odd levels of the nigra (ARA 81–91, odd levels). All tissue sections were registered to their closest ARA level (that is, a section containing GPe at ARA 61 was mapped onto either ARA 60 or ARA 62). The determination of which level a given section was assigned to was made by an experienced neuroanatomist. The process of registering to a standardized set of atlas levels provided a uniform dataset that was amenable to computational analysis. After registration, Connection Lens guided users through an interactive segmentation step, creating a binary output image of axons (black) and background (white). Since the images were previously registered, the resulting segmentations could be accurately projected onto the atlas frame (Extended Data Fig. 1g). Finally, Connection Lens applied an overlap algorithm to quantify the segmented axonal labelling by region (GPe and SNr). Each level of the GPe and SNr in the atlas depicts a single unitary region, yet we knew from the labelling patterns that the striatofugal axonal termination patterns project to a sub-region of each nucleus. We subsequently applied the grid quantification method used previously in our corticostriatal analysis[5], subdividing each nucleus at each atlas level into a square grid space (105 × 105 pixels per box,

equivalent to 63 μm²), and quantified the axons per grid box (Extended Data Fig. 1h). Any injection that contributed less than 1,500 pixels to a given level was excluded from the community analysis for that level. A small number of cases had labelling that slightly exceeded the 1,500-px threshold for a given level but the labelling was judged too diffuse to be a meaningful terminal field, and were similarly excluded. The surviving grid box data were subjected to network analysis to determine striatonigral and striatopallidal community structure (Extended Data Fig. 1i; see next section). The derived communities were visualized by recolouring the grid boxes according to community identity (Extended Data Fig. 1j). And finally, projection maps of striatofugal axon terminals were created by aggregating the registered segmented axonal images onto maps of SNr and GPe (Extended Data Fig. 1k).

## Network analysis

The network structure of the dataset was assessed with the Louvain community detection algorithm[52], obtained from the Brain Connectivity Toolbox (https://sites.google.com/site/bctnet), and executed in Python. Louvain is a greedy, non-deterministic algorithm, with multiple runs producing differing returns of maximized modularity. Importantly, a gamma variable modulates the number of communities detected in a dataset, with smaller gamma values leading to low-dimension network structures (fewer nodes, larger communities) and larger gamma values leading to high-dimension network structures (more nodes, smaller communities). While the default gamma value of 1 is used commonly (and useful for communicating a frame of reference, as it is a de facto standard), choosing the optimal gamma value is a non-trivial problem.

In an attempt to obtain the most descriptive network partition among this parameterization and variability, we performed a survey of community structures across different gamma values. The Louvain algorithm was run 100 times per each gamma value, over a gamma range of 0 to 2 at 0.05 increments for every nucleus level. A consensus community structure (conceptually, the 'average' community structure[53]) was calculated from each batch of 100 runs at every gamma. The resultant 41 consensus community structures were compiled into a frequency histogram, to determine the most common community structures to arise over the range of gammas for a nucleus level. The true network structure of the underlying data should act as a 'magnet' or attractor for a stable community structure over multiple gammas; therefore an accurately characterized community structure should be obtained across multiple gamma values[54], represented by peaks in the graph (Extended Data Fig. 12l–m).

We applied different analytical parameters to the direct (striatonigral) and indirect (striatopallidal) pathway data. The domains in the SNr and GPe exist in three dimensions, and for the SNr in particular likely extend across multiple levels of the nucleus. Our goal was to balance across-level similarity with high dimension domain structure, as the input data (that is, the striatofugal terminal fields) is verified higher dimensional. Moreover, we sought to parse the pallidum and nigra into more than the three classically recognized output channels.

For the direct pathway, most of the striatal domains send projections in longitudinal columns along the entire rostrocaudal extent of the SNr. This suggests that there is a relatively high degree of consistency in the community structures across adjacent levels of the SNr. However, caudally the SNr becomes physically smaller in cross-sectional area and the axonal terminal fields exhibit a higher degree of convergence than in rostral levels (Extended Data Figs. 4a, 12b, c; also see ref.[10]). We quantitatively characterized this convergence to describe the degree of integration, and used these data to inform our selection of gamma values. Using the quantified grid box data, we created frequency distributions of the boxes categorized by how many different striatal inputs they received (only boxes receiving input were included in this analysis) (Extended Data Fig. 12a). We applied a minimum threshold such that inputs contributing less than 5% overlap to a box (551 pixels in a 105 pixel² box) were excluded from that boxes' tally of inputs. When graphed together, the histograms show that the rostral SNr levels 81–85

have negatively skewed distributions, while the caudal levels 87–91 have more platykurtic distributions with relatively fatter tails in the 10–15 inputs per box range (Extended Data Fig. 12b), a trait that becomes more apparent when the rostral levels and caudal levels are averaged (Extended Data Fig. 12c). There is no significant correlation of mode (peak value) of each histogram with rostrocaudal level as assessed by linear regression ($r = 0.4252$, $P = 0.4006$; Extended Data Fig. 12e), and there is no significant difference in mode between the rostral (mean ± s.e.m.: $7.0 ± 0.577$) and caudal ($7.3 ± 1.856$) groups ($P = 0.8796$, $t = 0.1715$, d.f. = 4; two-tailed Welch's-corrected $t$-test), verifying that the graphs have similar central tendencies (Extended Data Fig. 12c). Two-way ANOVA of the rostral and caudal groups (SNr level × amount of integration) finds a significant main effect of integration ($P < 0.0001$, d.f. = 14, $F = 12.46$) and a significant interaction of integration and SNr level ($P = 0.0219$, d.f. = 14, $F = 2.136$). Post hoc Bonferroni tests reveal the caudal group has a significantly greater proportion of boxes receiving 11 inputs ($P < 0.0033$, $t = 3.385$) and a nearly significant difference for 10-input boxes ($0.05 > P > 0.025$, familywise-adjusted $α = 0.0033$, $t = 2.278$). Thus, the caudal three levels of the SNr have a significantly greater proportion of their area devoted to high integration (ten or more inputs per box) (Extended Data Fig. 12d–f), indicating that the rostral and caudal SNr should be analysed with different parameters since there is probably a different, more integrative domain structure caudally.

Since the integration analysis indicates that the rostral and caudal halves of the SNr form two groups, we selected community structures that were most common through the rostral and caudal groups (Extended Data Fig. 12l). By stacking the histograms of the component levels, the peaks in the graphs reveal the most stable community structures common to all constituent levels. For the rostral SNr group, there were two clear peaks, for a six-domain and a ten-domain structure (Extended Data Fig. 12l, SNr rostral group). We have selected to present the ten-domain structure, because we sought to subdivide the SNr into as fine a coherent partition as possible, although the 6-domain structure may also be a valid way to interpret the striatonigral data as well, since it is possible there is a nested multi-scale network architecture to the striatonigral pathway as with the corticostriatal pathway. For the caudal group, there was a clear peak for the seven-domain structure (Extended Data Fig. 12l, SNr caudal group). All gamma values returning the chosen domain structure for each nucleus-level were pooled and the consensus community structure of that pool was the final network output.

Most community detection algorithms, including Louvain, impose a unitary structure on an information network such that each node belongs to one and only one community. This simplified network structure is easier to interpret, but may not be reflective of the complex nature of real-world networks, wherein a single node may participate in multiple subnetwork structures, as is probably true in brain networks. Thus, after determining the community structure for each representative level of the SNr, we manually joined together communities on adjacent levels based on continuity and similarity of inputs (Supplementary Table 1). For example, the striatal inputs to the medial region of the SNr are similar enough when comparing adjacent levels (mean Jaccard index comparing all adjacent levels = 0.61) that we determine them to form one continuous domain, the medial domain (Supplementary Table 1). The average of mean Jaccard indices for all nigral domains that span more than one level ($n = 12$) is $0.61 ± 0.28$ (mean ± s.d.). Furthermore, all cross-level domains also have one or more inputs that consistently span the entire domain. It should be pointed out that the nigral terminations of a particular CP domain frequently shift in their mediolateral or dorsoventral position in the nigra at different rostrocaudal levels; consequently, even after our efforts to join together similar communities into unified nigral domains, many CP domains contribute projections to multiple SNr domains, such as the CPr.imd providing input to the SNr.m at levels 83, 85, 87 and 91 and to the SNr.v at level 89.

Each indirect pathway input tends to densely innervate just a few levels of the GPe. GPe boxes integrate at most 9 inputs, much more

restricted than the SNr (up to 15), and mean mode of the GPe ($3.33 \pm 0.558$ (mean $\pm$ s.e.m.); $n = 6$) is significantly smaller than mean mode of SNr ($7.17 \pm 0.872$; $n = 6$), indicating less integration and more segregated relaying of striatal activity through the indirect pathway ($P = 0.0060$, $t = 3.702$, d.f. = 8; two-tailed Welch's $t$-test). The frequency distributions of inputs per box show similarly shaped histograms across GPe levels, with decreasing mode value from rostral to caudal (Extended Data Fig. 12g, h). Linear regression of the histogram modes shows a significant correlation with GPE level ($r = 0.8607$, $P = 0.0278$), with mode decreasing towards caudal levels (Extended Data Fig. 12j), indicating that caudal GPe integrates progressively fewer inputs per box (Extended Data Fig. 12i–k). Given that the graphs vary continuously along the rostrocaudal axis, and the lower degree of continuity of striatopallidal projections along that axis, we evaluated each level of the GPe independently (Extended Data Fig. 12m). A far smaller proportion of pallidal domains exhibit cross-level structure than in the nigra (36% versus 86%). The average of mean Jaccard indices for the pallidal domains that span more than one level ($n = 10$) is $0.64 \pm 0.21$ (mean $\pm$ s.d.). No striatal domain innervates the entire rostrocaudal axis of the GPe (Supplementary Table 2).

### Whole-brain 3D imaging and reconstruction of neuronal morphology

Intact brain ($n = 1$) was SHIELD-cleared as described[55], placed in refractive index-matching solution (EasyIndex, LifeCanvas), and imaged on a LifeCanvas light-sheet microscope running SmartSPIM Acquisition software at 4× and 10× magnification. These images as well as $z$-stack images captured with the DragonFly confocal microscope were viewed within Aivia reconstruction software (v8.8.2. DRVision) and neurons were manually reconstructed. Geometric processing of the reconstructions was performed using the Quantitative Imaging Toolkit (http://cabeen.io/qitwiki), and morphometric data were obtained from the reconstructions with NeuTube (v1.0z). Descriptive statistics of the morphological features of these neurons were generated by NeuTube.

### Anterograde transsynaptic tracing

This technique leverages the fact that when AAV1 is injected at sufficiently high concentration into a neuronal population, viral particles will travel down the axons and be released from the synaptic terminals where they can infect postsynaptic neurons. Detailed methodology is as described[56,57]. In brief, anaesthetized mice were iontophoretically injected with Cre-dependent AAV-FLEX-RFP in the target nucleus (except the demonstration of the ACBc-SNr.dm-PF.vs pathway, which used pressure injection of 50 nl into medial SNr at atlas levels 81–85), and then pressure injected (20–80 nl) with AAV-Cre in an upstream nucleus. The AAV-Cre is transported anterogradely down the axons and is released from the terminals, where it transfects postsynaptic cells that have been infected with high concentrations of Cre-dependent AAV-FLEX-RFP. The scant Cre expression is sufficient to unlock strong fluorophore expression in the downstream neurons. After a three-week post-operative recovery, animals were pentobarbital-anaesthetized and perfused as above. The Cre injection site was verified by staining with mouse anti-Cre recombinase monoclonal antibody (Millipore Sigma, MAB3120) and donkey anti-mouse AlexaFluor 647 (Jackson ImmunoResearch, 715-605-150). Additionally, two Ai14 mice (Jackson Laboratories 007908) with endogenous tdTomato expression following Cre recombination were injected with AAV-Cre in mouth primary motor cortex MOp-m/i to demonstrate the corticonigral pathway (Extended Data Fig. 20a).

### Experimental designs for CRACM with anterograde transsynaptic tracing

To examine convergence and parallelism in the striatonigral and striatopallidal pathways, we used CRACM[58]. We injected AAV1-Cre (AAV1-hSyn-Cre-WPRE; 20–80 nl, pressure) into the striatal domain CPc.d.dl, which is taken up at the injection site, transported down the axon, crosses the synapse with low efficiency

and infects postsynaptic neurons in GPe.24, 26 and 32 and SNr.v. To label the Cre-expressing postsynaptic neurons with RFP, we injected those pallidal and nigral target domains with AAV-FLEX-RFP (AAV1-CAG-Flex-tdTomato-WPRE; 2–3 min, iontophoresis). We next injected AAV-ChR2 (AAV1-hSyn-ChR2-EYFP-WPRE; 2–3 min, iontophoresis) into striatal somatomotor trunk domain CPi.dl.d (tr), labelling its axons in GPe.6 and 25 and SNr.v with YFP and rendering them optically excitable. Thus in the pallidum the RFP neurons should not respond to stimulation of the ChR2 axons since they do not overlap, while in the nigra the RFP neurons should respond to stimulation of the ChR2 axons since they colocalize (Fig. 2c; $n = 3$).

To examine whether the direct (striatonigral) and indirect (striato–pallido–nigral) pathways re-converge onto individual SNr neurons, we injected AAV-Cre into the striatal domain CPi.dl.d (tr), which trans-synaptically infected postsynaptic neurons of the indirect pathway in GPe.6 and 25 and of the direct pathway in SNr.v, causing those cells to express Cre ($n = 7$). In the same surgery, AAV-DIO-ChR2 (2–3 min, iontophoresis) was injected into the region of GPe.6 and 25 and AAV-FLEX-RFP was injected into SNr.v, meaning only the Cre-expressing neurons postsynaptic to terminals originating from CPi.dl.d (tr) expressed ChR2 (YFP) and RFP, respectively. Three weeks later, acute slices of nigra were prepared, and RFP-expressing neurons of the SNr (postsynaptic to the direct pathway) were patched and recorded during channelrhodopsin stimulation (the indirect pathway). If there is re-convergence of homotypic direct and indirect pathways, then the RFP-expressing neurons in SNr should respond to optical stimulation of the pallidonigral pathway (Extended Data Fig. 8h).

Two experiments were conducted to examine whether the thalamo-cortical axons of a cortico–basal ganglia–thalamic subnetwork synapse upon the corticostriatal neurons that feed into that subnetwork. These experiments targeted the oro-brachial subnetwork ($n = 2$) and the associative subnetwork ($n = 4$). For the oro-brachial subnetwork experiment, AAV-ChR2 was iontophoretically injected into the PF.m thalamic domain and retrobeads (30 nl, pressure) were injected into the CPi.vm.v (m/i) domain. At least three weeks later, slices containing the mouth primary cortical region MOp-m/i were collected for electrophysiological recording as described below (Fig. 3c). For the associative subnetwork experiment, AAV-ChR2 was iontophoretically injected into the PF.a thalamic domain and retrobeads were injected into the CPi.dm.d domain. Slices containing the anterior cingulate area (ACA) were collected for electrophysiological recording (Extended Data Fig. 17c).

### Electrophysiological recording

At least two weeks following channelrhodopsin and tracer injections, acute brain slices were prepared for recording. Following anaesthesia, the animal was decapitated and the brain was quickly removed and immersed in ice-cold dissection buffer (cortical recording experiments (in mM): 60 NaCl, 3 KCl, 1.25 $NaH_2PO_4$, 25 $NaHCO_3$, 115 sucrose, 10 glucose, 7 $MgCl_2$, 0.5 $CaCl_2$; saturated with 95% $O_2$ and 5% $CO_2$; pH = 7.4; for basal ganglia recording experiments: 208 sucrose, 2.5 KCl, 1.25 $NaH_2PO_4$, 26 $NaHCO_3$, 1.3 $MgCl_2$, 8 $MgSO_4$ and 10 glucose). Brain slices of 200–300 μm thickness were cut in coronal plane using a vibrating microtome (Leica VT1000S). Slices were allowed to recover for 30 min in a submersion chamber filled with warmed (35 °C) ACSF containing (in mM) 130 NaCl, 3 KCl, 1.25 $NaH_2PO_4$, 26 $NaHCO_3$, 2 $CaCl_2$, 2 $MgCl_2$ and 10 glucose, oxygenated with 95% $O_2$ and 5% $CO_2$, pH 7.2–7.4, 290–310 mOsm, and then cooled gradually to room temperature until recording. The presence of retrobead or RFP and ChR2 (YFP) labelling was examined under green and blue fluorescence excitation in the slices before recording. Patch pipettes (Kimax) with -6–7 MΩ impedance were used for whole-cell recordings. Recording pipettes contained: 130 mM potassium-gluconate, 4 mM KCl, 2 mM NaCl, 10 mM HEPES, 0.2 mM EGTA, 4 mM ATP, 0.3 mM GTP, 14 mM phosphocreatine (pH 7.25; 290 mOsm) and biocytin (0.2%). For the thalamo–cortico–striatal experiments (Fig. 3c, Extended Data Fig. 17c) 0.1–1 μM tetrodotoxin

and 0.1–1 mM 4-aminopyridine was added to the external solution for isolation and recording of monosynaptic responses to blue light stimulation. For the oro-brachial subnetwork experiment, only layer 5 neurons were recorded. Signals were recorded from red-labelled neurons with a MultiClamp 700B amplifier (Molecular Devices) running pClamp software under voltage clamp mode at a holding voltage of −70 mV for excitatory currents and 0 or +10 mV for inhibitory currents, filtered at 2 kHz and sampled at 10 kHz. Blue light (470 nm) stimulus was delivered in a 0.5–5 ms pulse, at 1–3 mW power, for 3–5 trials, delivered via a mercury arc lamp gated with an electronic shutter. Signals were analysed using Clampfit. For responding neurons, peak responses were averaged across trials, and for non-responding neurons, peak recorded amplitudes within the 1-s window following stimulation were averaged across trials.

## Fluorescence micro-optical sectioning tomography

All fMOST experiments were conducted in accordance with the Institutional Animal Ethics Committee of Huazhong University of Science and Technology. For sparse labelling of striatal neurons, we created a single pAAV co-package of DNA cassettes of CMV-Cre and Cre-dependent EF1a-DIO-GFP at a ratio of 1:1,000,000, respectively, so that the final viral admixture contained one virus with both cassettes for every 1,000,000 viruses that contained only the EF1a.DIO.GFP cassette (total viral concentration $8 \times 10^{12}$ genome copies per ml, from BrainVTA). This admixture was pressure injected (100 nl) into CPi.dm (M–L, A–P, D–V: 0.14, −1.3, −2.6), allowing high viral load for the fluorophore gene with low frequency of co-expression of Cre, resulting in sparse yet bright GFP expression. A detailed protocol has been previously described[59]. After 5 weeks, mice ($n = 7$) were anaesthetized, perfused with 0.01 M PBS and 4% paraformaldehyde, and brains were post-fixed overnight. For whole-brain imaging, brains were rinsed in 0.01M PBS solution and dehydrated in a graded ethanol series (50, 70 and 95% ethanol), submerged in gradient series of Lowicryl HM20 resin, and polymerized at a gradient temperature in a vacuum oven.

The resin-embedded whole-brain samples were imaged using fMOST, a three-dimensional dual-wavelength microscope-microtome combination instrument (see ref. [60] for a detailed description). Block imaging mode was used to slice and scan layer by layer through the whole sample in the coronal plane. GFP-labelled neurons and propidium iodide-stained cytoarchitecture were acquired at a voxel resolution of $0.32 \times 0.32 \times 1 \mu m^3$. The raw images were first preprocessed for intensity correction, and then the image sequence was converted to TDat, an efficient 3D file format for large volume images, to facilitate the computing of terabyte- and petabyte-scale brain-wide datasets[61]. We employed GTree for semi-automated, manually assisted reconstruction of neuronal morphology in 3D[62]. Subsequently, neuronal morphological data were mapped to the Allen CCFv3 using BrainsMapi, a robust image registration interface for large volume brain images[63]. Because the contours of brain regions on the propidium iodide-stained images can be more clearly identified, this greatly reduces the difficulty of accurate registration.

## D1 and A2A cell-type specific tracing

For labelling D1 and D2 dopamine receptor-expressing medium spiny neurons (MSNs), adult Adora2a-Cre (GENSAT 036158-UCD, for labelling D2-MSNs) and Drd1a-Cre (GENSAT 017264-UCD, for D1-MSNs) congenic mice on a C57BL6/J background were obtained from GENSAT and backcrossed with wild type C57BL mice (Jackson Laboratories) for several generations. Surgeries were performed between 10–12 weeks of age. Subjects were anaesthetized with 2% isoflurane in oxygen. Buprenorphine SR (1 mg kg⁻¹) was administered at the beginning of the surgery as an analgesic. Glass micropipettes (10–30 μm diameter tip) filled with AAV-DIO-EGFP (AAVDJ-hSyn-DIO-EGFP-WPRE-bGH, B.K.L. laboratory) were lowered into the target region and delivered a localized injection by pressure (50–150 nl volume) at a rate of 100 nl min⁻¹. After 3 weeks, animals were deeply anaesthetized and perfused. Tissue sections were stained with DAPI and imaged on an Olympus VS120 epifluorescence microscope with a 10× objective lens. All procedures to maintain and use mice were approved by the Institutional Animal Care and Use Committee at the University of California, San Diego.

## Statistical analysis

All standard statistical analyses were performed with GraphPad Prism v4.0c for Macintosh. Sample sizes were not predetermined statistically, but are consistent with experiments of their type. Randomization was not necessary because no independent group comparisons were made.

## Reporting summary

Further information on research design is available in the Nature Research Reporting Summary linked to this paper.

## Data availability

An application presenting the projection maps of all axonal reconstructions, along with the quantified grid box data for the 36 representative striatal injection cases, the SNr neuronal reconstructions (from Fig. 1k), the main and extended data figures, and supplementary video are available at http://brain.neurobio.ucla.edu/publications/. Source data are provided with this paper.

## Code availability

An in-house software, Connection Lens, was used to register (warp), threshold (segment), and annotate the labelling in all image data used in the striatal output analysis (Extended Data Fig. 2). This software has not been publicly released yet. For the Louvain algorithm implementation, the Brain Connectivity Toolbox was employed, which is freely available at https://sites.google.com/site/bctnet/. Geometric processing of the SNr neuronal reconstructions was performed using the Quantitative Imaging Toolkit, available at http://cabeen.io/qitwiki.

**Acknowledgements** We thank J. Gonzalez, S. Kong, K. Kirio, F. S. Cano and Z. Hobel for technical support. Funding for this research is provided by: NIH Grants U01 MH114829 (H.-W.D.), R01 MH094360-06 (H.-W.D.), NIH U19MH114821 (Z. J. Huang), U19MH114831 (J. Ecker/E. Callaway); U01MH116990 (L.I.Z.); RF1MH114112 (L.I.Z. and H.-W.D.); R01DC008983 (L.I.Z.); U01MH117079 (X.W.Y. and H.-W.D.); Cell, Circuits and Systems Analysis Core is supported by USPHS grant U54HD087101; The National Science Fund for Creative Research Group of China (grants 61721092, 61890953, and 61890954 to Q.L.).

**Author contributions** H.-W.D. and N.N.F. conceived, designed and managed the project. N.N.F. and H.-W.D. wrote the manuscript. N.N.F. performed data analysis and prepared all figures for publication. J.B., C. Cepeda, B.P., N.N.F., M.S.L. and L.I.Z. conducted electrophysiology experiments, generated data, graphs and writing for those parts of the manuscript. L.K. conducted 3D reconstructions, generated morphometric data and images for figures, interpreted data and contributed to writing. L. Garcia conducted the Louvain analyses and consensus community analyses for the SNr and GPe data, generated all matrices for the figures (except the 3D matrix) and the domain output images which were the basis of the domain maps. L. Gao, M.B., Y.S., L. Guo and B. Zingg performed stereotaxic surgeries to produce and collect anatomical connectivity data. X.L., A.L. T.J., X.J. and Z.F. contributed fMOST 3D neuron labelling and reconstruction and method writing. H.G. and Q.L. managed the fMOST group. J.-H.C. and B.K.L. carried out the cell-type-specific D1/D2 experiment and contributed writing. N.K. and B. Zhang performed brain clearing and 3D lightsheet imaging. S. Azam, D.L., G.D., M.F., H.-S.M., T.B., S.U. and D.L.J. performed registration of imaging data. I.B. led the informatics team, wrote the code for computational network analysis and contributed writing. M.Z. developed the algorithm for 3D reconstruction of neurons. K.C. provided informatics support for the imaging process. S.Y. and A.T. developed informatics visualization tools for online presentation of data. J.S. created the 3D video. C. Cao and S. Aquino. participated in histological processing. A.S., N.L.B., M.S., H.X. and M.S.B. contributed to data interpretation. J.D.H., X.W.Y. and H.H. offered constructive guidance for manuscript edits. I.W. provided rabies viruses.

**Competing interests** The authors declare no competing interests.

**Additional information**
**Correspondence and requests for materials** should be addressed to Nicholas N. Foster or Hong-Wei Dong.

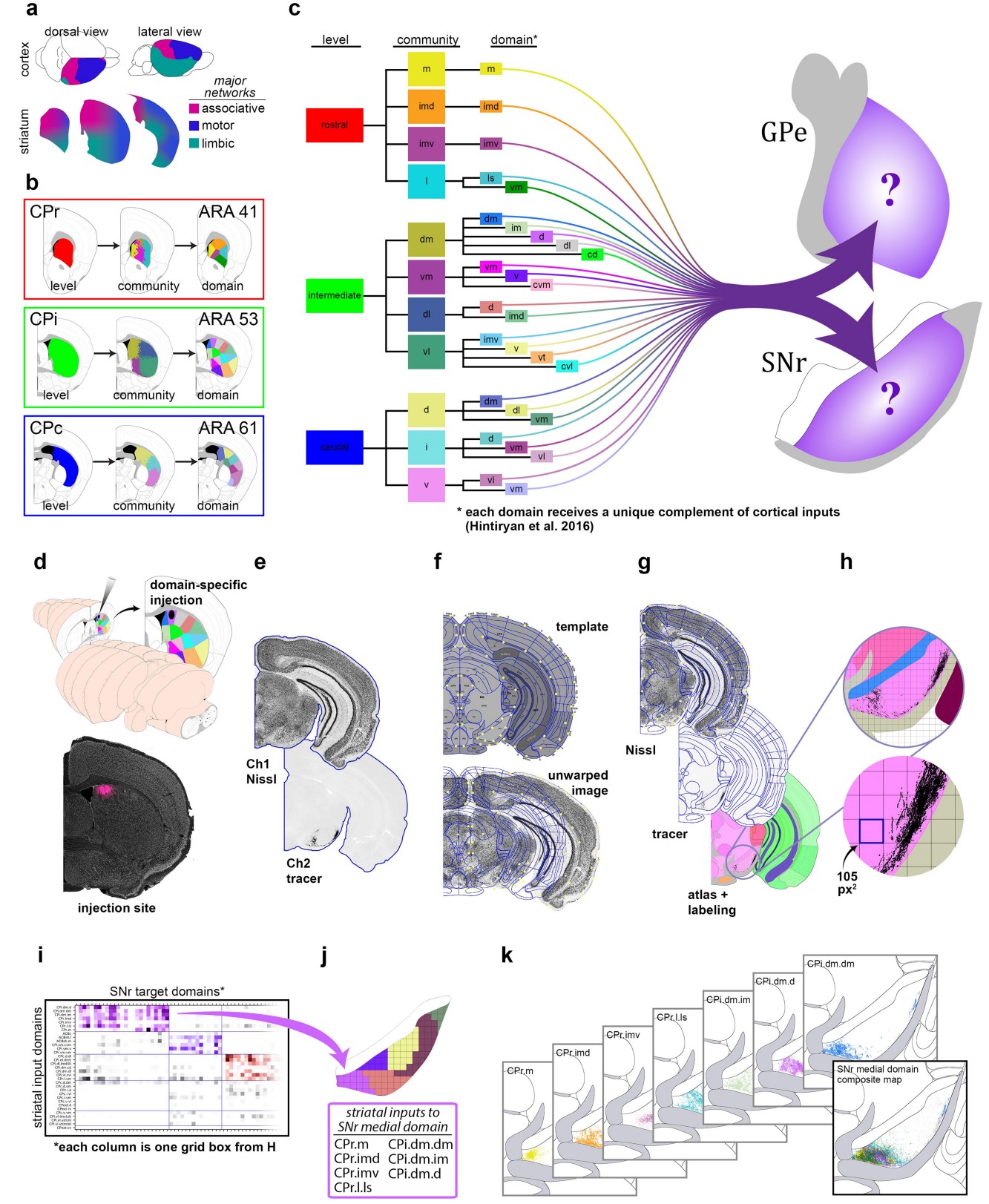

**Extended Data Fig. 1** | See next page for caption.

**Extended Data Fig. 1 | Rationale and workflow for the striatal output analysis. a**, General topography of the 3 classic corticostriatal pathways: motor, limbic, and associative. **b**, Map of the multi-scale subdivisions (level>community>domain) of the caudoputamen at rostral (CPr), intermediate (CPi), and caudal (CPc) levels. **c**, Dendrogram of the multi-scale, hierarchical structure of the CP, depicting how each level is composed of smaller communities and even smaller domains, each with a unique set of cortical inputs. The precise topography of the striatal output pathway is unknown. The CP caudal extreme and nucleus accumbens domains are not depicted here but were analyzed. The data production workflow starts with **d**, discrete injection of anterograde tracer into one of the striatal domains. Tissue sections are **e**, imaged and imported into **f**, Connection Lens where fiducial points in the Nissl channel are matched to the atlas template. **g**, Images are deformably warped and the tracer channel is segmented into a binary threshold image. **h**, The software subdivides all brain regions into a square grid space and quantifies the pixels of axon labeling in each grid box. **i**, The quantified axonal terminals from all injections to all grid boxes at each nucleus-level is visually summarized in a matrix (darker shading indicates denser termination; colors correspond to particular SNr domains). Statistical analysis reveals the subnetworks, groups of striatal domains that project to a common set of grid boxes. Those grid boxes in the SNr maps are then **j**, colored to visualize the new input-defined pallidal or nigral domain. **k**, Composite projection maps of the colored axonal terminals illustrate the striatofugal terminal pattern, with axonal color matching the source domain from the striatal domain map.

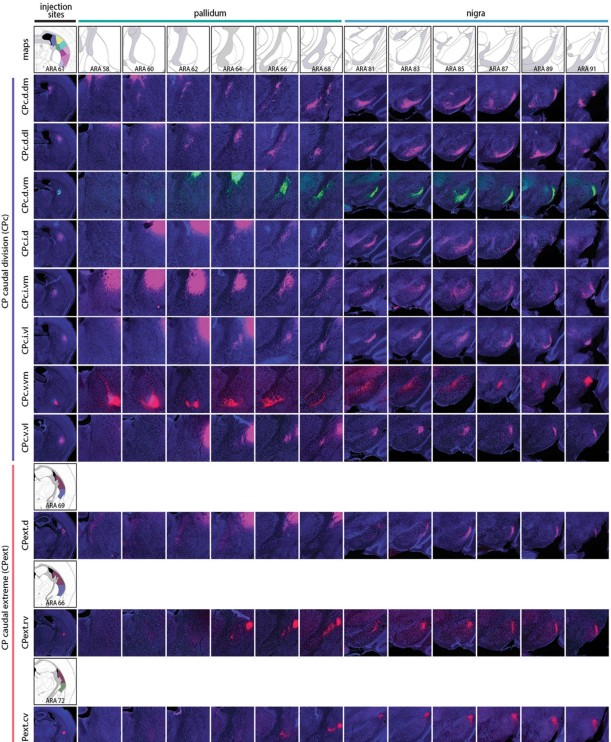

**a** Injections in CPr, ACB, and CPi rostral extension

**b** Injections in CPi

**c** Injections in CPc and CPext

**Extended Data Fig. 2 | All 36 striatal injections used in the striatal output analysis.** The injection sites and raw axonal labeling at each representative level of the pallidum and nigra are shown. **a**, injections in the rostral CP, nucleus accumbens, and intermediate CP rostral extension; **b**, injections in the intermediate CP; and **c**, injections in the caudal CP and caudal extreme. These images were analyzed to determine the 14 nigral and 36 pallidal domains. The entire striatal output dataset was subjected to computational analysis with the Louvain algorithm, revealing the domain structure at each representative level of the GPe and SNr (every other level of each structure). The CP intermediate rostral extension injections were included in the GPe analysis but not the SNr analysis, since those injections labeled unique areas in GPe but had labeling highly similar to their CPi-level counterparts (i.e., the CPi.dl.imd, CPi.vl.imv, and CPi.vl.v). All images are coronal, medial is left.

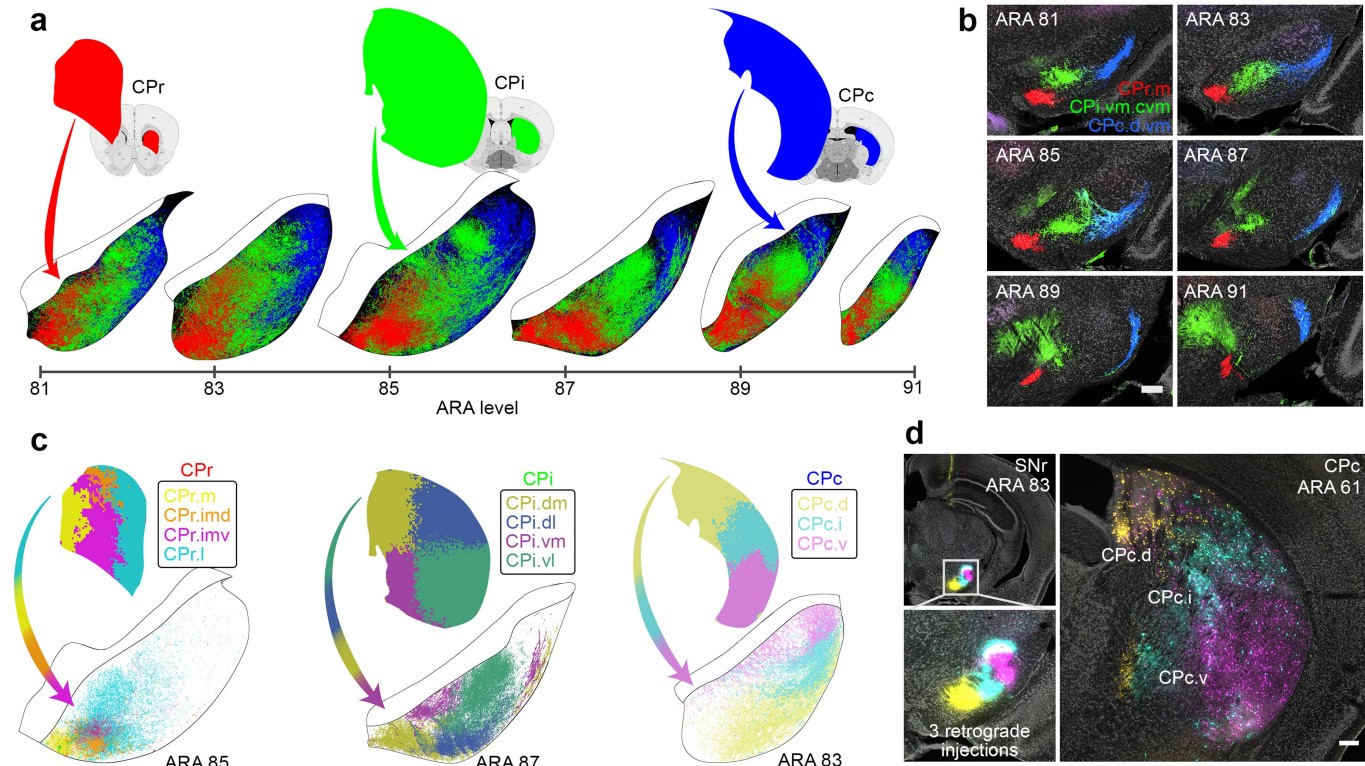

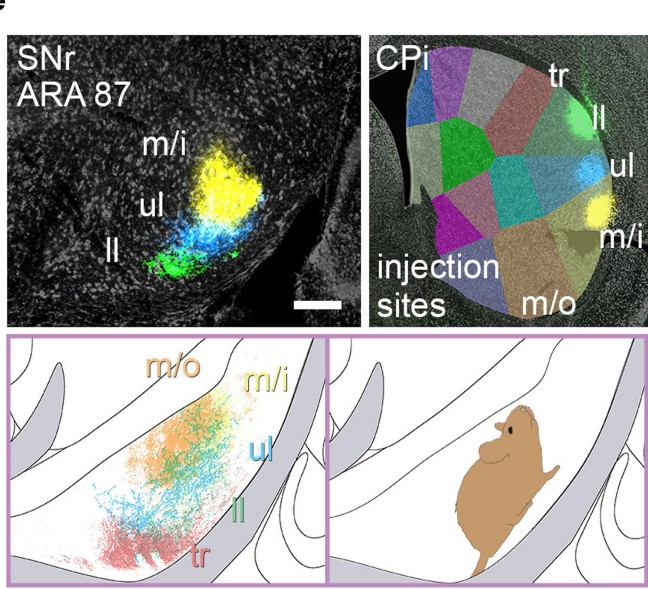

**Extended Data Fig. 3 | Striatonigral topographic trends by striatal level and community, and somatotopic map. a**, Composite maps depict axonal reconstructions originating from all striatal domains of each CP level (rostral, intermediate, and caudal) with axons colored red, green, or blue (respectively). The rostrocaudal axis of the CP terminates in a mediolateral pattern in the SNr, respectively. **b**, Triple anterograde injection experiment shows labeling from rostral (CPr.m), intermediate (CPi.vm.cvm), and caudal (CPc.d.vm) CP domains in a single brain. Tracers used were AAV-GVP (green), AAV-RFP (red), and PHAL (blue). **c**, Community-level view of striatal terminations reveals that the dorsoventral axis of the CP terminates in an inverted pattern in the SNr. **d**, Triple retrograde injection experiment in a single brain precisely back-labels

the 3 communities of the CPc. dG-rabies-GFP, dG-rabies-RFP, and CTB-647 were injected into SNr at ARA 83, and labeling was pseudocolored to match the three communities of the CPc. Note the resemblance of the labeling pattern in CPc with the community map of CPc shown in **c. e**, Triple anterograde experiment shows the terminal fields of the lower limb (ll), upper limb (ul), and inner mouth (m/i) striatal domains within the SNr of a single animal, as well as a composite reconstruction map of the full somatotopy (with trunk (tr) and outer mouth (m/o) regions as well). The mouse illustration depicts the approximate homunculus in the SNr. Photomicrographs are coronal, medial is left, and scalebars = 200μm.

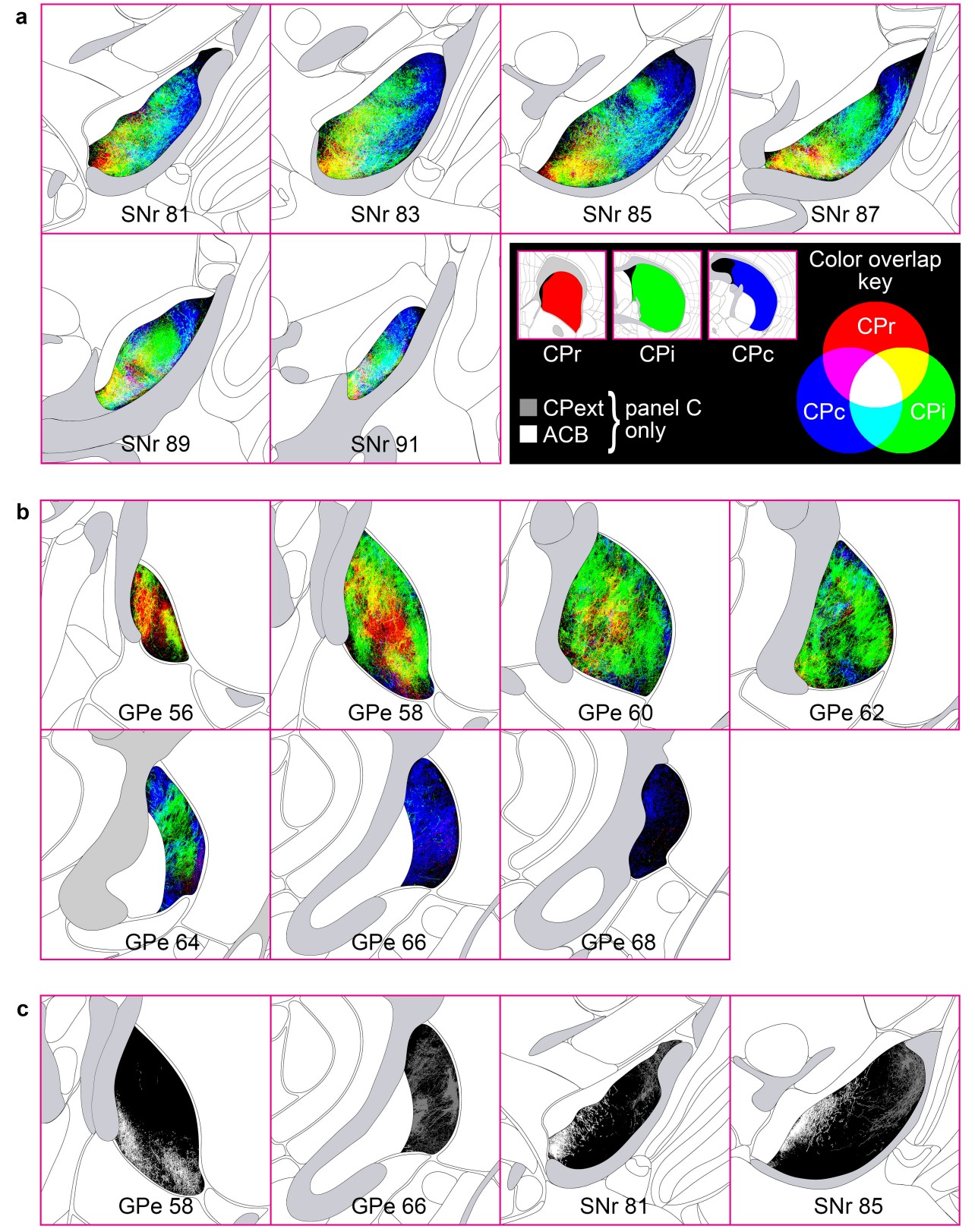

**Extended Data Fig. 4 | Striatal level topographic trends, blended-color style emphasizing convergence.** Similar to the striatal level maps presented in Extended Data Figs. 3a and 11c, the maps here present the striatal level termination patterns in red, green, and blue for CPr, CPi, and CPc, respectively, but when axons from different divisions overlap, the pixel color is a blended RGB value of the constituent axons (see Color overlap key). The maps in Extended Data Figs. 3a and 11c were made with opaque layers (CPr>CPi>CPc, top to bottom) such that pixels of axon in the upper layers occlude any other axons in lower layers in the same pixel; those maps were meant to emphasize the differences in level-scale terminal patterns, whereas the maps in this figure emphasize the areas of overlap within **a** SNr and **b** GPe. The terminations of the ACB and CPext (also known as the tail of the caudate) are shown in grayscale in **c**.

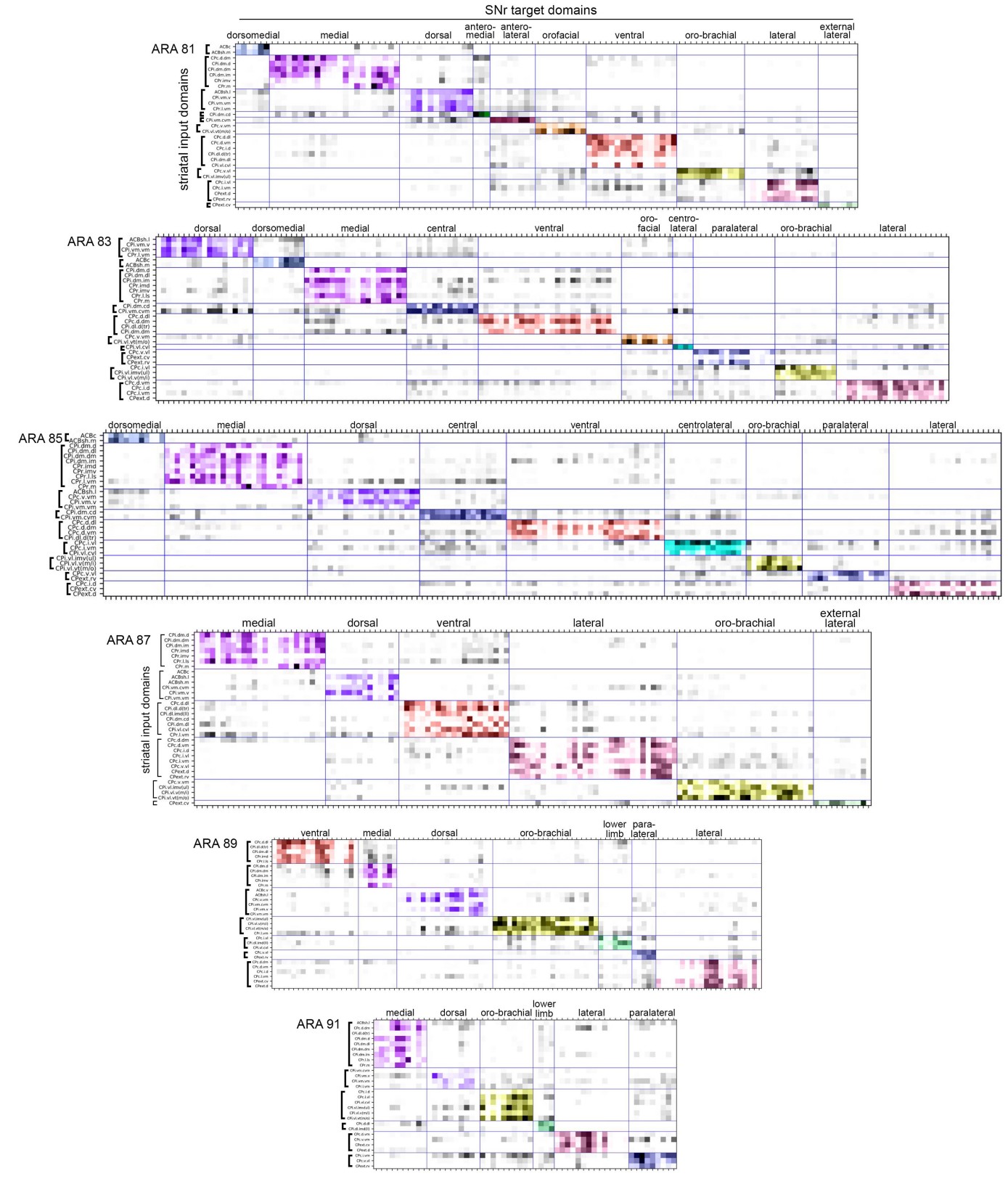

**Extended Data Fig. 5 | Striatonigral matrices.** The entire striatonigral dataset (33 injections, Extended Data Fig. 2) was quantified by box grid analysis, and the matrices are a visual representation of the percent coverage of a given grid box by axons from each striatal source domain (darker shading indicates greater coverage). These values were subjected to computational analysis with the Louvain community detection algorithm, which grouped together the striatal input domains (rows) that terminate in a common set of nigral grid boxes (columns) at each nucleus-level. The matrices presented here are ordered matrices, arranged to show the domains determined by the Louvain algorithm (for a full description of how the domains were determined, see Methods, *Network analysis*). The new nigral domains lie along the diagonal and are colored to match the SNr domain map (see Extended Data Fig. 1), and domain names are listed along the top of each matrix.

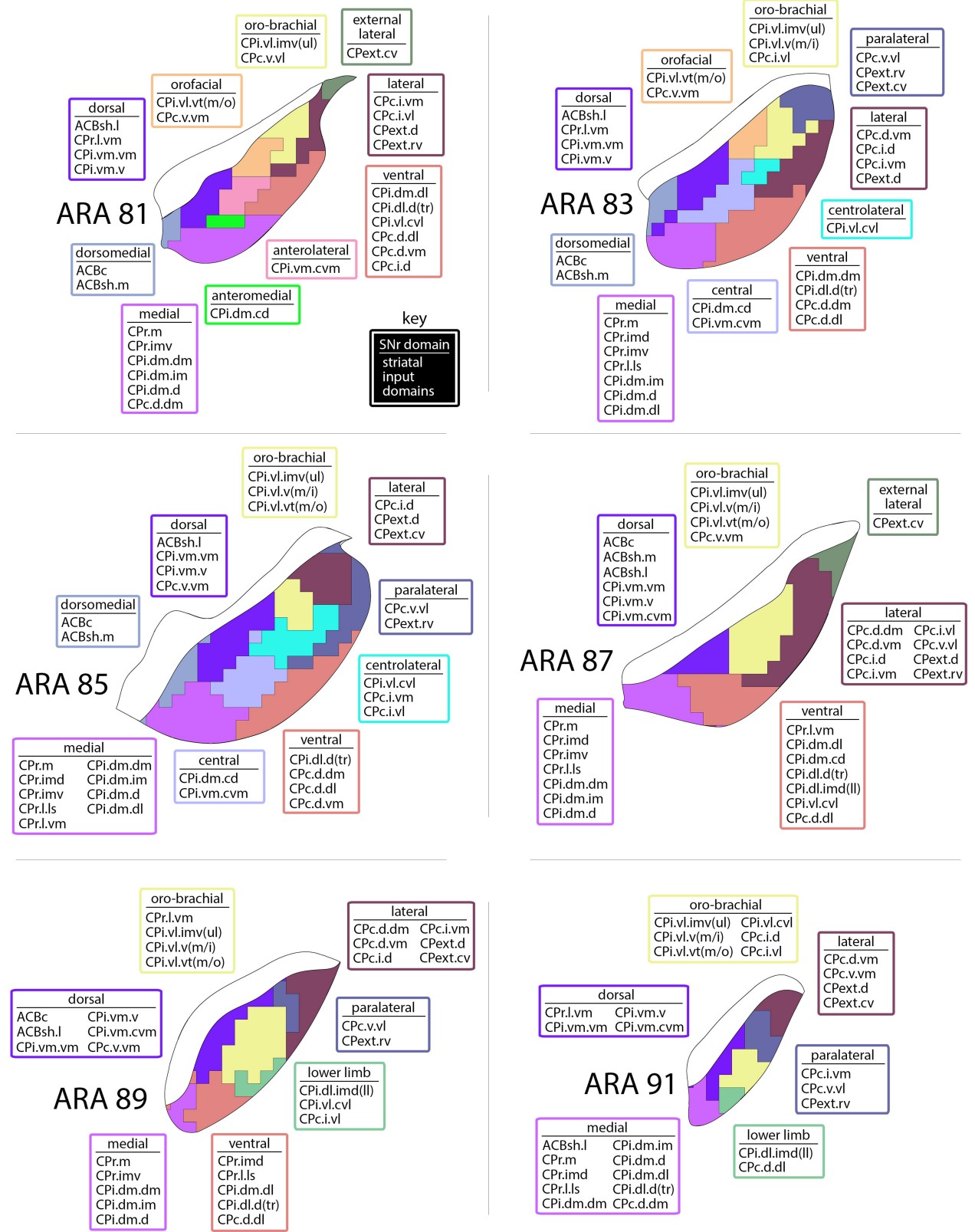

**Extended Data Fig. 6 | SNr domain input map.** At each level of the SNr, the striatal inputs to each nigral domain are listed in the boxes. Each box border is colored to match its domain in the map. The entire striatonigral dataset (Extended Data Fig. 2) was computationally analyzed, grouping striatal inputs that share common target zones in the nigra. Those zones were demarcated as the 14 SNr domains.

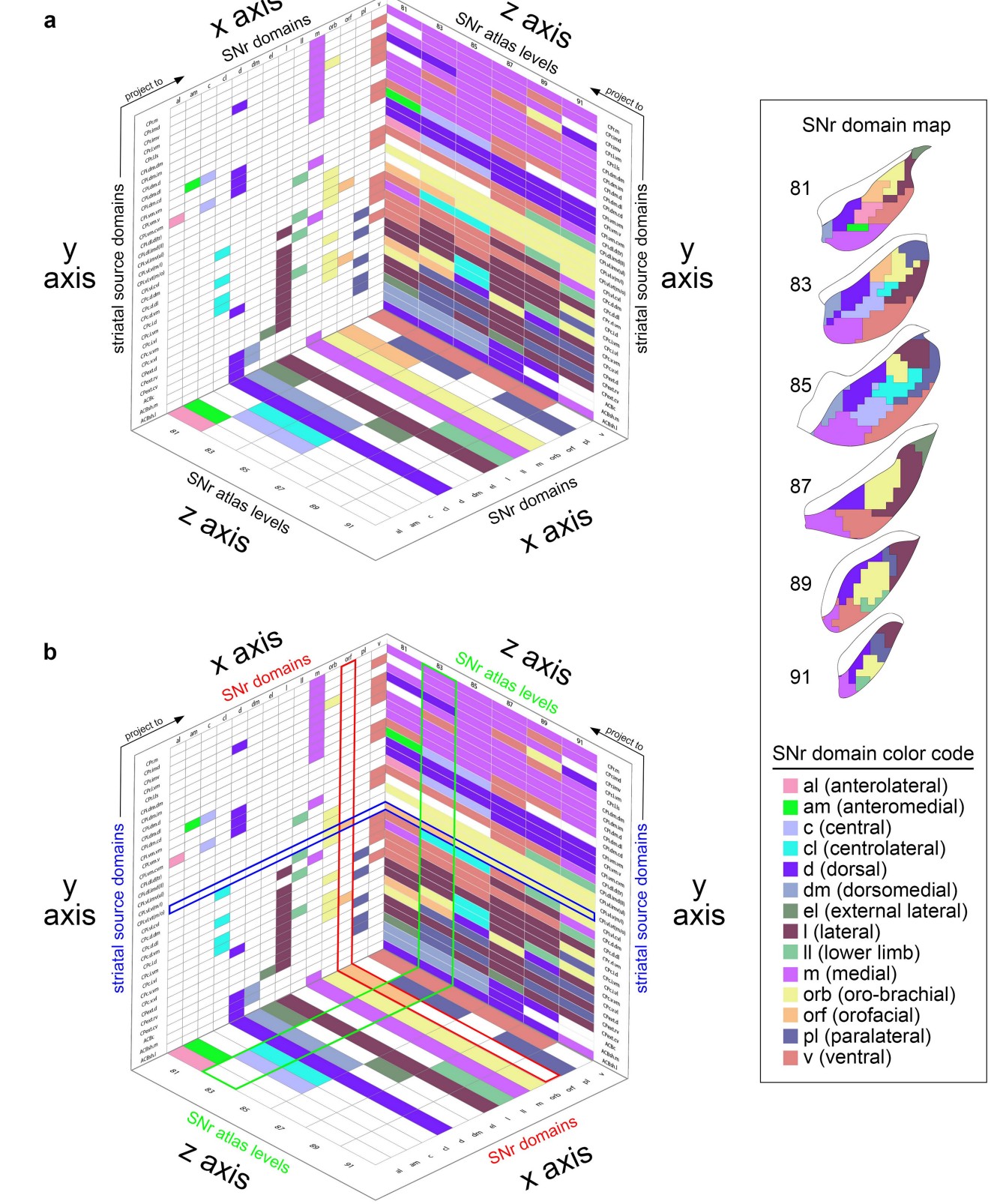

**Extended Data Fig. 7 |** See next page for caption.

**Extended Data Fig. 7 | 3D SNr matrix.** a) The 3D matrix shows the relationships between the striatal source domains, the SNr domains, and the SNr atlas levels. The striatal source domains (on the y axis) send inputs to the domains of the SNr (x axis) across its atlas levels (i.e., rostrocaudal level in the ARA, z axis). The x-y plane identifies the striatal source domains and the SNr domains they contribute to, with color-codes indicating the SNr domain contributed to (see the SNr color code key and SNr domain map, right; white cells indicate no contribution). The y-z plane depicts the striatal source domains and to which level of the SNr they provide input, with color-codes indicating the SNr domain they contribute to at each representative atlas level. The x-z plane illustrates each SNr domain plotted according to the SNr atlas levels it is present at. b) As an example, the row for the outer mouth/facial striatal domain CPi.vl.vt (m/o) is highlighted blue. This domain contributes to two SNr domains, the oro-brachial and the orofacial, as indicated by the yellow and orange colored cells, respectively, in the blue highlighted row of the x-y plane. The column corresponding to the SNr orofacial domain is highlighted red, and it can be seen that this SNr domain has two striatal inputs, CPi.vl.vt (m/o) and CPc.v.vm. Following the blue highlighted row onto the y-z plane, it can be seen that the CPi.vl.vt (m/o) contributes to the SNr.orf at atlas levels 81 and 83 (orange cells) and it contributes to the SNr.orb at 85-91 (yellow cells). On the y-z plane, the column for atlas level 83 is highlighted green, and it can be seen that the other orange cell in that column corresponds to striatal input domain CPc.v.vm. On the x-z plane, the SNr orofacial domain is highlighted red, and it can be seen that this domain only spans two levels of the SNr, 81 and 83. This 3D graph illustrates all of the interrelated factors of the SNr domains, their striatal inputs, and the atlas levels at which each are present.

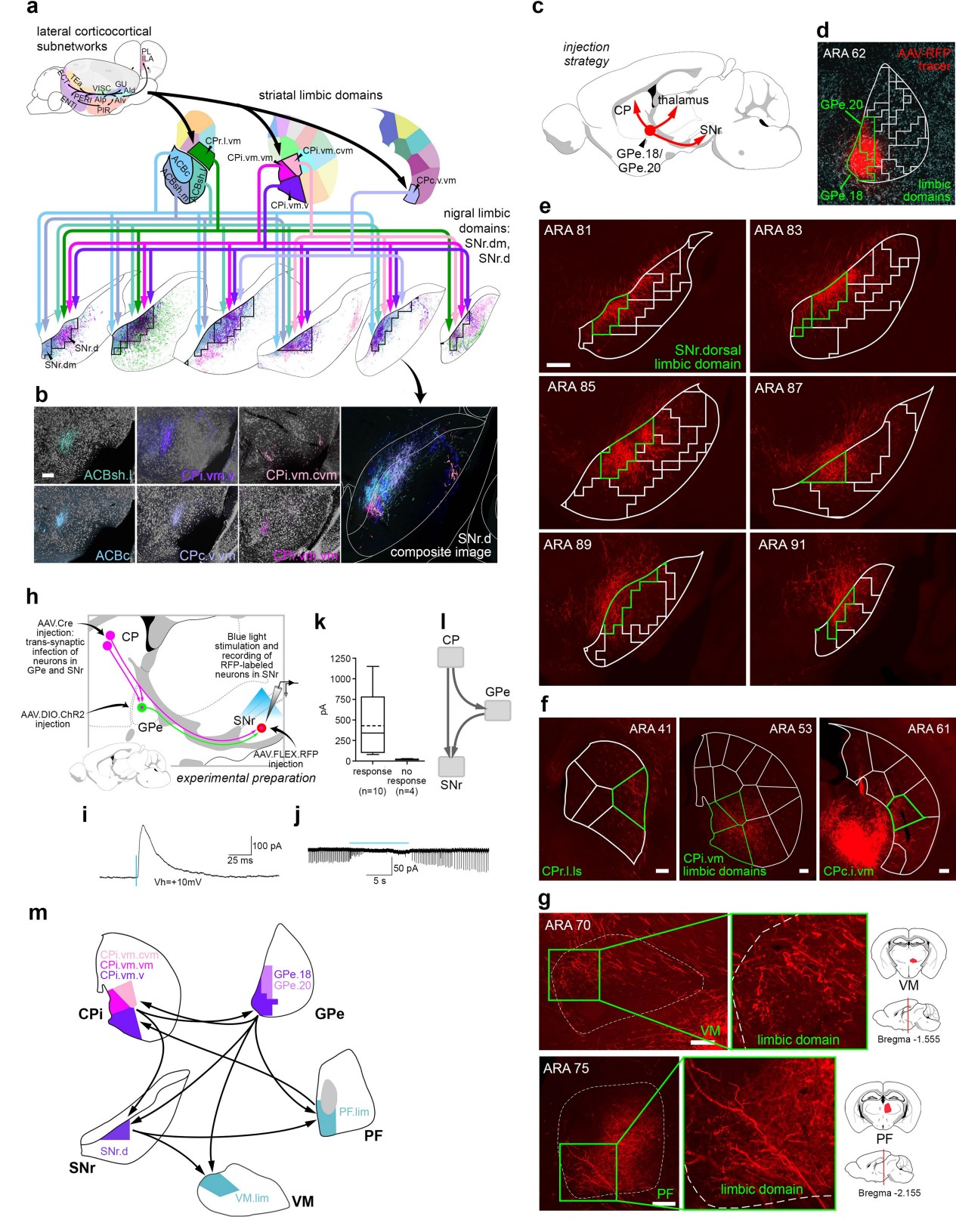

**Extended Data Fig. 8 |** See next page for caption.

**Extended Data Fig. 8 | The limbic subnetwork. a**, In the cortex, limbic areas like agranular insular areas (AId, AIp, AIv), temporal association area (TEa), visceral (VISC), gustatory (GU), perirhinal (PERI), ectorhinal (ECT), lateral entorhinal (ENTl), piriform (PIR), infralimbic (ILA), and prelimbic (PL) cortices, are networked together and form the lateral corticocortical subnetworks[25]. These areas project to a common set of striatal domains (CPr.l.vm, CPi.vm.vm, CPi.vm.v, CPi.vm.cvm, CPc.v.vm, ACBc, ACBsh.m, ACBsh.l)[5]. The colored lines show how those limbic striatal domains project in complex convergent/ divergent patterns to the SNr.dorsal (SNr.d) and SNr.dorsomedial (SNr.dm), forming the limbic cortico-striato-nigral pathways. The axonal reconstructions show the pattern of labeling for the inputs that defined these two nigral domains. **b**, Below the projection map, raw registered axonal images from ARA 89, pseudocolored to match their striatal source domain, are shown individually and in composite, illustrating their convergence (scalebar = 200μm). **c**, Injection of anterograde tracer demonstrates how the GPe projects to the other nodes of the basal ganglia-thalamic network. **d**, In this case (representative from n = 4), an injection of AAV-RFP in two of the GPe limbic domains (GPe.18 and 20, highlighted green) labels axonal terminals in **e**, the nigral limbic domain SNr.dorsal (highlighted green). The SNr.dorsal, GPe.18, and GPe.20 are all innervated by many of the same striatal domains. Thus the GPe projects densely to the SNr with a homotypic topography. Even though striatal projections to GPe innervate only a restricted rostrocaudal range of the pallidum, projections from GPe to nigra innervate its entire rostrocaudal length in a topographic fashion, similar to the striatonigral pathway. The same pallidal injection also labels axons that project: **f**, back into the striatum, reciprocally innervating the input striatal domains (highlighted green); and **g**, to the limbic domains of the ventromedial (VM) and parafascicular (PF) thalamic nuclei. **h**, To physiologically demonstrate re-convergence of the direct/indirect pathways on individual SNr neurons, one domain in the CP was injected with AAV1.Cre (pink), which can cross the synapse and infect neurons in GPe and SNr (pink dots); the SNr postsynaptic neurons are visualized with Cre-dependent RFP expression (red circles), while the GPe postsynaptic neurons were rendered optically excitable with injection of the Cre-dependent opsin AAV.DIO.ChR2 (green). Blue light stimulation while recording from RFP-tagged neurons in SNr elicited **i**, an inhibitory response that **j**, was capable of silencing activity in the SNr neurons. **k**, The majority (10/14) of neurons recorded showed a response to indirect pathway (pallidonigral) stimulation (mean±sd, responders: 427.7±393.5 pA; non-responders: 19.0±7.4 pA; n = 7 mice). This demonstrates that **l**, the direct and indirect pathways arising from the same striatal source re-converge in the nigra on the same set of postsynaptic neurons. **m**, All of this connectivity is summarized in the limbic subnetwork wiring diagram, demonstrating the precise intraconnectivity of the limbic subnetwork. For box plots, boxes demarcate first and third quartiles, whiskers show min/max, solid line is median and dashed line is mean. All photomicrographs are coronal, medial is left, and scalebars = 200μm.

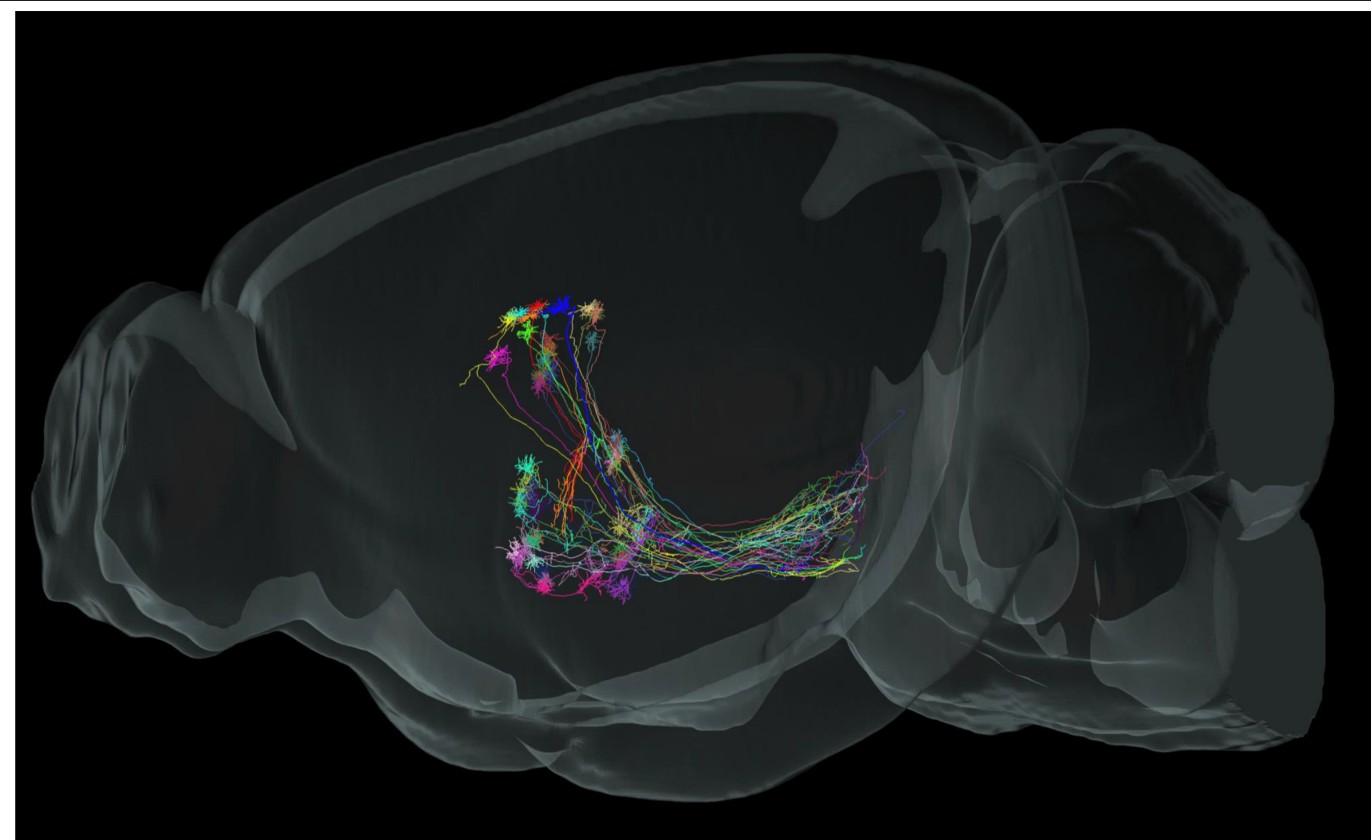

**Extended Data Fig. 9 | fMOST imaging and axonal trajectories of digitally reconstructed striatal neurons.** GFP expression was sparsely induced in a total of 30 striatal neurons (n = 7 mice), imaged with the fMOST system, reconstructed, and registered to the Allen Common Coordinate Framework version 3. This sagittal view of the data shows striatofugal axons in pallidum and nigra. Single-neuron tracing in primates showed that striatal axons terminate along their entire extent in SNr[27].

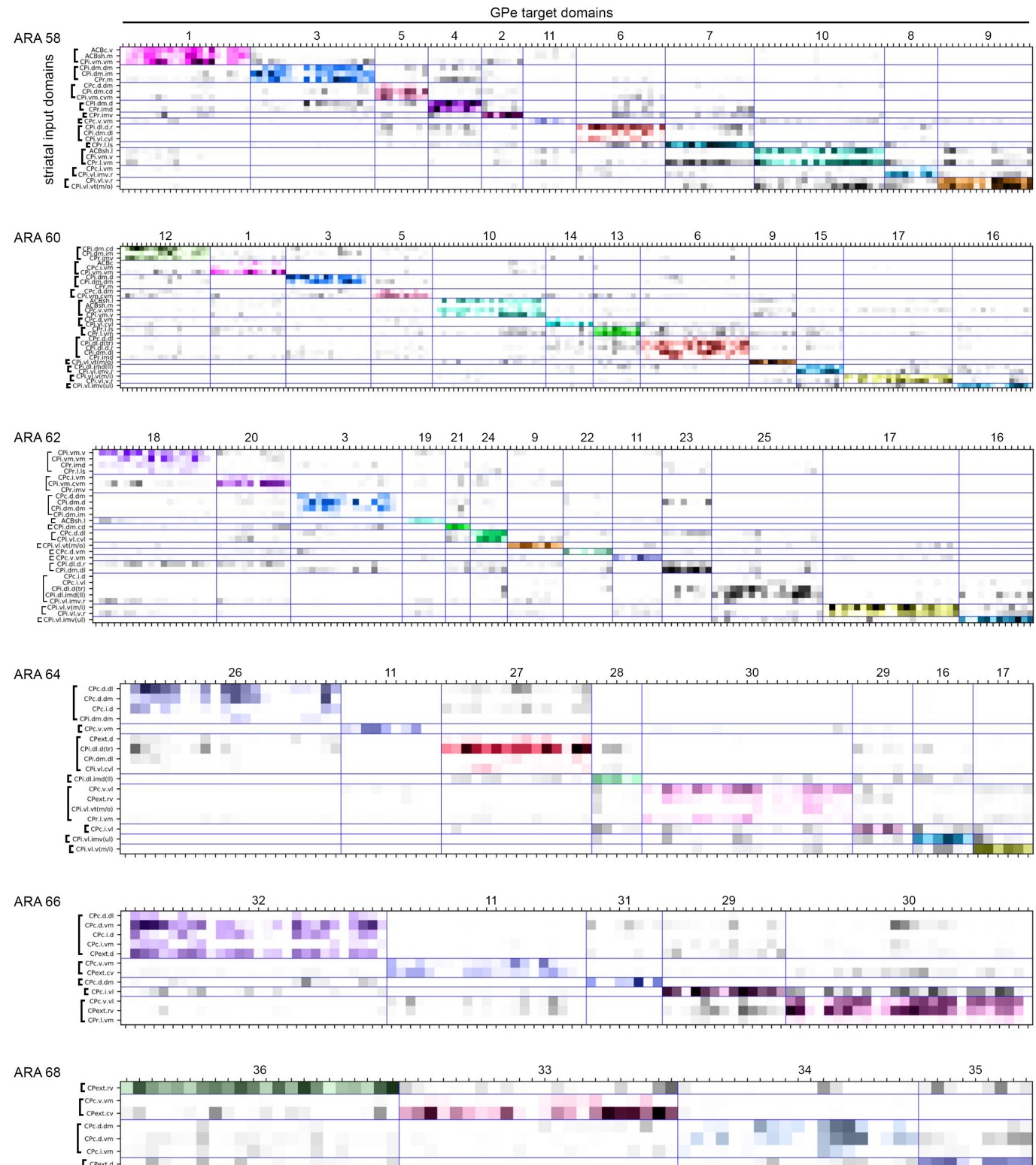

**Extended Data Fig. 10 | Striatopallidal matrices.** The entire striatopallidal dataset (36 injections, Extended Data Fig. 2) was quantified by box grid analysis, and the matrices are a visual representation of the percent coverage of a given grid box by axons from each striatal source domain (darker shading indicates greater coverage). These values were subjected to computational analysis with the Louvain community detection algorithm, which grouped together the striatal input domains (rows) that terminate in a common set of pallidal grid boxes (columns) at each representative level. The matrices presented here are ordered matrices, arranged to show the domains determined by the Louvain algorithm (for a full description of how the domains were determined, see Methods, *Network analysis*). The new pallidal domains lie along the diagonal and are colored to match the GPe domain map (see Fig. 2). The domain numbers are listed along the top of each matrix. Note how the domains tend to have fewer inputs than the nigral domains (cf. Extended Data Fig. 5), and even the pallidal domains with 4 or 5 inputs tend to be densely innervated by only 1 or 2 of those inputs.

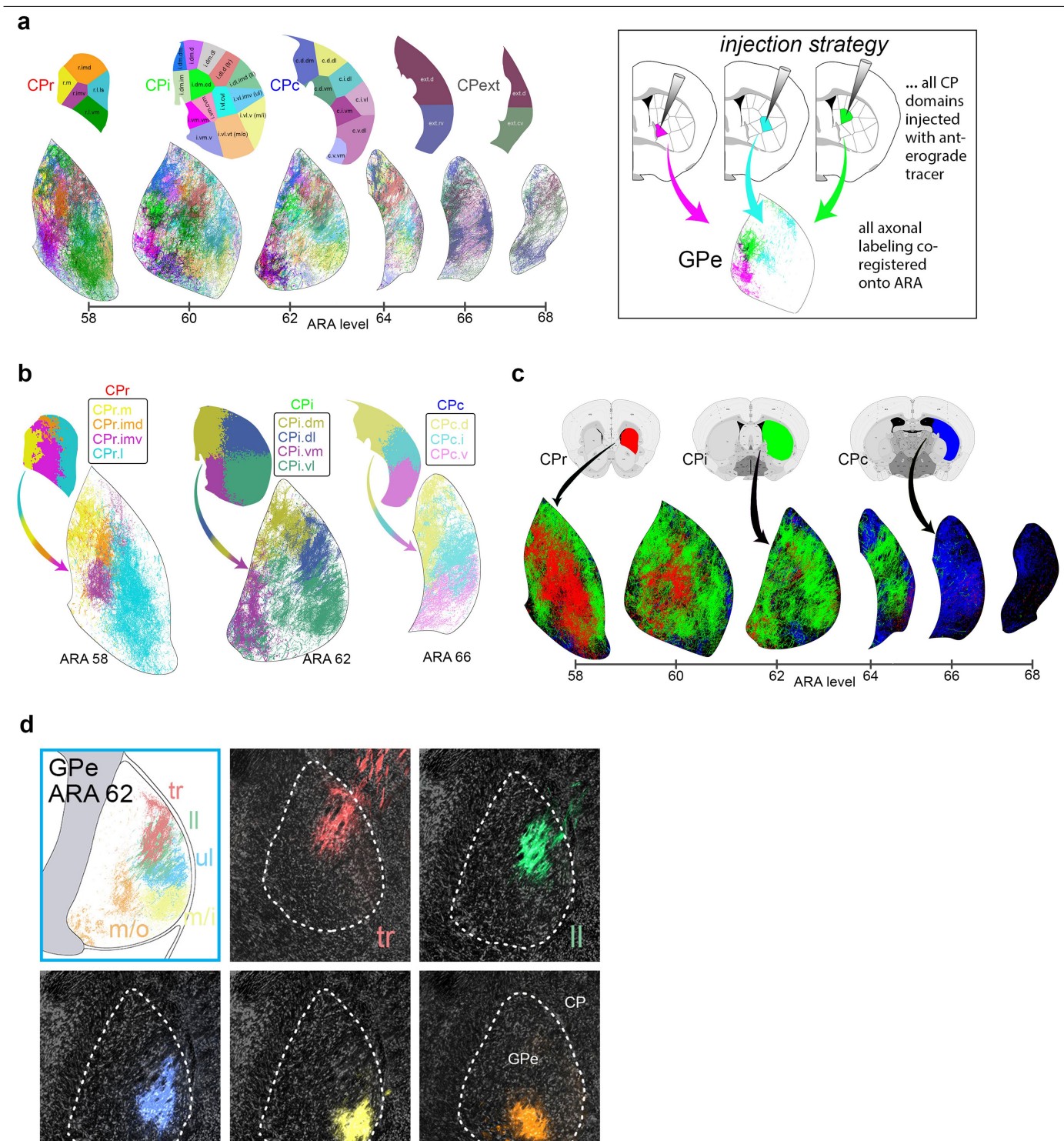

**Extended Data Fig. 11** | See next page for caption.

**Extended Data Fig. 11 | GPe projection maps.** The majority of striatal domains project to GPe with restricted terminal fields that have little overlap and convergence among one another. This largely parallelized striatopallidal topography is evident at the domain-, community-, and level-scale resolution in the reconstruction maps. **a**, Domain-level projection maps of all striatopallidal terminations, colored according to their striatal source domain. Collectively, the entire CP sends projections that cover the entire GPe. Projections from individual CP domains follow a rostral to rostral and caudal to caudal pattern (i.e., rostral CP to rostral GPe, etc.), a topography similar to the primate striatopallidal pathway[38]. No projections from any individual CP domain span the entire rostrocaudal extent of the pallidum. **b**, Community-level view of striatopallidal terminations depicts axons of all CP domains colored to match their parent community (e.g., reconstructions of axons from CPi.dl.d and CPi.dl.imd are colored blue for community CPi.dl), revealing that the dorsoventral and mediolateral topography of the CP is generally maintained in the GPe. **c**, Axonal reconstructions originating from all striatal domains of rostral, intermediate, and caudal CP levels are colored red, green, or blue, respectively. The rostral-to-rostral and caudal-to-caudal striatopallidal topography is easily apparent in these maps. The CPext projects to the most caudal levels of GPe (not shown here). **d**, The somatotopic map in the GPe is revealed by striatopallidal terminations from body region-specific striatal source domains: trunk (tr), lower limb (ll), upper limb (ul), inner mouth (m/i), and outer mouth (m/o). The raw source images are shown in the photomicrographs (coronal plane, medial is left, scalebar = 200µm) and the axonal reconstructions in the co-registered connectivity map. Axons were pseudocolored to match their striatal source domain. These images were used in the striatofugal analysis.

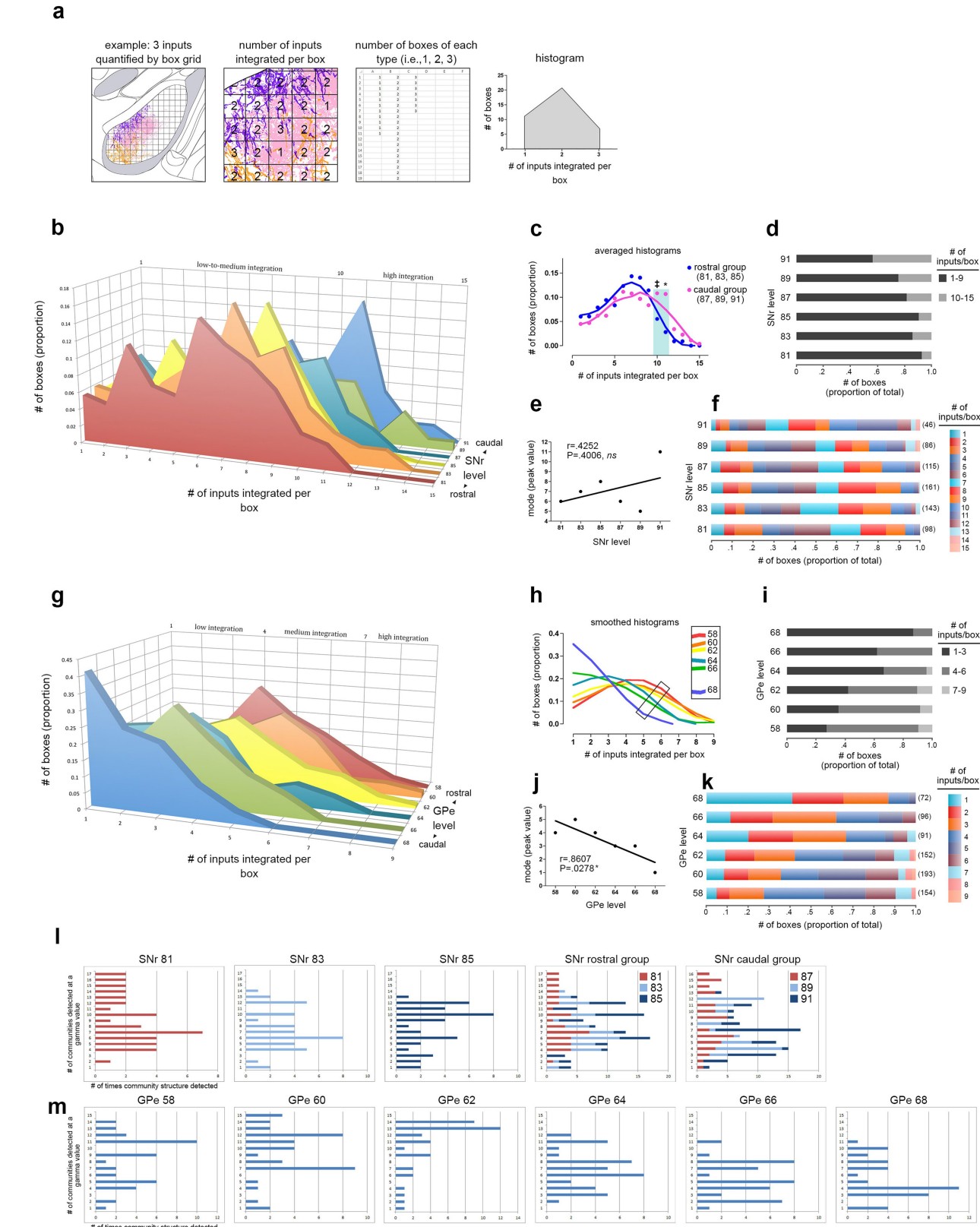

**Extended Data Fig. 12** | See next page for caption.

**Extended Data Fig. 12 | Box grid quantification of striatal output data.** Box grid data quantifies striatonigral and striatopallidal topography, describes integration trends in the pallidum and nigra, and serves as input data for the community analysis. The example in **a**, illustrates how the box grid analysis works using 3 inputs (pink, purple, and orange; left panel). Each whole and partial box in the ROI receives 1-3 inputs (center panel); the number of boxes of each input category is tallied (right panel) and plotted in a frequency distribution (histogram). **b**, Frequency distributions for each level of the SNr are shown rostral to caudal (front to back in the graph). The caudal 3 levels have a greater proportion of boxes devoted to integrating high numbers of inputs, as seen by the tails of their distributions sticking out in that range. This was validated statistically by comparing the caudal 3 and the rostral 3 distributions with ANOVA (* $P < .0033$, ‡ $.05 > P > .025$), which are averaged and summarized in **c**. The proportion of boxes integrating various numbers of inputs at each level is represented in the stacked bar charts: **d**, shows a categorized graph, with low-to-moderate integration (1-9 inputs) in black and high integration (10-15 inputs) in gray; **f**, shows the same data with each bin in a different color (the bars from left to right correspond to the legend from top to bottom), and the numbers in parentheses at the end of each bar are the total number of boxes at each level. **e**, Regression analysis of the peak of each histogram from **b** with SNr level shows no correlation ($r = 0.4252$, $P = 0.4006$). **g**, Frequency distributions for each level of GPe are shown caudal to rostral (front to back in the graph). The distributions are fairly similar in shape and exhibit a linear trend, from rostral GPe integrating higher numbers of inputs per box and shifting to successively lower integration with each caudal level. Smoothing of the histograms and plotting them in a single plane in **h** highlights this stepwise trend from higher to lower integration, rostral to caudal, respectively (inset). **j**, This trend was found to be significant when assessed by regression analysis ($r = 0.8607$, $P = 0.0278$). Stacked bar graphs in **i** and **k** are as in **d** and **f**. **l-m**, The Louvain community detection algorithm was run multiple times over a range of gamma values, with gamma modulating the number of communities (i.e., domains) detected in the nigra or pallidum. The bar graphs show the different community structures detected (y axis) and how many times they were detected (x axis); each integral increment on the x axis means one gamma detected that community structure, so the peaks represent the most commonly detected community structures. These survey analyses were run for each nucleus-level, and results for SNr 81–85 can be seen individually and stacked together (SNr rostral group), along with the stacked graph for the SNr caudal group (SNr 87–91, individual graphs not shown). The individual graphs for GPe are shown in **m**.

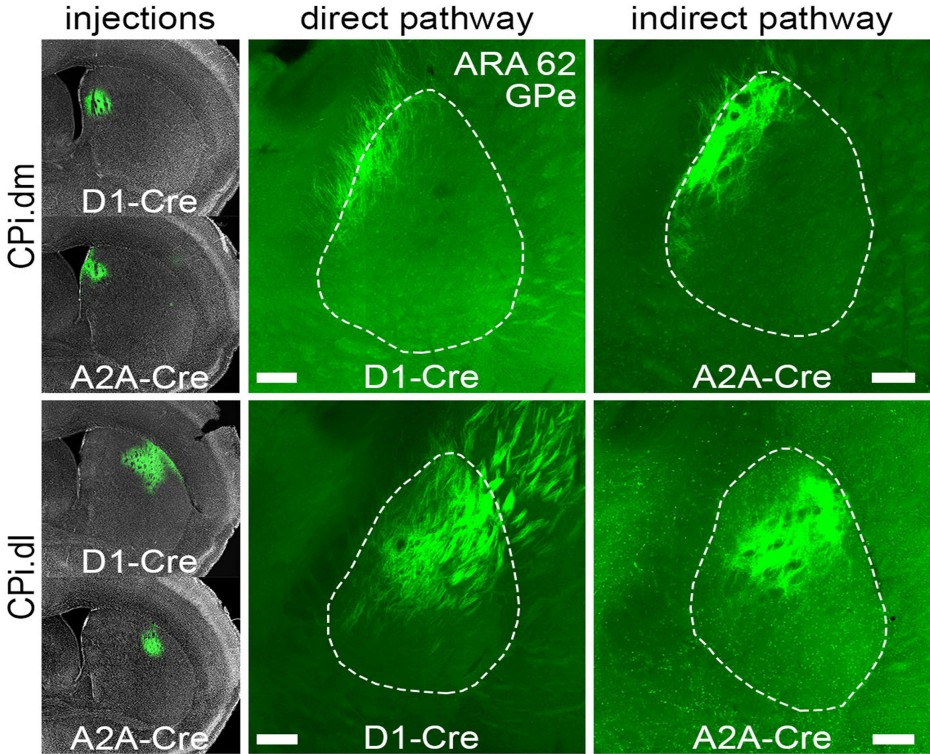

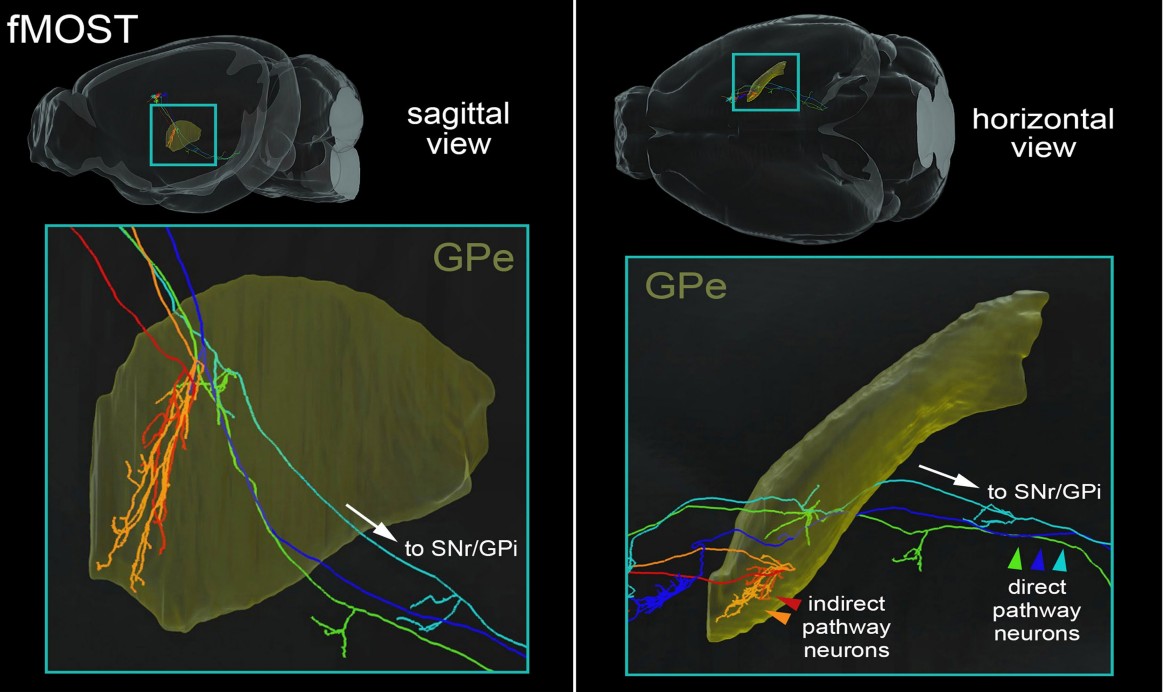

**Extended Data Fig. 13 | Direct and indirect pathway striatopallidal data.**
**a**, Direct and indirect pathway striatopallidal topography was compared by
Cre-dependent tracing of D1 dopamine receptor-expressing (direct) and A2A
adenosine receptor-expressing (indirect) medium spiny neurons. Injections of
AAV-DIO-EGFP were made into the same region of dorsomedial (CPi.dm) or
dorsolateral (CPi.dl) striatum in D1-Cre and A2A-Cre mouse lines (N = 4, one for
each target/Cre-line combination). Direct and indirect pathway terminals from
the same striatal region had the same topography in the GPe. This property is
also borne out at the single-cell resolution as seen in the fMOST data in **b**, where
2 indirect pathway neurons (red and orange) terminate in the same general
region as 3 direct pathway neurons (blue, green, and aqua). Given that the
direct pathway has "bridging collaterals" that terminate in the GPe (Cazorla
et al. 2014), we demonstrate that both the direct and indirect pathways'
terminals in the GPe have the same topography when arising from the same
striatal source region. Photomicrographs in **a** are coronal, medial is left, and
scalebars = 200µm.

**Extended Data Fig. 14 | Retrograde tracing of the SNr.m inputs with Fluorogold.** The striatal output tracing data indicate that the SNr.m and GPe.3 have very similar input identities, in that all of the inputs that group together in GPe.3 (i.e., CPi.dm.dm, CPi.dm.im, CPi.dm.d, CPr.m, and CPc.d.dm) also group together in SNr.m, and in fact are among the densest and most consistent inputs to SNr.m. These striatal regions comprise the dorsomedial corner of the CP adjacent to the lateral ventricular wall all along its rostrocaudal length. They collectively receive a highly similar set of cortical inputs as well[5], cortical areas that themselves form the two medial sensory associative cortico-cortical sub-networks[25]. Thus the two sensory associative cortical sub-networks project to a small set of striatal domains, which in turn send convergent projections to domains in GPe and SNr. Given their input similarity, we hypothesized that GPe.3 projects to SNr.m in an indirect pathway specific subnetwork (as summarized in the wiring diagram). **a**, To test this, we injected the retrograde tracer Fluorogold into SNr.m to trace its inputs. As predicted, a Fluorogold injection in the SNr.m domain at ARA 85 (representative case from n = 3) retrogradely labels input regions in the **b-c**, pallidum (GPe.3) and **d-e**, CP (all domains of CPr and CPi.dm.dm, CPi.dm.d, CPi.dm.im, and CPi.dm.dl). All photomicrographs are coronal, medial is left, and scalebars = 200μm. **f**, Wiring diagram models the medial associative striato-pallido-nigral subnetwork.

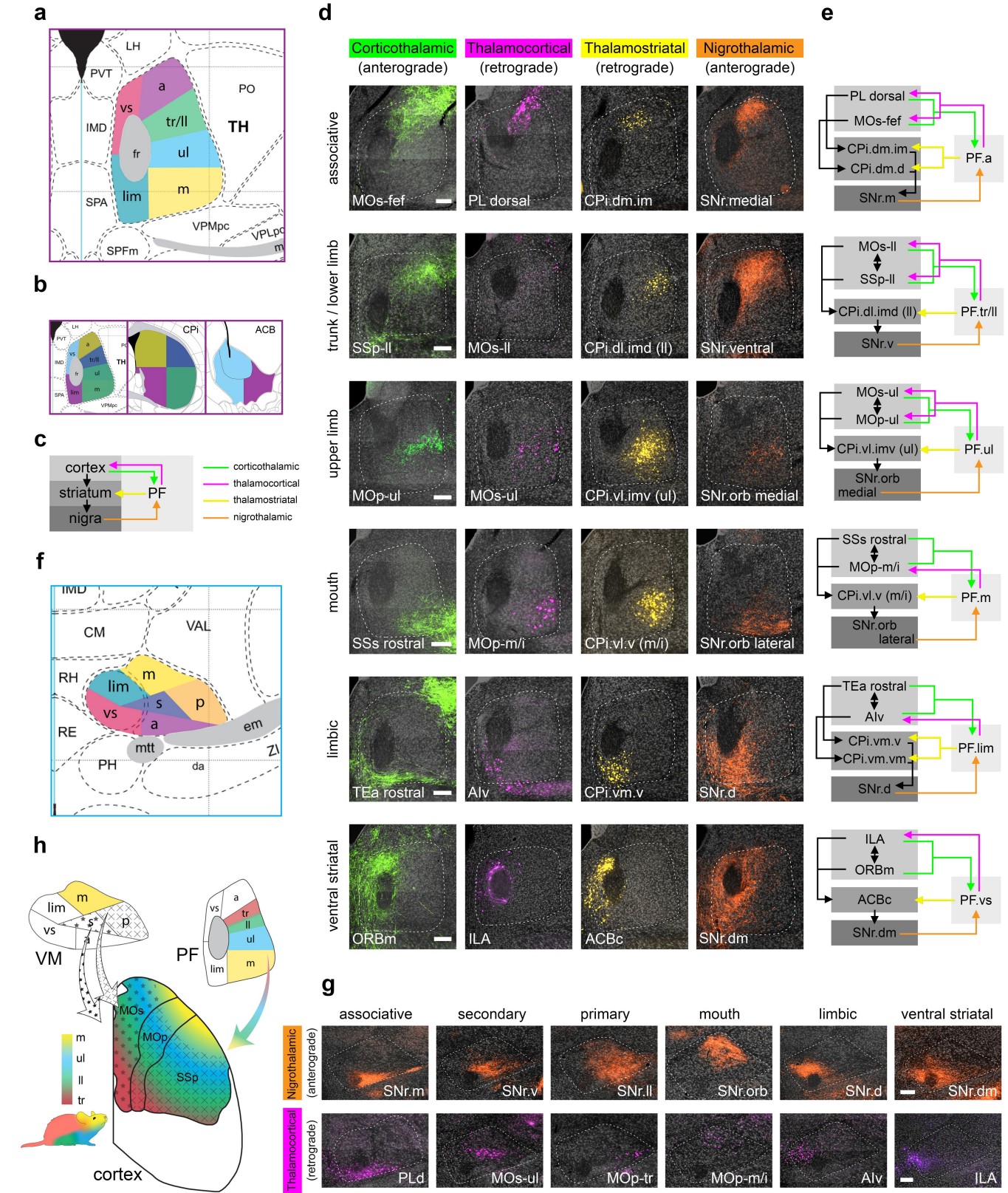

**Extended Data Fig. 15 |** See next page for caption.

**Extended Data Fig. 15 | Output channels in PF and VM thalamic nuclei. a**, The parafascicular thalamus (PF) has 6 output channels, territories that are interconnected with domains in the basal ganglia and cortical regions that are themselves interconnected; the colored maps in **b**, depict the general PF-striatal connectivity relationships. **c**, The prototypical network model illustrates the basic connectivity scheme of the cortico-basal ganglia-thalamo-cortical circuit. **d**, The connectivity of each of the 6 PF output channels with every other major node of the cortico-basal ganglia network (except pallidum and subthalamic nucleus) is shown. These data were generated with injections of standard anterograde (PHAL) and retrograde (CTB, Fluorogold) tracers in the cortex, striatum, and nigra (injection site is specified on each image). The labeling for each pathway is pseudocolored to match the color scheme in **c**. The nigrothalamic pathway data in the ventral striatal subnetwork (i.e., the image labeled 'SNr.dm') was generated using anterograde transsynaptic tracing, and thus represents synapse-specific pathway tracing from ACBc to SNr.dm to PF.vs (representative case from n = 3; see main text for details). The axonal labeling from SNr.dm to PF.vs terminates precisely in the predicted region of PF, the PF. vs (and VM.vs, see below). In **e**, the specific subnetwork models are shown to the right of each corresponding image series. **f**, The ventromedial thalamic nucleus (VM) also has 6 output channels, but in VM the motor output pathways to cortex are not organized somatotopically by body subregion, as they are in PF, but rather the VM.primary (VM.p) projects to all body regions of primary motor and primary somatosensory cortex, and the VM.secondary (VM.s) projects to all body regions of secondary motor cortex. The limbic (lim), ventral striatal (vs), associative (a), and mouth (m) output channels are similar to the corresponding channels in PF. **g**, Nigrothalamic and thalamocortical tracing data support the 6 channel model of VM. Note that the same transsynaptic tracing experiment described for **d** also resulted in the ventral striatal data in **g** (i.e., the image labeled 'SNr.dm'). **h**, A summary diagram illustrating the thalamocortical projection differences and similarities between VM (shapes) and PF (colors). The tr and ll zones are separated in this PF model because tr and ll exhibit a partially overlapping, partially separable thalamocortical connectivity pattern (data not shown), and emphasizing them separately underscores the somatotopic features of this pathway. All photomicrographs are coronal, medial is left, and scalebars = 200μm.

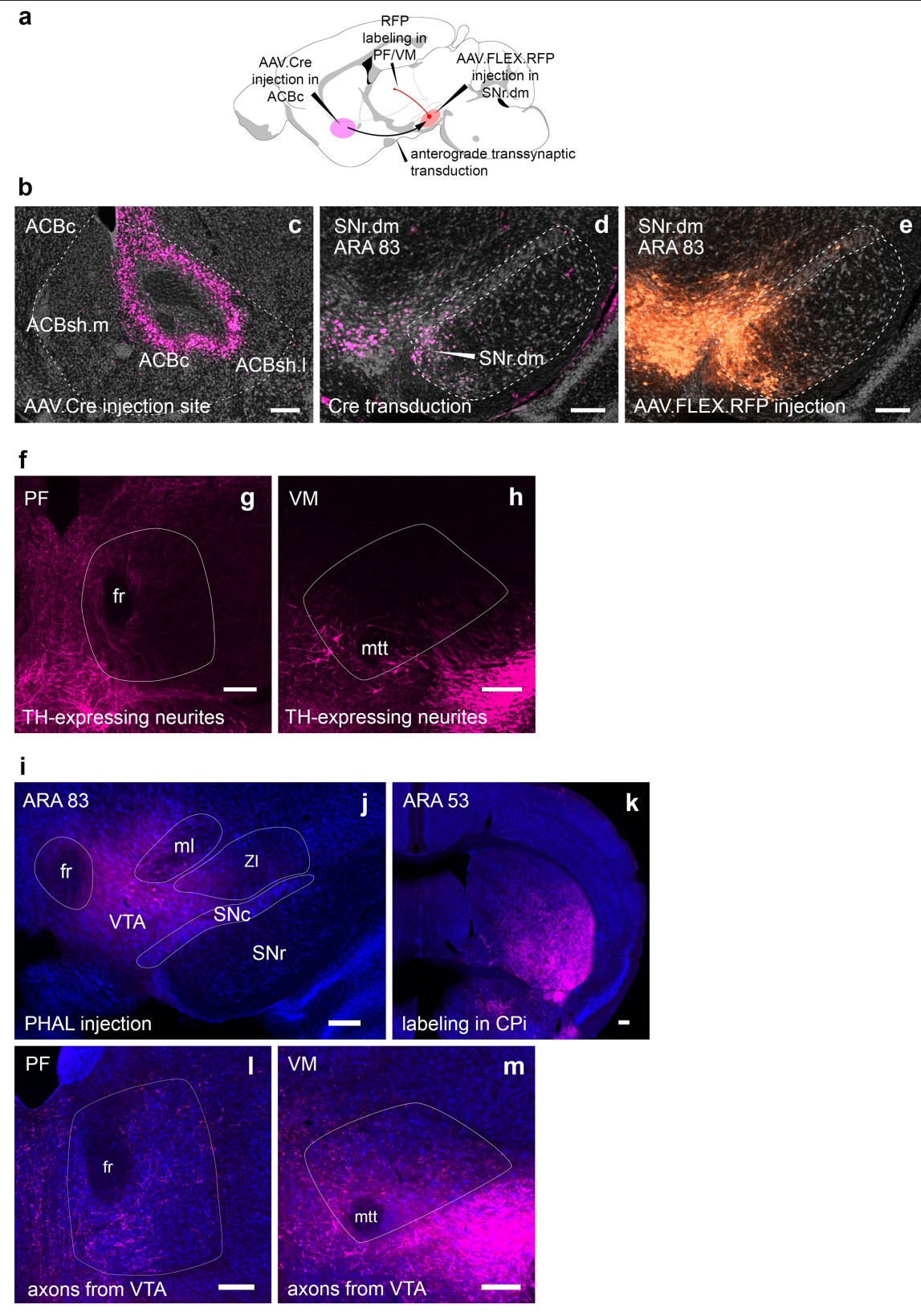

**Extended Data Fig. 16** | See next page for caption.

**Extended Data Fig. 16 | Anterograde transsynaptic tracing of the ACBc-SNr. dm-PF.vs pathway.** The injection strategy is summarized in **a** and injections are shown in **b**. AAV1-Cre was injected into the nucleus accumbens core **c** with Cre visualized by anti-Cre recombinase antibody (pink labeling). **d**, Anterograde transsynaptic transduction of neurons can be seen in SNr. dorsomedial. **e**, Cre-dependent fluorophore expression (AAV-FLEX-RFP, orange labeling) is seen in the transduced Cre-expressing neurons of the SNr, SNc, and VTA. The anterograde labeling from this injection in PF.vs and VM.vs can be seen in Extended Data Fig. 15d, g (images labeled 'SNr.dm'; representative case from n = 3 mice). To evaluate whether the RFP-expressing neurons in SNc and VTA contribute to the labeling seen in PF and VM, we evaluated **f**, catecholaminergic neurite distribution and **i**, VTA axonal terminations in PF and VM. Staining for tyrosine hydroxylase to reveal potential SNc/VTA dopaminergic innervation shows scant labeling in **g**, PF and **h**, VM thalamic nuclei. Images have an oversaturated contrast setting to reveal the neurites present, because they were few and faint. **j**, PHAL was injected into the VTA (pink cells are the injection site). **k**, Strong labeling in the striatum indicates that the injection quality was good. A sparse field of labeled axons is apparent over both **l**, PF and **m**, VM. Collectively, neither the TH staining nor the VTA axonal tracing appears sufficiently dense to account for the labeling shown in the 'SNr.dm' images in Extended Data Fig. 15d, g, which we assert arises from the nigral domain SNr.dm. All photomicrographs are coronal, medial is left, and scalebars = 200μm.

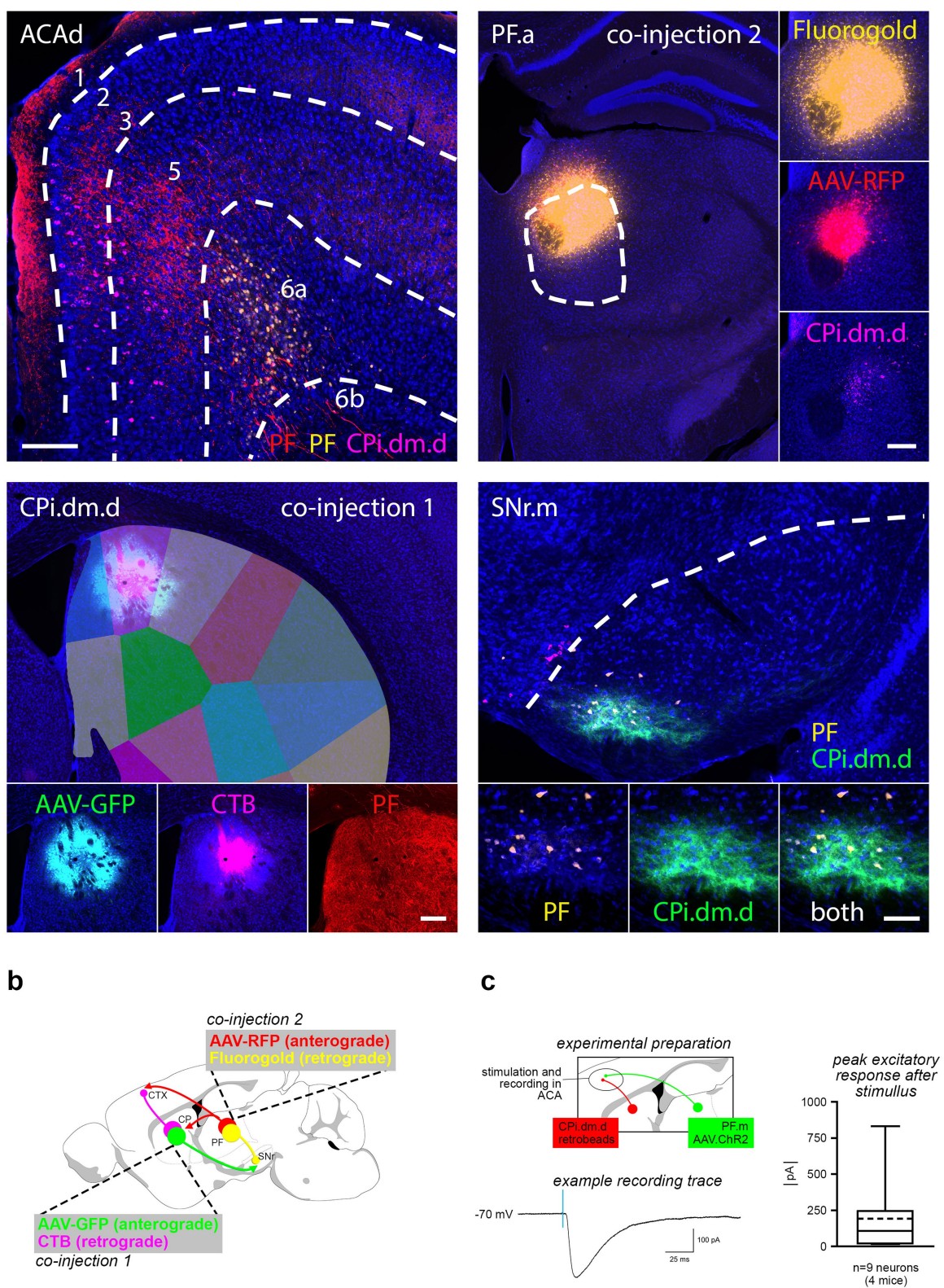

**Extended Data Fig. 17 | Double co-injection demonstrating the associative subnetwork loop. a**, One co-injection was placed in the striatum (CPi.dm.d, injected with AAV-GFP and CTB) and the other co-injection was made in the thalamus (PF.a, injected with AAV-RFP and Fluorogold). The transported tracers overlap in the cortex (ACAd) and nigra (SNr.m), revealing the discrete, closed-loop nature of the subnetwork (representative case from n = 3). Note that the RFP-labeled axons in the striatum and CTB-labeled somata in the PF reveal the direct thalamostriatal pathway between those two nodes. **b**, The injection strategy and prototypical circuit are shown in schematic. **c**, The closed-loop nature of this loop was verified electrophysiologically. The CPi.dm.d was injected with retrobeads to retrogradely label corticostriatal neurons, and PF.m was injected with AAV.ChR2 rendering the thalamocortical axons optically excitable. Retrobead-labeled neurons in anterior cingulate cortex were patch clamped and recorded during blue light stimulation with tetrodotoxin and 4-aminopyridine in the bath solution to ensure monosynaptic responses only. Nine neurons were recorded from 4 mice, and all 9 showed responses to stimulation (mean±sd, 191.3±256.4 pA), confirming that the associative subnetwork loop has a substantial recurrent component. For box plot, box demarcates first and third quartiles, whiskers show min/max, solid line is median and dashed line is mean. All photomicrographs are coronal, medial is left, SNr scalebar 100 μm, all other scalebars 200 μm.

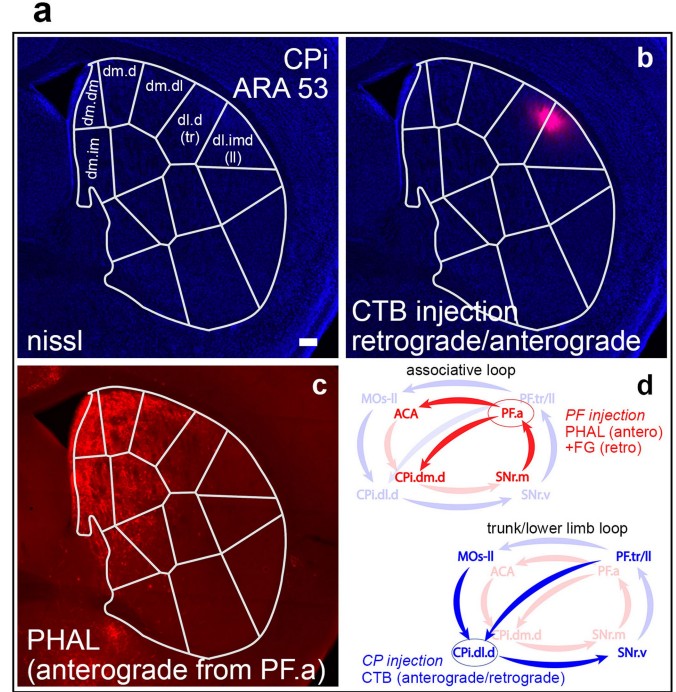

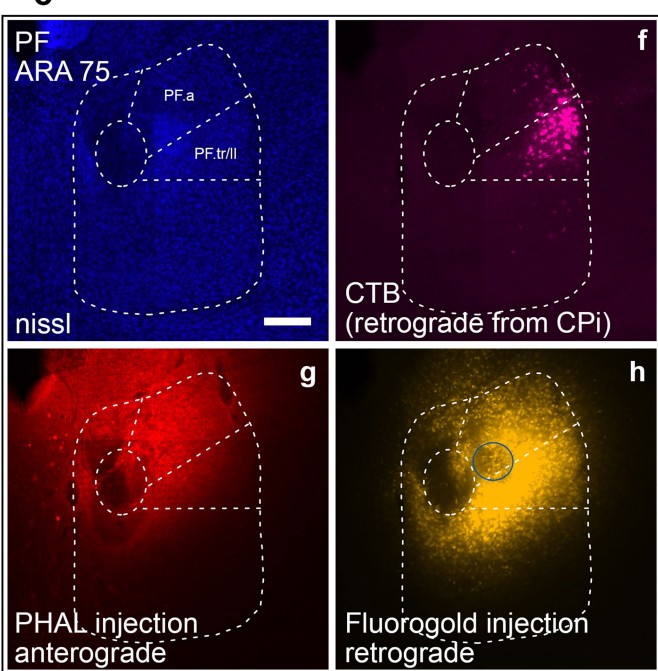

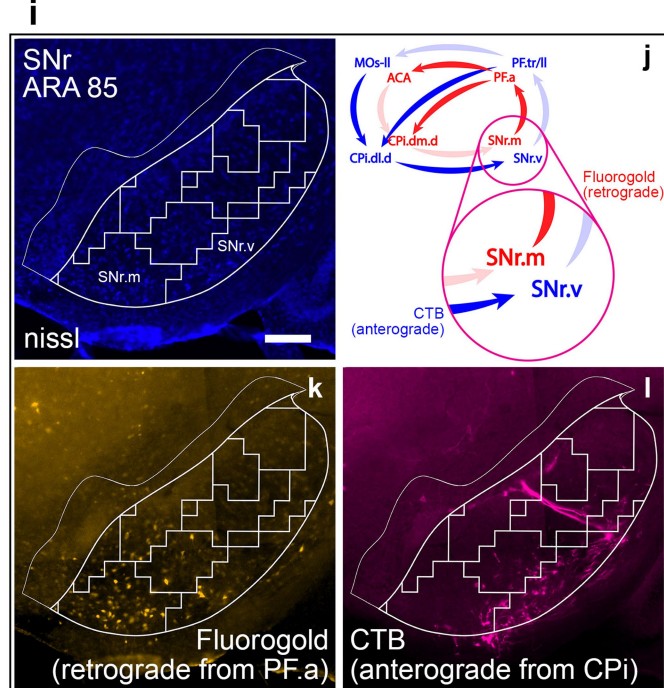

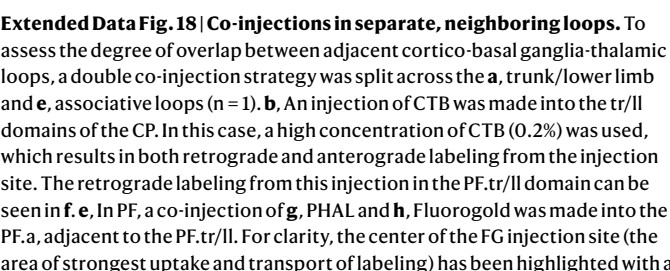

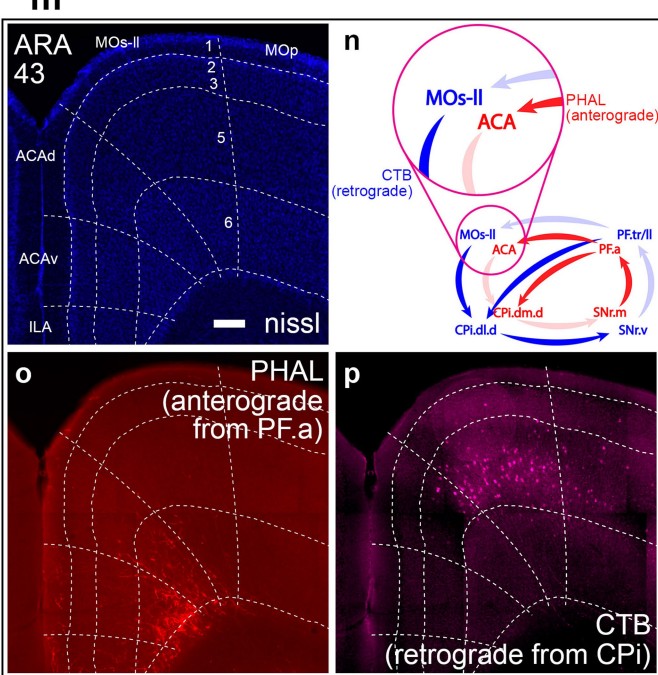

**Extended Data Fig. 18 | Co-injections in separate, neighboring loops.** To assess the degree of overlap between adjacent cortico-basal ganglia-thalamic loops, a double co-injection strategy was split across the **a**, trunk/lower limb and **e**, associative loops (n = 1). **b**, An injection of CTB was made into the tr/ll domains of the CP. In this case, a high concentration of CTB (0.2%) was used, which results in both retrograde and anterograde labeling from the injection site. The retrograde labeling from this injection in the PF.tr/ll domain can be seen in **f**. **e**, In PF, a co-injection of **g**, PHAL and **h**, Fluorogold was made into the PF.a, adjacent to the PF.tr/ll. For clarity, the center of the FG injection site (the area of strongest uptake and transport of labeling) has been highlighted with a blue circle. The injection strategy and predicted labeling patterns are summarized in **d**. Dense anterograde labeling from the PHAL can be seen in **c** in the CPi.dm.d and other domains of the CPi.dm but not in the tr/ll domains. The nigral and cortical labeling from the striatal and PF injections of these separate but neighboring loops shows only marginal overlap in their separate, adjacent regions in the **i-j** nigra and **m-n** cortex. The Fluorogold from PF.a strongly labels the **k** SNr.m while next to it the **l** SNr.v is labeled by anterograde CTB from the tr/ll domains of the CP. In the cortex, **o**, the PHAL labels the ACA while **p**, the retrograde CTB labels the adjacent MOs-ll. All photomicrographs are coronal, medial is left, and scalebars = 200µm.

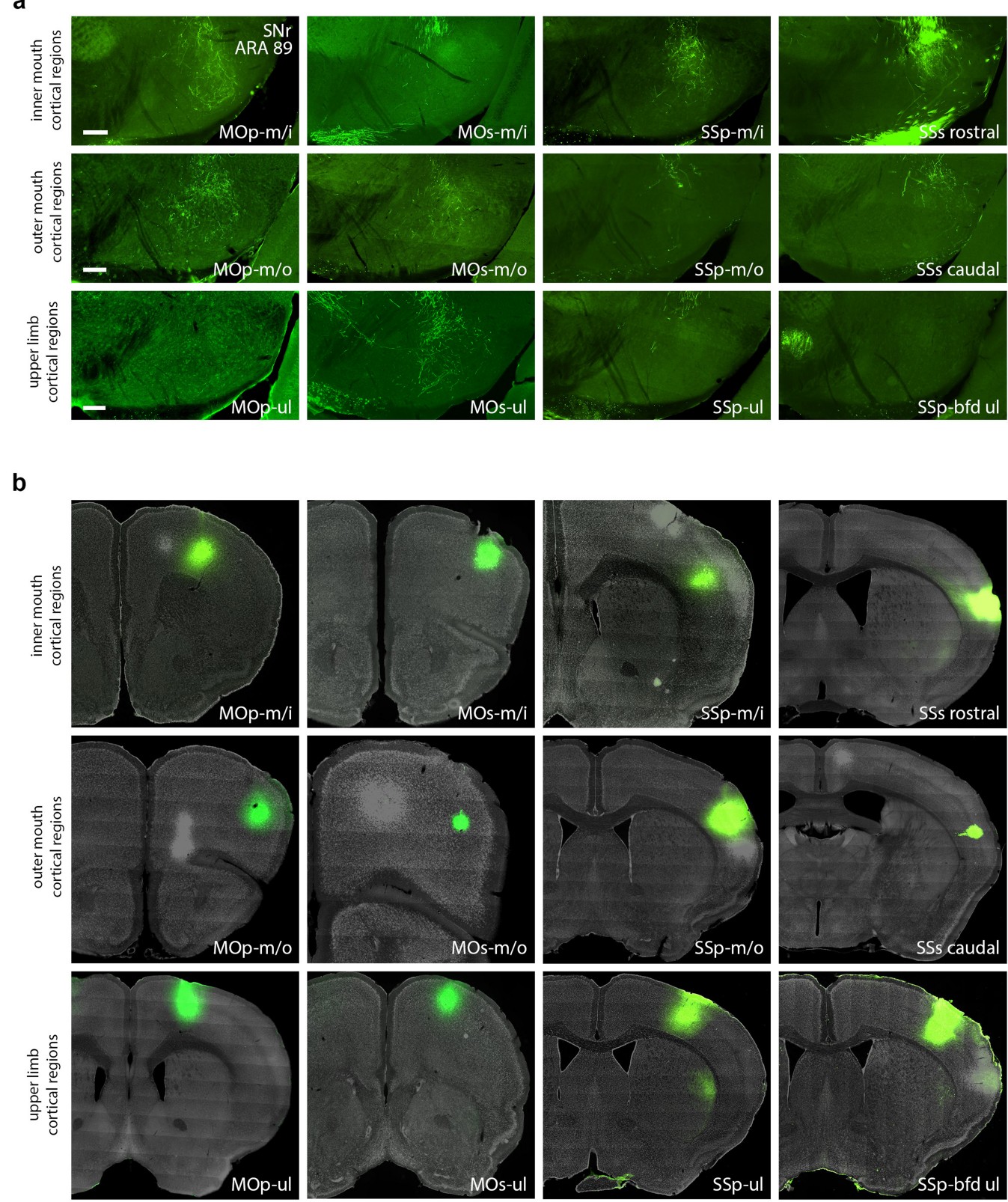

**Extended Data Fig. 19 | Corticonigral data. a**, Some but not all cortical regions of the inner mouth (m/i), outer mouth (m/o), and upper limb (ul) somatomotor subnetworks send direct axonal projections to the SNr.oro-brachial domain, as demonstrated by PHAL injections. **b**, Injection sites for the labeling in panel **a**. Other cortical regions do not appear to project directly to SNr (data not shown). All photomicrographs are coronal, medial is left, and scalebars = 200µm.

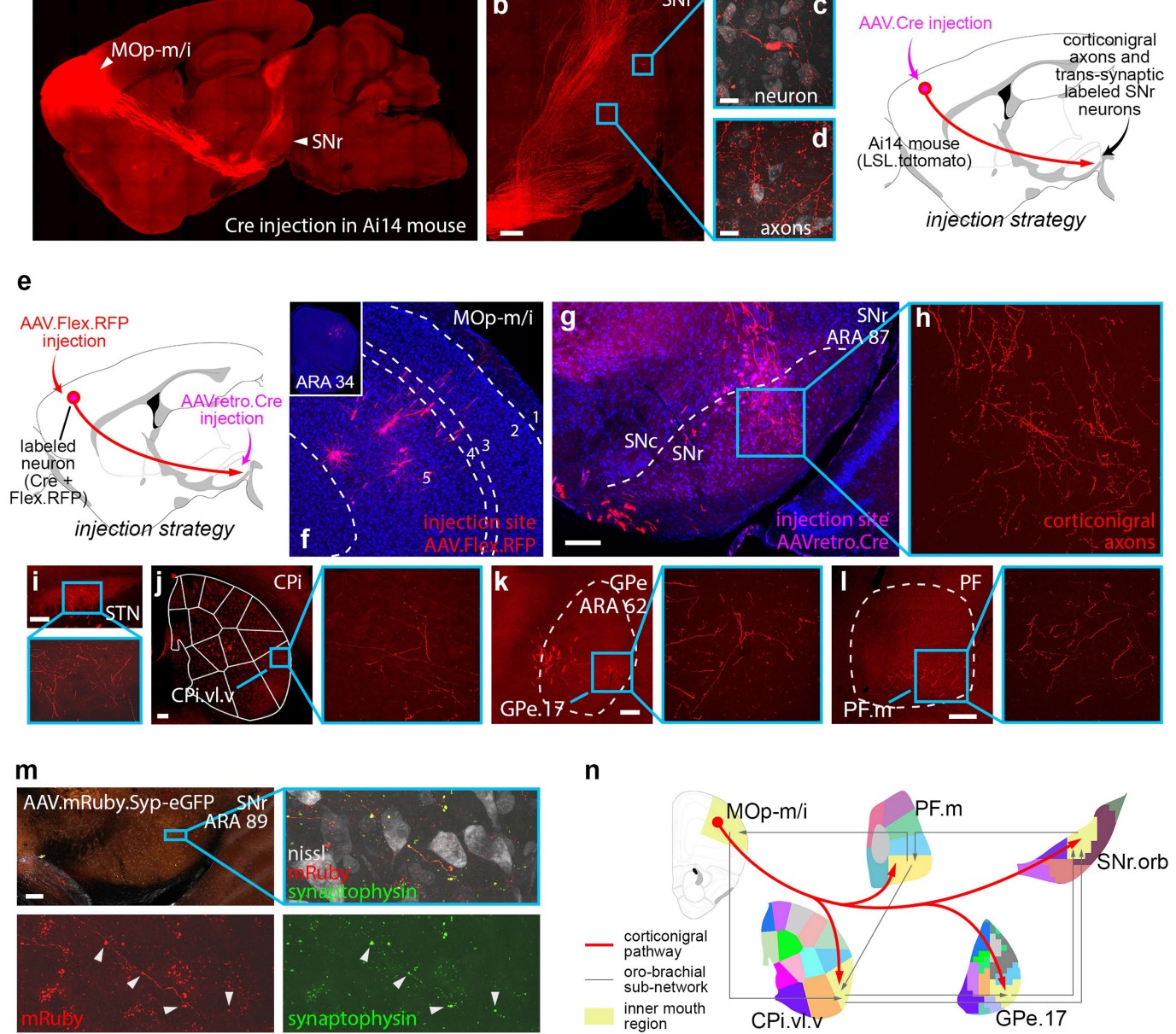

**Extended Data Fig. 20 | The corticonigral pathway. a**, The corticonigral pathway seen in sagittal section. Injection of AAV1-Cre into inner mouth primary motor cortex (MOp-m/i) in Ai14 mouse (representative from n = 2) releases tdtomato expression in the corticofugal pathway, revealing axons in **b**, the SNr. In the insets, **d**, individual axons bearing terminal boutons are discernable, and **c**, an SNr neuron is labeled through anterograde transsynaptic infection of the AAV.Cre virus. **e**, The corticonigral pathway from MOp-m/i was validated using Cre-dependent labeling, by **g**, infecting cortical axons in the nigra with AAVretro.Cre, which **f**, unlocked fluorophore expression in MOp-m/i neurons injected with AAV.FLEX.RFP (representative from n = 2). **h**, The labeled

MOp-m/i axons are seen in the SNr.orb, as well as the mouth domains of the **j** CP, **k** GPe, and **l** PF, as well as a restricted region of the **i** STN, indicating that at least some of these cortical neurons comprise the hyperdirect pathway. **m**, To visualize bona fide synaptic terminations in the SNr, the MOp-m/i was injected with AAV.mRuby.Syp-GFP (n = 1), which labels axons with red fluorescence and synaptic boutons with green fluorescence (arrowheads). **n**, The corticonigral pathway appears to be a unique feature of the oro-brachial subnetwork, contacting all mouth regions in it and suggesting that cortex can directly activate all other nodes in the subnetwork. Photomicrographs in **f-m** are coronal, medial is left. Scalebars in **c-d** = 20 μm, all others = 200 μm.

                             Nicholas N. Foster

# Reporting Summary

Nature Research wishes to improve the reproducibility of the work that we publish. This form provides structure for consistency and transparency in reporting. For further information on Nature Research policies, see our Editorial Policies and the Editorial Policy Checklist.

## Statistics

For all statistical analyses, confirm that the following items are present in the figure legend, table legend, main text, or Methods section.

| n/a | Confirmed | |
|---|---|---|
| ☐ | ☒ | The exact sample size (*n*) for each experimental group/condition, given as a discrete number and unit of measurement |
| ☐ | ☒ | A statement on whether measurements were taken from distinct samples or whether the same sample was measured repeatedly |
| ☐ | ☒ | The statistical test(s) used AND whether they are one- or two-sided<br>*Only common tests should be described solely by name; describe more complex techniques in the Methods section.* |
| ☒ | ☐ | A description of all covariates tested |
| ☐ | ☒ | A description of any assumptions or corrections, such as tests of normality and adjustment for multiple comparisons |
| ☐ | ☒ | A full description of the statistical parameters including central tendency (e.g. means) or other basic estimates (e.g. regression coefficient) AND variation (e.g. standard deviation) or associated estimates of uncertainty (e.g. confidence intervals) |
| ☐ | ☒ | For null hypothesis testing, the test statistic (e.g. *F*, *t*, *r*) with confidence intervals, effect sizes, degrees of freedom and *P* value noted<br>*Give P values as exact values whenever suitable.* |
| ☒ | ☐ | For Bayesian analysis, information on the choice of priors and Markov chain Monte Carlo settings |
| ☒ | ☐ | For hierarchical and complex designs, identification of the appropriate level for tests and full reporting of outcomes |
| ☒ | ☐ | Estimates of effect sizes (e.g. Cohen's *d*, Pearson's *r*), indicating how they were calculated |

*Our web collection on statistics for biologists contains articles on many of the points above.*

## Software and code

Policy information about availability of computer code

| | |
|---|---|
| Data collection | All epifluorescence images used in the striatal output axonal analyses were collected with the Olympus VS120 fluorescence microscope running Olympus VS-Desktop v2.9. High resolution confocal images were captured using an Andor DragonFly 202 spinning disk confocal microscope running Fusion v2.1.0.81 software. Lightsheet images were captured with a LifeCanvas lightsheet microscope running SmartSPIM Acquisition Software 2019v3. Electrophysiological data were collected using a MultiClamp700B Amplifier (Molecular Devices) running pClamp v. 10.7. |
| Data analysis | All standard statistical analyses were performed with GraphPad Prism v4.0c for Macintosh, including ANOVA, 2-sided t test with Welch's correction, Fisher's exact test, Pearson's r, and descriptive statistics. The algorithm implementing the Louvain analysis was obtained from the Brain Connectivity Toolbox (available at: https://sites.google.com/site/bctnet/) and executed in Python v2.7.<br><br>The Connection Lens v2.5.1 software used to register, threshold, and quantify the striatofugal axonal data was designed in-house. This software has not been released publicly yet, although it has been used in our previously published works (Hintiryan et al. 2016; Benavidez et al. 2021; Hintiryan et al. 2021).<br><br>Neurons from 3D images were reconstructed with Aivia v8.8.2, post-processed with the Quantitative Imaging Toolkit (available at: http://cabeen.io/qitwiki), and analyzed with neuTube v1.0z.<br><br>Electrophysiological signals were analyzed using Clampfit v. 10.7. |

For manuscripts utilizing custom algorithms or software that are central to the research but not yet described in published literature, software must be made available to editors and reviewers. We strongly encourage code deposition in a community repository (e.g. GitHub). See the Nature Research guidelines for submitting code & software for further information.

## Data

Policy information about availability of data

All manuscripts must include a data availability statement. This statement should provide the following information, where applicable:

- Accession codes, unique identifiers, or web links for publicly available datasets
- A list of figures that have associated raw data
- A description of any restrictions on data availability

Downsampled images of all data used in the striatofugal analyses are shown in Extended Data Figure 2. The quantified data for these cases can be accessed through our B.R.A.I.N. Lab website (http://brain.neurobio.ucla.edu/publications/). Also available are the SNr neuronal reconstructions from Figure 1k, the Supplementary Video, and an application presenting the projection maps of all axonal reconstructions.

# Field-specific reporting

Please select the one below that is the best fit for your research. If you are not sure, read the appropriate sections before making your selection.

☒ Life sciences ☐ Behavioural & social sciences ☐ Ecological, evolutionary & environmental sciences

For a reference copy of the document with all sections, see nature.com/documents/nr-reporting-summary-flat.pdf

# Life sciences study design

All studies must disclose on these points even when the disclosure is negative.

| | |
|---|---|
| Sample size | The sample size for the striatofugal analysis was determined by the number of domains in the striatum, i.e., 36. This consists of 29 domains in the dorsal striatum as described in Hintiryan et al (2016), as well as 4 additional dorsal striatal domains described in Methods and 3 domains in the ventral striatum: the core, medial shell, and lateral shell. The ventral striatum may or may not contain more subregions, but the core and shell are well-documented sub-compartments of the accumbens, and differences in connectivity patterns of medial and lateral shell have been described (e.g., Wright, Beijer & Groenewegen 1996, J Neurosci). Data were accumulated until each domain was injected with a distinct, isolated tracer deposit. All together, 138 animals received 1-3 anterograde tracer injections in the striatum. The grand total sample size of 268 reported in the Methods section includes these 138 plus animals with injections with injections targeting other parts of the cortico-basal ganglia-thalamic network.<br><br>Sample sizes for the electrophysiology experiments are comparable to previously published works both for number of subjects used and for number of neurons recorded per group (Thorn, Atallah et al. 2010, Neuron; Ji, Zingg et al. 2016, Cerebral Cortex). |
| Data exclusions | One and only one injection per striatal domain was used in the striatofugal analysis, as is standard in neuroanatomical research and as is necessary for the kind of community analysis we conducted. The best, most representative injection for each domain was chosen for the analysis. The others were excluded due to off-targeting of the injection site, missing/damaged tissue in the pallidal and nigral regions of interest, and weak tracer labeling of the axons or high background.<br><br>For electrophysiology experiments, data were excluded from neurons that were recorded outside of the target nuclei (i.e., the GPe, SNr, MOp-m/i, and ACA). |
| Replication | All striatal domains were targeted with injections multiple times to generate the striatofugal dataset. While the best, most representative cases were chosen for inclusion in the analysis data set, the other injections served as validation cases, demonstrating the replicability and consistency of labeling arising from each domain. |
| Randomization | Randomization is not relevant to the present work since animals were not compared across different conditions. |
| Blinding | Traditional blinding was not necessary since animals were not compared across different conditions. However, bias in image registration for the striatofugal analysis is the one area where the methods could have affected the results. In that regard, the image registration process, although not technically blinded, was performed by contributors without any a priori knowledge of the pathways under investigation. |

# Reporting for specific materials, systems and methods

We require information from authors about some types of materials, experimental systems and methods used in many studies. Here, indicate whether each material, system or method listed is relevant to your study. If you are not sure if a list item applies to your research, read the appropriate section before selecting a response.

## Materials & experimental systems

| n/a | Involved in the study |
|---|---|
| ☐ | ☒ Antibodies |
| ☒ | ☐ Eukaryotic cell lines |
| ☒ | ☐ Palaeontology and archaeology |
| ☐ | ☒ Animals and other organisms |
| ☒ | ☐ Human research participants |
| ☒ | ☐ Clinical data |
| ☒ | ☐ Dual use research of concern |

## Methods

| n/a | Involved in the study |
|---|---|
| ☒ | ☐ ChIP-seq |
| ☒ | ☐ Flow cytometry |
| ☒ | ☐ MRI-based neuroimaging |

## Antibodies

| Antibodies used | [antibody; vendor; catalog number]<br>1. rabbit anti-Phaseolus vulgaris leucoagglutinin antibody; Vector Labs; #AS-2300<br>2. monoclonal mouse anti-Cre recombinase, clone 2D8; Millipore Sigma; #MAB3120<br>3. donkey anti-rabbit AlexaFluor647 antibody; Jackson ImmunoResearch, #711-605-152<br>4. donkey anti-mouse AlexaFluor647 antibody; Jackson ImmunoResearch, #715-605-150 |
|---|---|
| Validation | Supporting documentation as to the validity of the above antibodies can be found at the following:<br>1. https://antibodyregistry.org/search.php?q=AB_2313686 ; see also Gerfen & Sawchenko, 2016, Brain Research<br>2. https://antibodyregistry.org/search.php?q=AB_2085748 and https://www.emdmillipore.com/US/en/product/Anti-Cre-Recombinase-Antibody-clone-2D8,MM_NF-MAB3120#documentation ; we have also used it previously in Benavidez et al. 2021 and Hintiryan et al. 2021 |

## Animals and other organisms

Policy information about <u>studies involving animals</u>; <u>ARRIVE guidelines</u> recommended for reporting animal research

| Laboratory animals | Mus musculus, male, 2-month old, wild type C57Bl6 and Ai14 (007908), obtained from Jackson Laboratories |
|---|---|
| Wild animals | No wild animals were used in this study. |
| Field-collected samples | No field samples were collected for this study. |
| Ethics oversight | Ethical oversight of experimental procedures was performed by the Institutional Animal Care and Use Committee (IACUC) of the University of California, Los Angeles, IACUC at the University of Southern California, IACUC at the University of California, San Diego, and the Institutional Ethics Committee of Huazhong University of Science and Technology. |

Note that full information on the approval of the study protocol must also be provided in the manuscript.

