## [Peer Review File · Nature]

Manuscript Title: The mouse cortico-basal ganglia-thalamic network

Editorial Notes:

Redactions – unpublished data

Reviewer Comments & Author Rebuttals

Reviewer Reports on the Initial Version:

Referee #1 (Remarks to the Author):

Review of Dong MS

This manuscript by Foster et al presents a new anatomical model of the cortico-basal ganglia (BG) circuits, based on a new and massive dataset. Extensive anatomical work over decades has produced a huge but often confusing literature, and numerous controversies remain. Foster et al use an impressive combination of modern anatomical tracers, unbiased digital reconstruction, automated image registration, and network structure analysis, to present a comprehensive connectome of key circuits in the BG. While confirming a few canonical features of the BG anatomy, albeit with finer resolution, this work also breaks new ground and reports a number of intriguing and surprising results, such as the detailed output channels in the SNr and parafascicular nucleus, the organized GPe output to the thalamus, and the highly parallel organization of GPe domains. The anatomical work is at once comprehensive and rigorous, and the results are of general interest, not only to basal ganglia researchers, but really to all students of neuroscience. It might become the new standard reference and an important resource for all neuroscientists.

However, in part due to the enormous complexity of the circuits covered, there are a number of weaknesses associated with the presentation and discussion of the data. In order for this to be used widely by researchers as a reference, clarity of exposition and illustration is critical. Unfortunately in its current form the manuscript is often difficult to follow. While the experiments are usually well conducted, the quality and extent of the analysis can be uneven. I hope the following remarks can be helpful to the authors as they prepare a revision.

1. The sheer amount of information makes the text difficult reading, even for experts. At least I found it difficult to follow at times. To reach a broader audience, it is recommended that the authors use some schematics to illustrate the major principles derived from this work.
2. Currently the paper cites many general references on BG function and disorders, but relevant anatomical papers with similar findings are not always adequately acknowledged.
3. The question of parallel organization vs. convergence, a key point of contention, is not discussed adequately. For example, see the old debate between Alexander et al and Percheron et al (TINS 1991). In addition, Figure 8 demonstrates a closed BG loop, but it would be useful to see an experiment which used injections in the orobranchial region of the PF but then used the striatal double-injections in a neighboring but separate loop, such as the trunk loop, to see how much the tracers are co-labeling in the SNR and in the cortex. Was this type of experiment done?
4. In an influential series of studies, Flaherty and Graybiel identified small and dispersed regions (matrisomes) in the striatum that receive divergent cortical inputs, but these striatal regions reconverge in the pallidum. Do present results shed light on this claim?

5. Discussion focuses on various broader implications. While such discussion is useful, it is also largely speculative. On the other hand, discussion of the relationship between current results and previous anatomical findings is too limited. It would be important to define exactly which findings are novel, and how they advance beyond previous work.

6. In the Results, there is limited information on the type of viral injections and mice. Such information should be provided consistently for each experiment. On a related note, it is sometimes not clearly explained what tracer is being used in the legends or the figure. For example, in Figure 5A, the method is explained in the caption but not in the figure. Then in figure 5C, the caption only says 'an injection' - so it leaves the reader concluding that the methods used in figures 5A and C are identical. Other figures have better descriptions, but there is no uniformity in the communication of the technique used.

7. They say that "many of the domains in the SNr formed by these inputs span multiple levels of the SNr., i.e., the SNr domains have a 3-dimensional shape." a 3D rendering of this would be helpful.

8. For tracer injections, they only show a schematic and a single representative example. More histology is needed to show their 33 injection sites. They say their data is fully presented on their connectome website, but I could not find it. Perhaps this could be made more explicit.

9. Contrast enhancement appears excessive in some pictures (See fig 2b, 2d, 2f 4d,,8a, s6).

10. how did they control spread in their viral injections? It appears that they used iontophoresis for some, but they also just used pressure injections. This causes a decent amount of spread with the amount used (e.g. 50 nL). Moreover, how did the authors avoid tracer and or virus leaks into the cortex during insertion and withdrawal of the pipet? This could result in unintended cortico-cortical projection labeling, and could potentially alter interpretation of the results in Figure 8.

11. They say "Although not quantified, the density of the indirect pathway terminations in GPe appears denser than the direct pathway" (L458-459). They should either quantify this or remove the statement.

12. The 3D matrix (Figure 2H) that is presented not well explained and it is difficult to understand how to use it. Diagrams that explain the tracing experiments need to be more extensively used. Currently there is only an occasional graphical description of the injection strategy used, such as in Figures 7B or 8A. These should be more frequent in the paper.

13. It would be helpful to provide more quantitative analysis to support the major conclusions. The anatomical description is sometimes anecdotal. Perhaps in modifying the presentation of the 3D matrix, it could be used as a recurring motif to better communicate the statistical evidence for the claims that are being made in each figure - for example by highlighting the relevant cells of the matrix in each figure,

14. Figure 2 and Figure 3B: The authors claim that the SNR domains are extended in the rostro-caudal axis and show that the axons from the striatum pass through this axis. What about synaptic contacts made?

Minor points (mostly grammar and usage, careful proofreading is needed as there are numerous examples)

15. L116: serial circuit? Not exactly serial if the loop is closed.

16. L139: 'Comprised of.' Should be 'composed of' or 'comprise'. 'Each receive.'

17. L149: foreshortened? Not sure what is meant here. 'Inadequate' perhaps.

18. L264: 'inputting', providing input to?

19. L679: 'The major likeliest.'

20. L690: missing 'in' after 'resides'.

Referee #2 (Remarks to the Author):

In this manuscript Foster and colleagues perform a tour-de-force anatomical dissection of the cortico-basal ganglia- thalamic network in the mouse. Using advanced technical approaches that they have previously established, they performed over 700 injections of various tracers (mostly viral) in striatum, GPe, SNr, thalamus, and cortex. Using this rich dataset they have identified multiple structural subdivisions they call 'domains' in SNr (14), GPe (36), and parafascicular and ventromedial thalamic nuclei (6). They also state that they identified 6 parallel cortico-basal ganglia-thalamic subnetworks that "sequentially transduce specific subsets of cortical information with complex patterns of convergence and divergence through every elemental node of the entire cortico-basal ganglia loop". With these techniques they uncover several interesting properties of cortico-basal ganglia thalamic circuits, including the presence of direct cortico-nigral projections in the oro-brachial domain, nigral dendritic arbors that appear to conform to the size of their particular domain (with potentially some overlap at domain boundaries), and a higher degree of structural convergence within the striatonigral "direct" pathway than the striatopallidal 'indirect' pathway. In addition, a single example of a functional closed loop was provided via slice physiology in the thalamo-cortico-striatal segment of the oro-brachial subnetwork. They conclude that these data provide significant conceptual advances regarding both the structural and functional organization of cortico-basal ganglia circuits.

These data provide an incredibly rich source of information regarding cortico-basal ganglia thalamic networks that will be a valuable resource for the field. Understanding structural and functional connectivity in these networks is very important for understanding the processes of action selection in both healthy and disease states, so the topic is quite significant. However, I have several significant issues with the manuscript as written as noted below.

Major points

1. The provided anatomical data is very interesting and important, but for many readers the most exciting part of the manuscript will be how to translate these structural findings into information regarding functional connectivity. From the abstract, it appeared that this would be one of the main points of the manuscript, but functional data were limited to Figure 8C. I would have liked to see more electrophysiologic analysis, ideally *in vivo*, to determine if the observed structural integration actually leads to functional integration. Related to Fig. 8C, one of the most exciting aspects of the manuscript is the electrophysiologic (*ex vivo*) demonstration of a closed-loop circuit, which has been proposed as an important circuit motif but has been difficult to directly demonstrate. However, there are significant issues with the design and interpretation of this experiment. The authors explain that AAV1 can be used for anterograde transsynaptic infection of post-synaptic neurons in a particular pathway. However, they then go on to use AAV1 to attempt to perform typical anterograde (non-synaptic) infection of neurons. This is problematic, given that in later experiments, unintended transsynaptic infection could affect the results. This is a particular problem in the slice electrophysiology experiments, where use of AAV1 injected into thalamus could trans-synaptically infect cortical neurons. Thus, the optogenetically-evoked synaptic responses observed in cortical neurons may be coming from neighboring cortical neurons, and not thalamic neurons. As this finding is a key point of the paper, the authors should repeat these slice experiments with a non-transsynaptic AAV serotype. Additionally, the authors should increase power in this experiment, since they are currently drawing these conclusions from only 2 animals. In addition, they should make clear that these findings may or may not generalize to other cortico-basal ganglia-thalamic loops (i.e. even if this is a closed loop, other circuits may not be).

2. Related to this is the fact that one of the major claims of the manuscript is that there is clear evidence of segregated closed loops, implying that this is the general motif used by cortico-basal-ganglia-thalamus circuits. However, to make that claim, which is exciting, there would need to be clear demonstrations of this phenomenon in other circuits as well. This is particularly the case since the authors demonstrated the closed-loop motif in the oro-brachial subnetwork, which was

unique in having direct cortico-nigral projections. Thus, findings in this subnetwork may not generalize to other networks.

3. The authors show that there is a pathway from accumbens, to SNr, to PF and VM thalamus using a dual-virus trans-synaptic AAV1/cre strategy. They claim that, while they also observed infected neurons in VTA and SNc (neurons downstream of accumbens), none of these neurons are projecting to PF/VM thalamus, because they don't express TH. However, a substantial number of neurons in VTA are GABAergic or Glutamatergic, do not express TH, and project to various cortical and subcortical targets throughout the brain (Breton, 2019). Thus, a TH stain is not sufficient to rule out projections to thalamus from VTA. The authors should do more to address this concern.

4. Because there is so much information in the manuscript, it can be difficult for the reader to distill the most important messages from the available information. Focusing in on one specific important question in the field (e.g. presence or absence of closed loops), and then digging into it more deeply, would make for a more compelling manuscript, as it would showcase the value of this resource.

5. Although past work demonstrating convergence in cortico-basal ganglia thalamic loops is acknowledged (e.g. from Suzanne Haber's lab), and the authors state they are not suggesting this is not present, the manuscript overall does not seem to give these data much weight. I suggest a more balanced discussion of the two alternative views, especially because direct assessments of neuromodulatory systems that might serve as agents for spiraling loops were not assessed in this manuscript.

6. The authors pay special attention to a circuit involving GPe.3 and SNr.m, showing that both regions receive inputs from the same several regions on the medial wall of the striatum. While they show later that, in general, direct and indirect pathway striatal pathways converge into the same areas of SNr, they don't explicitly show that the GPe.3 region projects to SNr.m. This fact should be explicitly shown in the section when they are discussing the case of GPe.3 and SNr.m.

7. Please clarify how representative injections were chosen, and how were possible differences in injection volume/cell transduction by virus were accounted for in the analysis procedures (e.g. was there normalization for this?).

Minor points

1. In the first paragraph of the Results, the authors simply say "injections of anterograde tracers were made ..." – this should be more descriptive (were AAVs used? PHAL?).

2. Figure 1E is not very clear. An outline or darker background on the brain slices may help clarify that there are two slices of brain being showed there. Also eliminating the overlap between the two slices may help.

3. The point being made about striatal projections along the rostrocaudal range of the SNr is a bit confusing. "Even though the striatal projections to the GPe innervate only a restricted rostrocaudal range of the pallidum, projections from GPe to nigra innervate its entire rostrocaudal length.." The figures pointed to (4A, 5B), to me, look like striatal projections to GPe do pretty much cover the entire rostrocaudal length. If this is an important point to make, the authors should clarify what they mean. If it is not important, this section in the paper should be limited to the convergence of striatonigral and pallidonigral projections, as this is a much more interesting and important point.

Referee #3 (Remarks to the Author):

In this manuscript by Foster et al, the authors present extensive rodent neuroanatomical tracing studies to characterize basal ganglia networks. This work builds on studies performed in the 1970s-90s using traditional anterograde and retrograde tracing methods in nonhuman primates and rodents, as well as their own more recent work using high-throughput analysis of tracing to describe cortico-striatal pathways in rodents. The current study's central premise is that the traditional 3-channel view of cortico- basal ganglia-thalamocortical pathways (sensorimotor, associative, and limbic), which in turn was based on older tracing studies from the 70s-90s, does not capture the complexity of the connections within this circuit, and that there are many more channels. While in their previous study, the authors focused on corticostriatal connections, in this study they performed numerous injections to examine the succeeding steps in the cortico-basal ganglia-thalamocortical circuit: the GPe, SNr, and thalamus. These studies are powerful in that they provide a more extensive "catalog" of the connections, indicating areas of convergence, divergence, and overlap between channels. Evidence of such overlap or cross-talk between channels is important, given the multiple models (both theoretical and computational) for basal ganglia functions, including those that rely on the idea of re-entrant loops vs cross-talk within a given area or pair of areas. In addition, using two-injection animals, they were able to show multiple segments of the closed basal ganglia loop. They also show evidence for a direct corticonigral projection, not previously described.

The large dataset represented by this manuscript is likely to be an important resource for basal ganglia researchers, testing existing hypotheses and correlating with other types of data that reflect basal ganglia connectivity and functionality. In the process of creating this large dataset, they also identify a couple of novel findings, but also have identified a number of broader themes in connectivity. The conclusions one can draw from this study, however, are limited somewhat by the lack of clear quantitation/statistics to support their conclusions, and the fact that the results are almost entirely anatomical, rather than physiological.

Major points:

1. The experiments are of interest to a broad group of investigators, but the presentation and interpretation of the findings is not always clear. I would advocate that the authors try to reorganize both the visual presentation (try to make more uniform across figures) and text. The text of the main results is very lengthy, and each section feels slightly disjointed from the others, rather than integrated in a larger plan. Given the large territory that is being covered, it might be helpful to either structure the Results section as answering a series of key questions (that may cut across different circuit segments) or by making the structure of each section more uniform (answer the same smaller questions for each connection that is examined).
2. The number of animals included is not clear in many of the experiments. For the main anatomical connectivity analyses, a large N is reported in the Methods, but it is unclear, for example, how many animals contributed to the result for one specific domain injection. Given that much of the results are descriptive in nature, and that in many cases examples are shown, it would be helpful to know how many individual animals were examined for each conclusion made. In Figure 8, they report in the Results section that in fact N=2, a very small number for the only physiological study in the paper. The lack of reporting # of animals weakens the conclusion and could affect the interpretation of the data. For an ambitious manuscript of this type, the N (animals) should be substantial and the N reported in the legends.
3. Though many of the results are descriptive in nature, the authors do make some conclusions which should be quantitatively justified. For example, they emphasize the convergence vs parallelism of direct vs indirect pathways. However, I could not find a clear quantification of these characterizations, nor a measure of the variance between animals in such a metric. For some of the bigger picture conclusions the authors would like to make, it would be useful to include quantitation and statistical measures.
4. Demonstrating connections through anatomical methods is extremely helpful. The authors do show some very limited physiology to support the closed loop functionality of the basal ganglia (Figure 8). However, some of the more interesting conclusions or findings, regarding convergence/parallelism, shared targets, and even the cortico-nigral pathway, are not backed up

with physiological assays. Synaptic strength varies greatly within and between the known connections of the basal ganglia loop, which complicates the picture. The manuscript would also greatly benefit from placing the anatomical findings here within the context of the many physiological studies of basal ganglia connectivity that have been done over the past several decades.

5. In Figure 7, the authors show data supporting a new pathway between cortex and SNr (direct corticonigral pathway). Their data include traditional anterograde tracing in the upper part of the figure, but the images only show terminals in SNr. It would be useful to include a schematic (like they do in the lower part of the figure) and representative histology showing the size of the injection in the cortex. This is especially relevant since any stray tracer in the striatum (directly underneath much of the cortex) would produce terminals in SNr. They also use a retrograde approach, with AAVretro, and identify corticonigral neurons. It would be helpful to show in a representative low-mag sagittal section the course of the projection, as well as terminal boutons of corticonigral axons in GPe, PF and CPi areas (not just SNr, as is currently shown).

Minor points:

1. Many images are missing scale bars (Fig. 2B, 3C, 4D, 5A...). The figures need to be reviewed and revised to include scale bars.
2. Figure 1A, B and C are from previous publications (Hintiryan, Foster et al. 2016); I would consider removing from Figure 1.
3. What is the size of a square grid space in Fig. 1H?
4. State which kind of anterograde tracers used in line 180.
5. Line 222-225, "Axons from CPr converge with axons from the other striatal divisions along the whole extent of the medial SNr: some CPi axons project medially and converge with CPr axons at levels 81-87, and both CPi and CPc send some axons to medial SNr at levels 89 and 91 (Figure 2A; Supplementary Figure 2A)." The projecting pattern author demonstrated here is not obvious in Fig 2A. For example, both CPi and CPr project medially at levels 89 and 91, not merely 81-87.
6. I found the analysis the authors conducted in line 250-252 and line 270-279 somewhat confusing. Are they different approaches (computational analysis vs. manual analysis). The authors might want to make it clear about different approaches.
7. Consider simplifying line 283-324.
8. Please refer to the correct figure in line 335.
9. In Figure 3C, Authors should include the number of animals used for the quantification, and the injection of rabies is in specific sub-region of PF? There are only 9 cells for quantification and the labeling is relatively sparse, could these bias the interpretations?
10. It's not clear that line 366 is referring to the correct figure.
11. Consider moving line 388-392 to methods section.
12. I would consider moving the section Line 429-441 to the discussion.
13. There seems to be an inconsistency between line 455 and Figure. 5A. In Figure 5A, the injection sites are CPi.dm and CPi.dl, while in line 455, they are cited as CPi.dm and CPi.vm.
14. The authors should provide data quantification to support the conclusion mentioned in Line 459-463.
15. In discussing the idea of the basal ganglia loop, the authors say no studies have demonstrated a functional loop. However, the authors have not cited Oldenburg and Sabatini (2015), a study in which the authors used optical stimulation of direct and indirect pathway neurons *in vivo*, and shown opposing modulation of motor cortex neuronal activity. The direct pathway stimulation would be hypothesized to represent striatum \diamond SNr \diamond thalamus \diamond motor cortex. The indirect pathway stimulation would be hypothesized to represent striatum \diamond GPe \diamond STN/SNr \diamond thalamus \diamond motor cortex. Though in the Sabatini study they did not march through every node in the circuit, all of the steps have been physiologically tested by this point.

Author Rebuttals to Initial Comments:

Referee #1 (Remarks to the Author)

This manuscript by Foster et al presents a new anatomical model of the cortico-basal ganglia (BG) circuits, based on a new and massive dataset. Extensive anatomical work over decades has produced a huge but often confusing literature, and numerous controversies remain. Foster et al use an impressive combination of modern anatomical tracers, unbiased digital reconstruction, automated image registration, and network structure analysis, to present a comprehensive connectome of key circuits in the BG. While confirming a few canonical features of the BG anatomy, albeit with finer resolution, this work also breaks new ground and reports a number of intriguing and surprising results, such as the detailed output channels in the SNr and parafascicular nucleus, the organized GPe output to the thalamus, and the highly parallel organization of GPe domains. The anatomical work is at once comprehensive and rigorous, and the results are of general interest, not only to basal ganglia researchers, but really to all students of neuroscience. It might become the new standard reference and an important resource for all neuroscientists.

However, in part due to the enormous complexity of the circuits covered, there are a number of weaknesses associated with the presentation and discussion of the data. In order for this to be used widely by researchers as a reference, clarity of exposition and illustration is critical. Unfortunately in its current form the manuscript is often difficult to follow. While the experiments are usually well conducted, the quality and extent of the analysis can be uneven. I hope the following remarks can be helpful to the authors as they prepare a revision.

Overview (response): We thank the reviewer for their positive assessment of the value of our work, and for their helpful constructive suggestions to improve the manuscript, which we have endeavored to implement, as described below in our responses to specific points raised.

Major points.

1. The sheer amount of information makes the text difficult reading, even for experts. At least I found it difficult to follow at times. To reach a broader audience, it is recommended that the authors use some schematics to illustrate the major principles derived from this work.

#1 response: We thank the reviewer for this feedback and recognize that this was the biggest area for improvement of the manuscript. We have tried to focus the main body of the text on a few major principles, and have edited the text to improve

generally readability. Additionally, as suggested, we have added 4 additional schematics to illustrate the major principles derived from the work.

2. Currently the paper cites many general references on BG function and disorders, but relevant anatomical papers with similar findings are not always adequately acknowledged.

#2 response: We have expanded our inclusion of relevant primary literature to better acknowledge the contribution of previous studies, including some excellent new studies published since our original submission (e.g., Lee et al. 2020, McElvain et al. 2021). A number of new references have been added throughout the manuscript, but in particular, paragraphs 3 (line 480), 5 (line 519), 6 (line 548), and 8 (line 594) of the Discussion now include many additional references to relevant anatomical and electrophysiological papers with similar and related findings.

3. The question of parallel organization vs. convergence, a key point of contention, is not discussed adequately. For example, see the old debate between Alexander et al and Percheron et al (TINS 1991). In addition, Figure 8 demonstrates a closed BG loop, but it would be useful to see an experiment which used injections in the orobranchial region of the PF but then used the striatal double-injections in a neighboring but separate loop, such as the trunk loop, to see how much the tracers are co-labeling in the SNR and in the cortex. Was this type of experiment done?

#3 response: We appreciate the reviewer pointing us to this useful background literature. We have expanded the Discussion in relation to the parallel vs. convergence (funneling) debate, adding a paragraph (Discussion paragraph 8, line 594) that includes the references recommended by the reviewer and other relevant studies. While most of our manuscript emphasizes the parallel aspects of the cortico-basal ganglia-thalamic loops that we have identified, this paragraph in the Discussion, along with paragraph 4 (line 505) of the Discussion, attempts to highlight the convergent aspects of these pathways.

The experiment recommended by the reviewer is an excellent idea, and we have carried it out. We conducted a double co-injection in two separate, neighboring loops. Specifically, we injected the CP of the trunk/lower limb loop and the thalamus (PF.associative domain) of the associative loop. The transported tracers occupy separate, neighboring domains of the SNr and regions of cortex. We present these new data in Extended Data Figure 15, and describe these results in Results section 'Demonstration of whole cortico-basal ganglia-thalamic loops' (line 405).

4. In an influential series of studies, Flaherty and Graybiel identified small and dispersed regions (matrisomes) in the striatum that receive divergent cortical inputs, but these striatal regions reconverge in the pallidum. Do present results shed light on this claim?

#4 response: We thank the reviewer for directing our attention to this relevant work. The matrisomes described by Flaherty and Graybiel for non-human primate appear similar to the domain structure of the mouse CP: they appear to be precisely wired connectivity patterns between corticostriatal input and striatopallidal output, highly similar to the precisely wired inputs and outputs we describe here for the striato-nigro-thalamic pathway (Figure 3), the striato-pallido-nigral pathway (Figure 5, Extended Data Figure 12), and the cortico-basal ganglia-thalamic loop in general (Figure 7, Extended Data Figure 14). Thus our present results appear to be in good accord with the findings of Flaherty and Graybiel, with these precisely wired connectivity patterns being a general feature of the cortico-basal ganglia-thalamic network. The matrisomes they describe do have a more complex structure than the mouse striatal domains, which is not too surprising since the primate cortico-basal ganglia-thalamo-cortical network is generally more complex compared to the mouse (e.g. as demonstrated by Hoover and Strick, 1993, Science). We have incorporated their findings into the Discussion (Line 493).

5. Discussion focuses on various broader implications. While such discussion is useful, it is also largely speculative. On the other hand, discussion of the relationship between current results and previous anatomical findings is too limited. It would be important to define exactly which findings are novel, and how they advance beyond previous work.

#5 response: This is a helpful suggestion. We have eliminated the more speculative parts of the Discussion, in particular the section speculating on the potential impact of our findings in models of stimulus-response and action-outcome learning. We have added a number of new paragraphs in the Discussion to relate each major new finding to the existing literature. Specifically, paragraph 3 (line 480) relates our new domain model to previous anatomical and electrophysiological findings, including reaching; paragraph 5 (line 519) discusses anatomical and electrophysiological data that are in accord with our finding of parallelism/convergence in the striatopallidal/striatonigral pathways, respectively; paragraph 6 (line 548) puts into context our finding of re-convergence of the

striatonigral and striato-pallido-nigral pathways; and the true closed-loop nature of the cortico-basal ganglia-thalamic loop is discussed in paragraph 2 (line 469).

6. In the Results, there is limited information on the type of viral injections and mice. Such information should be provided consistently for each experiment. On a related note, it is sometimes not clearly explained what tracer is being used in the legends or the figure. For example, in Figure 5A, the method is explained in the caption but not in the figure. Then in figure 5C, the caption only says 'an injection' - so it leaves the reader concluding that the methods used in figures 5A and C are identical. Other figures have better descriptions, but there is no uniformity in the communication of the technique used.

#6 response: We thank the reviewer for calling our attention to this lack of clarity. To improve communication of experimental techniques, we have ensured that detailed methodological information regarding the tracers is described for each experiment throughout the Results section and figure captions. Additionally, we have added information regarding numbers of mice used in each experiment, either in the main text or figure captions. Furthermore, to help convey the experimental approach utilized, we have added a number of injection strategy schematic illustrations (9 new illustrations). To address the lack of clarity in Figure 5, we have moved Figure 5a to Extended Data Figure 11, with a detailed description of this experiment given in the figure caption as well as in Methods; for Figure 5c, we have added an injection strategy schematic, and written the tracer type (AAV-RFP) on the injection site image (new Figure 5a).

7. They say that “many of the domains in the SNr formed by these inputs span multiple levels of the SNr, i.e., the SNr domains have a 3-dimensional shape.” a 3D rendering of this would be helpful.

#7 response: We concur that a 3D model of the SNr will be a helpful way to represent this dataset. We have endeavored to create this 3D model. Unfortunately at the present time we have been unable to complete it because the 3D visualization expert in our lab accepted employment with another company. Since we have yet to fill her position, work on the 3D model has temporarily stopped.

8. For tracer injections, they only show a schematic and a single representative example. More histology is needed to show their 33 injection sites. They say their data is fully presented on their connectome website, but I could not find it. Perhaps this could be made more explicit.

#8 response: We apologize for the missing online data presentation. As a result of disruptions to our information technology infrastructure associated with the recent relocation of our lab, online availability of the pathway tracing data in question is temporarily delayed. However, to ensure timely data availability, we have included a new supplementary figure (Extended Data Figure 1) showing all striatal tracer injections used in the striatal output analysis, complete with injection sites alongside anterograde terminal labeling arising from them in the globus pallidus external segment (GPe) and substantia nigra reticular part (SNr). Moreover, as soon as our web infrastructure is fully operational, in addition to making these data available online, we will also make available the following datasets: projection maps; 3D renderings of SNr neuron reconstructions; Excel spreadsheet tables of all quantified grid box data used for GPe and SNr domain analyses; all manuscript figures. We anticipate being able to make these data available at or very close to the publication date. [REDACTED]

9. Contrast enhancement appears excessive in some pictures (See fig 2b, 2d, 2f 4d,,8a, s6).

#9 response: The figures the reviewer refers to show injection sites, which tend to stand out because they are typically the location of strongest fluorescent reporter signal. To achieve unbiased data representation within and between experiments we applied consistent imaging exposure settings that were optimized for visual qualitative fidelity. This can at times result in an excessively bright injection site.

10. how did they control spread in their viral injections? It appears that they used iontophoresis for some, but they also just used pressure injections. This causes a decent amount of spread with the amount used (e.g. 50 nL). Moreover, how did the authors avoid tracer and or virus leaks into the cortex during insertion and withdrawal of the pipet? This could result in unintended cortico-cortical projection labeling, and could potentially alter interpretation of the results in Figure 8.

#10 response: Most injections were performed iontophoretically, including all viral-tracer and PHAL injections for analyses of GPe and SNr domains. Carefully controlled pressure injection was used to inject rabies virus, retrobeads, and AAV1.Cre for anterograde transsynaptic tracing (iontophoresis was used for other Cre-dependent tracing in Figure 8e. For all injections (pressure and iontophoretic), to reduce the possibility of extraneous tracer deposition, the outer barrel of every micropipette was wiped gently with a lint-free tissue prior to its insertion into the

brain, and each was left *in-situ* for 5 minutes at the site of injection prior to withdrawal to enable any buildup of pressure to dissipate. These control measures (as are typically used) helped to reduce the possibility of unwanted tracer deposition. Tracer injections that resulted in substantial amounts of unwanted (extraneous) labeling were not included in the analysis.

11. They say “Although not quantified, the density of the indirect pathway terminations in GPe appears denser than the direct pathway” (L458-459). They should either quantify this or remove the statement.

#11 response: We have removed the statement.

12. The 3D matrix (Figure 2H) that is presented not well explained and it is difficult to understand how to use it. Diagrams that explain the tracing experiments need to be more extensively used. Currently there is only an occasional graphical description of the injection strategy used, such as in Figures 7B or 8A. These should be more frequent in the paper.

#12 response: We thank the reviewer for identifying this point of confusion, and we hope the changes we have made have improved the utility of the 3D matrix. The 3D matrix has been moved to its own supplementary figure (Extended Data Figure 7), and now includes additional labeling of the axes, an extended explanation in the figure caption, and a usage example along with a detailed description.

The reviewer’s request for more illustrations of experimental strategy for the tracing experiments is another good suggestion. We have included 9 additional diagrams that explain the tracing experiments (in Figures 2a, 4a, 4f, 5a-d, 5f, 8a, and Extended Data Figures 13a, 14c, and 15). We have also added more figure labeling and textual descriptions in this regard to improve clarity.

13. It would be helpful to provide more quantitative analysis to support the major conclusions. The anatomical description is sometimes anecdotal. Perhaps in modifying the presentation of the 3D matrix, it could be used as a recurring motif to better communicate the statistical evidence for the claims that are being made in each figure - for example by highlighting the relevant cells of the matrix in each figure.

#13 response: This is an excellent suggestion. We have added more quantitative analysis to support the conclusions we drew based on the anatomy. A quantification of the parallelism/convergence of the striatopallidal/striatonigral pathways, respectively, has been conducted to validate our anatomical description of this

phenomenon; this is described in the text (Line 277) and depicted graphically in Figure 4e and Extended Data Figure 10b,g. Additionally, new electrophysiology experiments were carried out to demonstrate and validate the parallelism/convergence (Figure 4f; line 285) and direct/indirect pathway re-convergence (Figure 5f; line 327) phenomena we described anatomically; quantification of these electrophysiological data support our anatomical descriptions.

To avoid confusion (and to improve clarity) with respect to application of the 3D matrix, we now present it in a standalone supplementary figure with supporting explanatory legend (Extended Data Figure 7), in addition to keeping a small version of it in Figure 2c. The reviewer's suggestion to use the 3D matrix as a recurring motif is an interesting idea. In this instance its application (as used here) is limited to striatal-SNr connections, but we thank the reviewer for their creative suggestion, and will consider broader future applications.

14. Figure 2 and Figure 3B: The authors claim that the SNR domains are extended in the rostro-caudal axis and show that the axons from the striatum pass through this axis. What about synaptic contacts made?

#14 response: Although we did not demonstrate this ourselves, Levesque & Parent (2005) used single neuron reconstruction in primate to show that striatonigral axons have terminal boutons, boutons en passant, and synaptic varicosities all along their length in the nigra, in both the highly branched axonal terminal fields and in the straight unbranched regions as well. This conforms well with the concept of longitudinal domains. We have added a statement about this in our description of the striatonigral pathway (Line 209).

Minor points

(mostly grammar and usage, careful proofreading is needed as there are numerous examples)

15. L116: serial circuit? Not exactly serial if the loop is closed.

#15 response: That phrase has been removed.

16. L139: 'Comprised of.' Should be 'composed of' or 'comprise'. 'Each receive.'

17. L149: foreshortened? Not sure what is meant here. 'Inadequate' perhaps.

18. L264: 'inputting', providing input to?

19. L679: 'The major likeliest.' 20. L690: missing 'in' after 'resides'.

#16-19 response: We have amended the text to address the reviewer's points.

Referee #2 (Remarks to the Author)

In this manuscript Foster and colleagues perform a tour-de-force anatomical dissection of the cortico-basal ganglia- thalamic network in the mouse. Using advanced technical approaches that they have previously established, they performed over 700 injections of various tracers (mostly viral) in striatum, GPe, SNr, thalamus, and cortex. Using this rich dataset they have identified multiple structural subdivisions they call 'domains' in SNr (14), GPe (36), and parafascicular and ventromedial thalamic nuclei (6). They also state that they identified 6 parallel cortico-basal ganglia-thalamic subnetworks that "sequentially transduce specific subsets of cortical information with complex patterns of convergence and divergence through every elemental node of the entire cortico-basal ganglia loop". With these techniques they uncover several interesting properties of cortico-basal ganglia thalamic circuits, including the presence of direct cortico-nigral projections in the oro-brachial domain, nigral dendritic arbors that appear to conform to the size of their particular domain (with potentially some overlap at domain boundaries), and a higher degree of structural convergence within the striatonigral "direct" pathway than the striatopallidal 'indirect' pathway. In addition, a single example of a functional closed loop was provided via slice physiology in the thalamo-corticostriatal segment of the oro-brachial subnetwork. They conclude that these data provide significant conceptual advances regarding both the structural and functional organization of cortico-basal ganglia circuits.

These data provide an incredibly rich source of information regarding cortico-basal ganglia thalamic networks that will be a valuable resource for the field. Understanding structural and functional connectivity in these networks is very important for understanding the processes of action selection in both healthy and disease states, so the topic is quite significant. However, I have several significant issues with the manuscript as written as noted below.

Overview (response): We thank the reviewer for their overall positive assessment of the value of our work with respect to these data being a valuable resource for the field of sensory-motor neuroscience research (in general), and specifically in relation to cortico-basal ganglia-thalamic circuits. We are also grateful for their constructive critique. We have substantially revised the manuscript, in line with their critique, including numerous textual edits and stylistic changes, as well as additional figures and new data. These revisions are described below in response to specific points that were raised by the reviewer.

Major points.

1. The provided anatomical data is very interesting and important, but for many readers the most exciting part of the manuscript will be how to translate these structural findings into information regarding functional connectivity. From the abstract, it appeared that this would be one of the main points of the manuscript, but functional data were limited to Figure 8C. I would have liked to see more electrophysiologic analysis, ideally *in vivo*, to determine if the observed structural integration actually leads to functional integration. Related to Fig.8C, one of the most exciting aspects of the manuscript is the electrophysiologic (*ex vivo*) demonstration of a closed-loop circuit, which has been proposed as an important circuit motif but has been difficult to directly demonstrate. However, there are significant issues with the design and interpretation of this experiment. The authors explain that AAV1 can be used for anterograde transsynaptic infection of post-synaptic neurons in a particular pathway. However, they then go on to use AAV1 to attempt to perform typical anterograde (non-synaptic) infection of neurons. This is problematic, given that in later experiments, unintended transsynaptic infection could affect the results. This is a particular problem in the slice electrophysiology experiments, where use of AAV1 injected into thalamus could trans-synaptically infect cortical neurons. Thus, the optogenetically-evoked synaptic responses observed in cortical neurons may be coming from neighboring cortical neurons, and not thalamic neurons. As this finding is a key point of the paper, the authors should repeat these slice experiments with a non-transsynaptic AAV serotype. Additionally, the authors should increase power in this experiment, since they are currently drawing these conclusions from only 2 animals. In addition, they should make clear that these findings may or may not generalize to other cortico-basal ganglia-thalamic loops (i.e. even if this is a closed loop, other circuits may not be).

#1 response: The reviewer has raised several important points. Regarding the potential for false-positive electrophysiological data resulting from undesired transsynaptic AAV transport. The AAV1-ChR2 construct that we used was previously shown to not pass transsynaptically in sufficient quantity to produce electrophysiological responses in post-synaptic neurons (see Zingg et al., 2017, Fig. S3). This is also supported by use of the method in other recent optogenetic studies in tissue slices without evidence of confounding data (e.g., Baimel et al. 2019, Cell Reports; Doan et al. 2019, Cell Reports; Sermet et al. 2019, eLife). Similarly, use of related common AAV constructs by our lab and others for anterograde tracing has not produced evidence of visually detectable levels of the tracer in post-synaptic neurons. However, the critical difference between these techniques, and the method of Cre-dependent AAV tracing that *does* take advantage of the very low levels of transsynaptic AAV transport, is that in the latter case the very low quantity of transsynaptic transport is sufficient to enable amplification of Cre-dependent AAV expression (injected into the post-synaptic target region) to a detectable level.

We agreed with the reviewer's suggestion to increase the power of the electrophysiological data. Accordingly (despite disruptions associated with our lab's recent relocation and the pandemic), we were able to perform an additional experiment with a larger n (n = 4 mice) for the associative network loop. To investigate this, we injected two pathway tracers: 1) a retrograde tracer (red retrobeads) into a striatal target site (CP intermediate division, dorsomedial community, dorsomedial domain -- CPi.dm.d), resulting in retrograde neuronal labeling of the anterior cingulate area (ACA); 2) an anterograde tracer expressing ChR2 (AAV.ChR2) into a thalamic target site (parafascicular nucleus, associative subregion – PF.a), resulting in axonal labeling in the ACA. We then performed patch clamp recording of ACA neurons retrogradely labeled with retrobeads from the CP in while conjunction with optogenetic stimulation of ChR2-expressing anterogradely labeled PF.a. axons. In each of 9 patch clamp recordings (from 4 mice), electrophysiological responses to the stimulation were measured. These additional data are presented in Extended Data Figure 14c. The results of these new experiments concur with our original findings for the oro-brachial loop. Together, these findings are consistent with the hypothesis that the closed loop motif can be generalized, but we acknowledge the possibility of exceptions.

2. Related to this is the fact that one of the major claims of the manuscript is that there is clear evidence of segregated closed loops, implying that this is the general motif used by cortico-basal-ganglia-thalamus circuits. However, to make that claim, which

is exciting, there would need to be clear demonstrations of this phenomenon in other circuits as well. This is particularly the case since the authors demonstrated the closed-loop motif in the oro-brachial subnetwork, which was unique in having direct cortico-nigral projections. Thus, findings in this subnetwork may not generalize to other networks.

#2 response: This is an important point. As indicated above, we performed an additional electrophysiological recording experiment to test the hypothesis that the associative loop is a closed loop (ACA → CPi.dm.d → SNr.m → PF.a → ACA), as we show for the oro-brachial subnetwork (MOp-m/i → CPi.vl.v → SNr.orb → PF.m → MOp-m/i). Additionally, we anatomically validated this loop using the double coinjection technique, by injecting two anterograde/retrograde tracer pairs: 1) an AAV-GFP/CTB coinjection into the striatal target site (CP intermediate division, dorsomedial community, dorsal domain -- CPi.dm.d), resulting in retrograde neuronal labeling of the anterior cingulate area (ACA) and anterograde labeling in the substantia nigra reticular part, medial domain (SNr.m); 2) an AAV-RFP/Fluorogold coinjection into the thalamic target site (parafascicular nucleus, associative subregion – PF.a), resulting in anterograde labeling in ACA and retrograde labeling in the SNr.m. Critically, the labeling from the two coinjections occupies the same zones in ACA and SNr.m, demonstrating that the ACA, CPi.dm.d, SNr.m, and PF.a form a continuous, connected loop. These additional data are presented in Extended Data Figure 14a. Moreover, in a new experiment using the same technique, the anterograde/retrograde tracer pairs were injected into separate, neighboring loops, i.e., one co-injection was placed into the associative loop thalamic domain (PF.a) and the other injection was placed into the trunk/lower limb loop striatal domain (the trunk and lower limb domains of the CP, CPi.dl.d (tr) and CPi.dl.imd (ll)); these two loops are topographically adjacent. Their transported labeling occupied separate, adjacent regions of SNr and cortex (Extended Data Figure 15), again demonstrating that these loops appear to be segregated and that this is a general motif of the cortico-basal ganglia-thalamic circuits.

3. The authors show that there is a pathway from accumbens, to SNr, to PF and VM thalamus using a dual-virus trans-synaptic AAV1/cre strategy. They claim that, while they also observed infected neurons in VTA and SNc (neurons downstream of accumbens), none of these neurons are projecting to PF/VM thalamus, because they don't express TH. However, a substantial number of neurons in VTA are GABAergic or Glutamatergic, do not express TH, and project to various cortical and subcortical targets throughout the brain (Breton, 2019). Thus, a TH stain is not sufficient to rule

out projections to thalamus from VTA. The authors should do more to address this concern.

#3 response: The reviewer raises a good point. To investigate further output connections from the VTA, we performed anterograde tracing with PHAL injected into the VTA (to effect non-selective tracing of all neuron types). We found a weak density projection from VTA to both PF and VM (see Extended Data figure 13i-m) that stands in contrast to our existing demonstration of a very dense SNr to PF connection (Figure 6d,g images labeled 'SNr.dm'). We also observed very dense PHAL labeling in the CP, consistent with the classic strong dopaminergic projection to the striatum. This is evidence consistent with the labeling we show in Figure 6d,g arising from an SNr.dm->PF.vs pathway, although we acknowledge the possibility that the other regions labeled by our injection could contribute to the labeling we depict.

4. Because there is so much information in the manuscript, it can be difficult for the reader to distill the most important messages from the available information. Focusing in on one specific important question in the field (e.g. presence or absence of closed loops), and then digging into it more deeply, would make for a more compelling manuscript, as it would showcase the value of this resource.

#4 response: Following the reviewer's constructive suggestion, we have focused the manuscript to the few important points we wish to emphasize, and have structured the abstract, introduction, the subsections of the results, and the discussion to highlight these points and to follow a consistent presentation format. More tangential data and text has been moved to figure captions, extended data figures, the Discussion, or eliminated. We hope that this has resulted in a more 'reader-friendly' manuscript.

5. Although past work demonstrating convergence in cortico-basal ganglia thalamic loops is acknowledged (e.g. from Suzanne Haber's lab), and the authors state they are not suggesting this is not present, the manuscript overall does not seem to give these data much weight. I suggest a more balanced discussion of the two alternative views, especially because direct assessments of neuromodulatory systems that might serve as agents for spiraling loops were not assessed in this manuscript.

#5 response: We now elaborate on the convergence data in the Results (Lines 272-301) and Discussion (paragraph 2, line 505-518, and paragraph 8, lines 594-624), particularly within the striatonigral pathway where we see the most convergence. Extended Data Figure 10 presents this data in the form of a histogram, and we will

make the raw data available online. We would like to convey that the manuscript is weighted towards discussion of the parallel features of neighboring cortico-basal ganglia loops because it is a reflection of the underlying data, data which strongly indicated precise intraconnectivity within subnetwork loops and parallelism between subnetwork loops. We have further elaborated this parallelism with a new double co-injection experiment in separate, neighboring loops (Extended Data Figure 15), which again validated the parallel nature of subnetwork loops as a general motif of the cortico-basal ganglia-thalamic circuit. This striking parallelism between subnetworks has also been noted in other recent studies of the basal ganglia (e.g., Mandelbaum et al. 2018, *Neuron*; Lee et al. 2020, *Nature Neuroscience*). Admittedly, nigrostriatal, corticocortical, and divergent thalamocortical connections represent possible areas where more convergence and overlap could occur between loops, as it was beyond the scope of the current manuscript to thoroughly analyze these pathways; this has been included in the Discussion (line 605).

6. The authors pay special attention to a circuit involving GPe.3 and SNr.m, showing that both regions receive inputs from the same several regions on the medial wall of the striatum. While they show later that, in general, direct and indirect pathway striatal pathways converge into the same areas of SNr, they don't explicitly show that the GPe.3 region projects to SNr.m. This fact should be explicitly shown in the section when they are discussing the case of GPe.3 and SNr.m.

#6 response: Another excellent suggestion. We have added a figure showing the GPe.3 connection to the SNr.m (Extended Data Figure 12) and describe this directly in the results (line 321).

7. Please clarify how representative injections were chosen, and how were possible differences in injection volume/cell transduction by virus were accounted for in the analysis procedures (e.g. was there normalization for this?).

#7 response: Criteria for selection of representative injection data (now stated at the start of the Results) were as follows: restriction of injection site to region of interest, quality of axonal labeling and tissue histology. In addition to careful neuroanatomical analysis, our use of cluster analysis (community detection with the Louvain algorithm) enabled us to identify divisions based on statistical grouping.

A known limitation of the pathway tracing method is that every tracer injection is unique (it is impossible to reproduce the same volume, placement, and distribution of tracer molecules). However, careful selection of the best injections using the criteria

noted above, combined with careful qualitative (neuroanatomical) and appropriate statistical analysis enables the identification of reproducible differences in connections between different regions. A further consideration is the spatial resolution of the investigations because the analysis is applicable only at the observed resolution. The present analysis is applied at the resolution of gray matter regions and their subdivisions (the resolution of the present analysis) but not at the level of individual neurons (microscale). If marked differences are found at the resolution used for a given division, then subdivision is suggested; likewise, if none are found then a division can be considered as a single entity (at that resolution). This principle is illustrated in the figure below showing two differently sized injection sites that were centered in the same CP division but produced highly comparable labeling in the GPe. For further discussion of this topic see Swanson and Lichtman, 2016, Annual Review of Neuroscience.

(Figure above): Two injection sites following PHAL and AAV-RFP injections, centered in the same CP domain (in the same animal). The cross-sectional area of the AAV-RFP injection is nearly double that of the PHAL injection, with an overlap of 11.2%. Despite the difference in size and location within the CP domain, the injections produce nearly identical terminal field patterns in the GPe and SNr.

Minor points.

1. In the first paragraph of the Results, the authors simply say “injections of anterograde tracers were made ...” – this should be more descriptive (were AAVs used? PHAL?).

#1 response: We have elaborated the description.

2. Figure 1E is not very clear. An outline or darker background on the brain slices may help clarify that there are two slices of brain being showed there. Also eliminating the overlap between the two slices may help.

#2 response: We have added outlines and reduced overlap.

3. The point being made about striatal projections along the rostrocaudal range of the SNr is a bit confusing. “Even though the striatal projections to the GPe innervate only a restricted rostrocaudal range of the pallidum, projections from GPe to nigra innervate its entire rostrocaudal length..” The figures pointed to (4A, 5B), to me, look like striatal projections to GPe do pretty much cover the entire rostrocaudal length. If this is an important point to make, the authors should clarify what they mean. If it is not important, this section in the paper should be limited to the convergence of striatonigral and pallidonigral projections, as this is a much more interesting and important point.

#3 response: We thank the reviewer for pointing this out. We have amended the text to clarify the intended meaning: projections from an individual CP domain innervate a restricted rostrocaudal portion of the GPe, while innervating the entire rostrocaudal range of the SNr; also, considered as a whole, the CP innervates the full rostral-caudal extent of both the pallidum and substantia nigra.

Referee #3 (Remarks to the Author)

In this manuscript by Foster et al, the authors present extensive rodent neuroanatomical tracing studies to characterize basal ganglia networks. This work builds on studies performed in the 1970s-90s using traditional anterograde and retrograde tracing methods in nonhuman primates and rodents, as well as their own more recent work using high-throughput analysis of tracing to describe cortico-striatal

pathways in rodents. The current study's central premise is that the traditional 3-channel view of cortico- basal ganglia-thalamocortical pathways (sensorimotor, associative, and limbic), which in turn was based on older tracing studies from the 70s-90s, does not capture the complexity of the connections within this circuit, and that there are many more channels. While in their previous study, the authors focused on corticostriatal connections, in this study they performed numerous injections to examine the succeeding steps in the cortico-basal ganglia-thalamocortical circuit: the GPe, SNr, and thalamus. These studies are powerful in that they provide a more extensive "catalog" of the connections, indicating areas of convergence, divergence, and overlap between channels. Evidence of such overlap or cross-talk between channels is important, given the multiple models (both theoretical and computational) for basal ganglia functions, including those that rely on the idea of re-entrant loops vs cross-talk within a given area or pair of areas. In addition, using two-injection animals, they were able to show multiple segments of the closed basal ganglia loop. They also show evidence for a direct corticonigral projection, not previously described.

The large dataset represented by this manuscript is likely to be an important resource for basal ganglia researchers, testing existing hypotheses and correlating with other types of data that reflect basal ganglia connectivity and functionality. In the process of creating this large dataset, they also identify a couple of novel findings, but also have identified a number of broader themes in connectivity. The conclusions one can draw from this study, however, are limited somewhat by the lack of clear quantitation/statistics to support their conclusions, and the fact that the results are almost entirely anatomical, rather than physiological.

Overview (response): We thank the reviewer for their assessment of the value and importance to the field of the datasets we have generated. Regarding their view that the conclusions are limited due to lack of clear quantitation/statistics, we have endeavored in our revision (as detailed in our responses below) to improve their clarity substantially. Regarding the conclusions being limited due to their being primarily neuroanatomical in nature, we concur that the conclusions one can draw in any study are limited by the nature of the data generated. However, there is long tradition of neuroanatomical research providing a foundation and springboard for future functional investigations. This is in keeping with the focus of our research group, and it was not within the scope of this study that was conceived originally as purely neuroanatomical (in keeping with our lab's research focus and resources) to expand the physiological aspects of the research. Nevertheless, we thought the reviewer's suggestion to add more electrophysiological data was a valuable one that

would greatly enhance this manuscript, and we exerted a great deal of effort to realize the additional experiments. The physiological data that we were able to generate do support the conclusions based on the underlying network structure. In addition, the very extensive and rich neuroanatomical data we provide, constitute a rich resource for the development and testing of functional hypothesis in the future by other groups (including the possibility for future collaborative efforts). We are grateful for the reviewer's helpfully constructive critique and provide responses below to their specific points.

Major points.

1. The experiments are of interest to a broad group of investigators, but the presentation and interpretation of the findings is not always clear. I would advocate that the authors try to reorganize both the visual presentation (try to make more uniform across figures) and text. The text of the main results is very lengthy, and each section feels slightly disjointed from the others, rather than integrated in a larger plan. Given the large territory that is being covered, it might be helpful to either structure the Results section as answering a series of key questions (that may cut across different circuit segments) or by making the structure of each section more uniform (answer the same smaller questions for each connection that is examined).

#1 response: We thank the reviewer for their helpful suggestions, which we followed in our revision. Accordingly, we have reorganized the text around addressing a series of key questions. These questions are posed first in the Introduction, then the sections of the Results address these questions in the same order, and finally they are revisited in the Discussion. Additionally, we have simplified several figures, and have reorganized these so that material less central to the main points are now included as Extended Data rather than within the main body of the manuscript and main figures. We have also increased uniformity across figures by: providing injection strategy diagrams for more experiments (9 new diagrams), and injection strategies for the same technique were kept consistent in appearance; by presenting summary diagrams related to the main points (4 new diagrams added); and by making graphs more uniform across figures. Admittedly, increasing the uniformity of figures was difficult because most figures are particular to the key question they are addressing, involving different experimental techniques at each step of the cortico-basal ganglia-thalamic loop.

2. The number of animals included is not clear in many of the experiments. For the main anatomical connectivity analyses, a large N is reported in the Methods, but it is unclear, for example, how many animals contributed to the result for one specific domain injection. Given that much of the results are descriptive in nature, and that in many cases examples are shown, it would be helpful to know how many individual animals were examined for each conclusion made. In Figure 8, they report in the Results section that in fact N=2, a very small number for the only physiological study in the paper. The lack of reporting # of animals weakens the conclusion and could affect the interpretation of the data. For an ambitious manuscript of this type, the N (animals) should be substantial and the N reported in the legends.

#2 response: We concur with the reviewer's assessment, and have made changes to address this deficit. We have clarified all statements regarding the number of animals used for each experiment, either in the main text or in the accompanying figure captions (or both). The ~700 injections we report is an approximation of the total number analyzed during the research (including review of archival data from >10 years of pathway tracing data from of our group). The figure of ~700 was based on an average of 3-4 injections/animal multiplied by the sum of data from some 68 archival animals and the remainder from the 200 animals from which new data were generated for this research (less those that were not included due to data of insufficient quality).

For our major analysis of the striatal output pathway, a total of 138 animals received a total of 448 striatum-targeted tracer injections; 36 representative injections (from 29 animals) were selected from this data set. Criteria for selection of representative injection data (now stated at the start of the Results) were as follows: restriction of injection site to region of interest, quality of axonal labeling, and tissue histology.

Regarding the number of animals used for the supporting physiological data, despite disruptions associated with our lab's recent relocation and the pandemic, we were able to perform an additional series of experiments: a test of the closed-loop nature of the associative network loop (ACA->CPi.dm.d->SNr.m->PF.a->ACA) with n=4; a test of the parallelism vs convergence of the striatopallidal vs striatonigral pathways with n=3; a test of the convergence of the direct (striatonigral) and indirect (striato-pallido-nigral) pathways onto individual neurons in the SNr with n=7. The results of the latter two new experiments support the major anatomical findings we report. The new associative network loop experiment concurs with our original findings for the oro-brachial loop (MOp-m/i->CPi.vl.v->SNr.orb->PF.m->MOp-m/i).

Together, these findings are consistent with the hypothesis that the closed loop motif can be generalized, but we acknowledge the possibility of exceptions.

3. Though many of the results are descriptive in nature, the authors do make some conclusions which should be quantitatively justified. For example, they emphasize the convergence vs parallelism of direct vs indirect pathways. However, I could not find a clear quantification of these characterizations, nor a measure of the variance between animals in such a metric. For some of the bigger picture conclusions the authors would like to make, it would be useful to include quantitation and statistical measures.

#3 response: This is an excellent suggestion. We have added more quantitative analysis to support the conclusions we drew based on the anatomy. A quantification of the parallelism/convergence of the striatopallidal/striatonigral pathways, respectively, has been conducted to validate our anatomical description of this phenomenon; this is described in the text (Line 277) and depicted graphically in Figure 4e and Extended Data Figure 10b,g. For all variables statistically analyzed, central tendencies of the data are given as mean±standard deviation, either in the main text, figure captions, or both. Box plots of all quantified data also give a clear visual representation of the range and variability of the data, by depicting min/max, first and third quartiles, median, and mean for each data group. Additionally, new electrophysiology experiments were carried out to demonstrate and validate the parallelism/convergence (Figure 4f; line 285) and direct/indirect pathway re-convergence (Figure 5f; line 327) phenomena we described anatomically; quantification of these electrophysiological data support our anatomical descriptions.

4. Demonstrating connections through anatomical methods is extremely helpful. The authors do show some very limited physiology to support the closed loop functionality of the basal ganglia (Figure 8). However, some of the more interesting conclusions or findings, regarding convergence/parallelism, shared targets, and even the cortico-nigral pathway, are not backed up with physiological assays. Synaptic strength varies greatly within and between the known connections of the basal ganglia loop, which complicates the picture. The manuscript would also greatly benefit from placing the anatomical findings here within the context of the many physiological studies of basal ganglia connectivity that have been done over the past several decades.

#4 response: We thank the reviewer for this great suggestion. We have performed additional physiological experiments to support our anatomical findings, including the 3 noted above (in response #2). Additionally, we have included another experiment regarding the corticonigral pathway derived from pathway tracing of the mouth area of the motor cortex: AAV1.hSyn.Cre.WPRE injections in Ai14 mice (Figure 8a), releasing expression of tdTomato reporter in these specific cortical neurons, and resulting in robust anterograde labeling, including within the substantia nigra. In addition, using this method allowed us to also take advantage of transsynaptic transport of the AAV construct for Cre-based amplification, demonstrating functional synapses in this pathway since only functional synapses can transmit the virus anterogradely (Zingg et al. 2020). These new experiments greatly strengthen our conclusions. We have also incorporated previous physiological findings in our discussion of the data we present, in particular in Discussion paragraphs 3 (line 480), 5 (line 519), and 6 (line 548).

5. In Figure 7, the authors show data supporting a new pathway between cortex and SNr (direct corticonigral pathway). Their data include traditional anterograde tracing in the upper part of the figure, but the images only show terminals in SNr. It would be useful to include a schematic (like they do in the lower part of the figure) and representative histology showing the size of the injection in the cortex. This is especially relevant since any stray tracer in the striatum (directly underneath much of the cortex) would produce terminals in SNr. They also use a retrograde approach, with AAVretro, and identify corticonigral neurons. It would be helpful to show in a representative low-mag sagittal section the course of the projection, as well as terminal boutons of corticonigral axons in GPe, PF and CPi areas (not just SNr, as is currently shown).

#5 response: We have supplemented the traditional anterograde tracing data with additional supporting images of injection sites; these corticonigral data are now presented separately as Extended Data Figure 16. All injection sites are focal, limited in size, and within the cortex, and in none of these cases is any extraneous labeling seen in the striatum. For the main figure, we have included data from an additional experiment depicting the corticonigral pathway in sagittal section (AAV1.Cre was injected into the mouth primary motor cortex of Ai14 mice; Figure 7a). This illustrates the corticonigral pathway within the context of the larger cortical output pathway. However, to avoid a possible ambiguity, we chose to not show high resolution images of axons in the GPe, PF, and CPi because they may have arisen from cortical neurons that do not project to the substantia nigra (even though axons of the latter are also

present). This experiment did give the benefit of allowing an anterograde transsynaptic labeling of postsynaptic neurons in the SNr (Figure 8c), demonstrating this pathway in yet another way. And corticonigral axon collaterals are shown in coronal section in CP, GPe, PF, and STN in Figure 8i-l.

Minor points.

1. Many images are missing scale bars (Fig. 2B, 3C, 4D, 5A...). The figures need to be reviewed and revised to include scale bars.

#1 response: We have added scale bars throughout.

2. Figure 1A, B and C are from previous publications (Hintiryan, Foster et al. 2016); I would consider removing from Figure 1.

#2 response: We considered removing Fig 1A, B and C, but decided to keep them in place because they serve to summarize preceding work from our group regarding the organization of the corticostriatal pathway that is foundational to the current study (that is a logical extension). We think this may be especially helpful for a broader readership.

3. What is the size of a square grid space in Fig. 1H?

#3 response: Each grid square (box) is 105 x 105 pixels, equating to about 63 μm^2 . The grid box dimension is now given in Figure 1h and is stated in the Methods.

4. State which kind of anterograde tracers used in line 180.

#4 response: We have added the names of the anterograde tracers (PHAL, AAV-GFP, and AAV-RFP) to that section.

5. Line 222-225, "Axons from CPr converge with axons from the other striatal divisions along the whole extent of the medial SNr: some CPi axons project medially and converge with CPr axons at levels 81-87, and both CPi and CPc send some axons to medial SNr at levels 89 and 91 (Figure 2A; Supplementary Figure 2A)." The projecting pattern author demonstrated here is not obvious in Fig 2A. For example, both CPi and CPr project medially at levels 89 and 91, not merely 81-87.

#5 response: We have amended the description to avoid ambiguity.

6. I found the analysis the authors conducted in line 250-252 and line 270-279 somewhat confusing. Are they different approaches (computational analysis vs. manual analysis). The authors might want to make it clear about different approaches.

#6 response: We have clarified that computational data analyses were performed for each level of the substantia nigra and GPe. We then took the results of these analyses and manually juxtaposed computed domains with substantially similar input patterns. For sake of clarity, these points are now described in paragraph 1 of Results (lines 191-195), and are also elaborated the Methods section (lines 814-830).

7. Consider simplifying line 283-324.

#7 response: We have simplified the text.

8. Please refer to the correct figure in line 335.

#8 response: We have corrected the labeling error.

9. In Figure 3C, Authors should include the number of animals used for the quantification, and the injection of rabies is in specific sub-region of PF? There are only 9 cells for quantification and the labeling is relatively sparse, could these bias the interpretations?

#9 response: Animal number used for quantification (n=1) is now stated in the caption for Figure 3C. Subregional PF specificity of the injection is not given. Interpretative bias is always a possibility (regarding the n= 9 for the reconstructed neurons); however, the morphometric values we obtained are consistent with previous reports (for examples see: Juraska et al. 1977; Yelnik et al. 1987). We have revised the text in relation to these points (lines 495-498).

10. It's not clear that line 366 is referring to the correct figure.

#10 response: We have clarified the reference.

11. Consider moving line 388-392 to methods section.

#11 response: We have moved lines 388-392 to the Methods.

12. I would consider moving the section Line 429-441 to the discussion.

#12 response: We have moved the section to the legend of Extended Data Figure 12, GPe.3 and retrograde tracing of SNr.m.

13. There seems to be an inconsistency between line 455 and Figure. 5A. In Figure 5A, the injection sites are CPi. dm and CPi.dl, while in line 455, they are cited as CPi.dm and CPi.vm.

#13 response: We have amended the text, and the information is now included in the legend of Extended Data Figure 11.

14. The authors should provide data quantification to support the conclusion mentioned in Line 459-463.

#14 response: We have removed the statement (although we attempted to generate additional data to support the conclusion, were unable to obtain this in a timely manner due to logistical challenges to arranging this with one of our collaborating groups).

15. In discussing the idea of the basal ganglia loop, the authors say no studies have demonstrated a functional loop. However, the authors have not cited Oldenburg and Sabatini (2015), a study in which the authors used optical stimulation of direct and indirect pathway neurons in vivo, and shown opposing modulation of motor cortex neuronal activity. The direct pathway stimulation would be hypothesized to represent striatum @ SNr @ thalamus @ motor cortex. The indirect pathway stimulation would be hypothesized to represent striatum @ GPe @ STN/SNr @ thalamus @ motor cortex. Though in the Sabatini study they did not march through every node in the circuit, all of the steps have been physiologically tested by this point.

#15 response: We thank the reviewer for bringing a relevant study (Oldenburg and Sabatini, 2015) to our attention. This was a superb study with fascinating results, but after careful review, we conclude that they did not show the corticostriatal stretch of the loop. Although they recorded from hundreds of cortical neurons while stimulating either direct or indirect pathway, and it seems likely they recorded from at least a few corticostriatal neurons, this was not unambiguously shown. One possible alternative explanation for their findings is that they recorded from cortical neurons projecting to the spinal cord, but not to the basal ganglia. We are aware that other stretches of the loop have also been demonstrated previously (for example, studies by Deniau and Chevalier (1985) and recently Lee et al. (2020), also demonstrating $\frac{3}{4}$ of the loop). However, to our knowledge ours is the first study to demonstrate simultaneously every node of the cortico-striato-nigro-thalamic loop in a single experiment, and the first to our knowledge to demonstrate

electrophysiologically that the thalamic output feeds directly back onto the corticostriatal input within a loop.

Reviewer Reports on the First Revision:

Referee #1 (Remarks to the Author):

The authors have revised the manuscript extensively and addressed my concerns. I appreciate the insightful discussion of convergence vs. parallel organization. This will be a very useful and influential resource for basal ganglia research. Henry Yin

Referee #2 (Remarks to the Author):

We commend the authors for thoroughly responding to our questions and concerns, including the addition of new experiments. We think that this manuscript will be an excellent resource for the field.

Referee #3 (Remarks to the Author):

The authors have made excellent additions and revisions to their manuscript, and it much more effectively conveys their conclusions to a broad audience. Though they have not addressed every concern or request made by the reviewers, they have made a good faith effort on most fronts and the writing and graphical presentation is much, much clearer. At this point, I suspect the data will be a very valuable resource for people in the basal ganglia field, as well as an example of how detailed connectomic analyses can address important circuit-level questions.

I have only a few remaining comments/questions regarding the revised manuscript:

1. line 222-231/ Fig 3a. The transition from describing striatum \rightarrow SNr projections to "different striatal domains receive distinct cortical inputs" in this paragraph felt abrupt, particularly because it drew on the group's previously published work on corticostriatal projections. It would make more sense to move this section to the "Demonstration of whole cortico-basal ganglia-thalamo-cortical loop" section, which is more integrative in nature. Also, the font size in the fig.3a (left corner one: lateral corticocortical subnetworks) is extremely small, making it hard to really glean much from the data as displayed.

2. In Fig 3c, the authors include a scale bar for the whole brain image, but not for the reconstruction of SNr. This scale bar is important to add, as they go on to quantify the dendritic length in the figure.

3. Fig 8k/line 445. In the example image, collaterals from the cortex to SNr appear to be located in the oromotor regions of the GPe, but also there seem to be red fibers in other domains within GPe. Are these not terminals but fibers of passage? Or are these other terminals due to leak (off-target expression) of the virus? I do not think there is an image for the other animal, but whether or not terminals are restricted to the oromotor region seems important to the model proposed in fig. 8n.

Author Rebuttals to First Revision:

Responses to reviewers' final comments. We are grateful for the excellent feedback from all three reviewers. Without a doubt their suggestions and critiques helped to strengthen the paper, both in terms of scientific rigor and of communicating the findings to other scientists and readers. Reviewer 3 had several more questions regarding the revised manuscript.

Question. “line 222-231/ Fig 3a. The transition from describing striatum-> SNr projections to “different striatal domains receive distinct cortical inputs” in this paragraph felt abrupt, particularly because it drew on the group’s previously published work on corticostriatal projections. It would make more sense to move this section to the “Demonstration of whole cortico-basal ganglia-thalamo-cortical loop” section, which is more integrative in nature. Also, the font size in the fig.3a (left corner one: lateral corticocortical subnetworks) is extremely small, making it hard to really glean much from the data as displayed.”

Response. We thank the reviewer for pointing out this non sequitur. We have rewritten that passage to have a clearer logic and rationale to it, and have removed the technical jargon that was occluding the main message we were trying to convey. Although we ultimately left the passage in place, rather than removing it to the “Whole loop” section, we believe that it has more of a logical conceptual connection to the rest of the section now. The new paragraph reads as follows:

The topographical organization of the striatonigral (direct) pathway is depicted in the projection maps (Fig. 1a, Extended Data Fig. 3a, 4a), and aligns well with data from rat and monkey^{8,10,12,24}. Following network analysis of this data, we identified 14 domains in SNr (Figure 1b-c; Extended Data Fig. 5-7; Supplementary Table 1). Most SNr domains receive convergent inputs from multiple striatal domains (Supplementary Table 1), each of which in turn receives a unique set of distinct cortical inputs⁵. For instance, SNr dorsal (SNr.d) and dorsomedial (SNr.dm) are limbic domains that collectively receive inputs from a number of limbic striatal domains (Extended Data Fig. 8a-b). These striatal domains in turn receive inputs from limbic cortical areas that are themselves interconnected and constitute the lateral cortico-cortical subnetworks^{5,25}, which are involved in perception of internal states and memory associated with emotion^{25,26}.

In the case of the limbic SNr domains we describe, providing this chain of connectivity from cortex to striatum to nigra provides insight into why we identified SNr.dm and SNr.d as limbic domains. We have also enlarged the text in the accompanying figure to ensure it is legible.

Question. In Fig 3c, the authors include a scale bar for the whole brain image, but not for the reconstruction of SNr. This scale bar is important to add, as they go on to quantify the dendritic length in the figure.

Response. Good point, we have added a scalebar to the inset.

Question. Fig 8k/line 445. In the example image, collaterals from the cortex to SNr appear to be located in the oromotor regions of the GPe, but also there seem to be red fibers in other domains within GPe. Are these not terminals but fibers of passage? Or are these other terminals due to leak (off-target expression) of the virus? I do not think there is an image for

the other animal, but whether or not terminals are restricted to the oromotor region seems important to the model proposed in fig. 8n.

Response. We thank the reviewer for pointing out this important issue. And the reviewer is correct, these are fibers of passage passing through the internal capsule and medial edge of the GPe. They are fasciculated into thick axonal bundles, which is why the labeling looks so bright, and why we know they are not a terminal field. The cell bodies giving rise to these axons are located only in the MOp-m/i cortical area. Once more, we thank all the reviewers for their valuable feedback.